# Larger colony sizes favoured the evolution of more worker castes in ants

Louis Bell-Roberts [1] ✉, Juliet F. R. Turner[1], Gijsbert D. A. Werner[1,2], Philip A. Downing [3], Laura Ross [4] & Stuart A. West [1]

The size–complexity hypothesis is a leading explanation for the evolution of complex life on earth. It predicts that in lineages that have undergone a major transition in organismality, larger numbers of lower-level subunits select for increased division of labour. Current data from multicellular organisms and social insects support a positive correlation between the number of cells and number of cell types and between colony size and the number of castes. However, the implication of these results is unclear, because colony size and number of cells are correlated with other variables which may also influence selection for division of labour, and causality could be in either direction. Here, to resolve this problem, we tested multiple causal hypotheses using data from 794 ant species. We found that larger colony sizes favoured the evolution of increased division of labour, resulting in more worker castes and greater variation in worker size. By contrast, our results did not provide consistent support for alternative hypotheses regarding either queen mating frequency or number of queens per colony explaining variation in division of labour. Overall, our results provide strong support for the size–complexity hypothesis.

Division of labour has played a pivotal role in the evolution of life on earth, by facilitating evolutionary transitions to more complex organisms[1–4]. In these transitions, individuals cooperate to form a new higher-level individual (organism). For example, cells formed multicellular organisms, with germline and soma, and insects formed complex colonies termed 'superorganisms', with queen and worker castes[3]. Division of labour is fundamental to these transitions because specialization allows the individuals that form a higher-level organism to perform more diverse functions and to become more reliant upon each other for reproduction (mutual dependence)[4]. Consequently, a major aim in evolutionary biology is to understand the factors that favour an increased level of division of labour[2,5,6].

The size–complexity hypothesis is one of the leading explanations for the evolution of increased division of labour because of the generality of its explanatory power across different levels of life. In lineages that have undergone a major evolutionary transition, such as to obligate multicellularity or superorganismality, the hypothesis predicts that larger numbers of lower-level individuals (subunits) select for increased division of labour[1,2,6–9]. This prediction arises because organisms or superorganisms formed by larger numbers of individuals have more tasks that require doing and are better able to maintain the optimal ratio of specialized individuals performing each task[2,6,10,11]. Support for the size–complexity hypothesis has been provided by examining division of labour in both individual multicellular organisms and insect colonies. Across multicellular species, organisms with larger numbers of cells have more cell types[7–9]. Across ants, species with larger colony sizes have more worker castes[12–17]. However, despite these correlations, challenges to the size–complexity hypothesis remain.

First, alternative factors have been suggested to explain variation in the level of division of labour. For example, in ants, variation in both queen mating frequency and number of queens per colony has been predicted to influence the evolution of number of worker castes[6,18,19]. When colonies are formed by a single monogamous pair, relatedness among colony members will be maximal, which may

[1]Department of Biology, University of Oxford, Oxford, UK. [2]Netherlands Scientific Council for Government Policy, The Hague, The Netherlands. [3]Ecology & Genetics Research Unit, University of Oulu, Oulu, Finland. [4]Institute of Ecology and Evolution, University of Edinburgh, Edinburgh, UK. ✉e-mail: louis.bellroberts@gmail.com

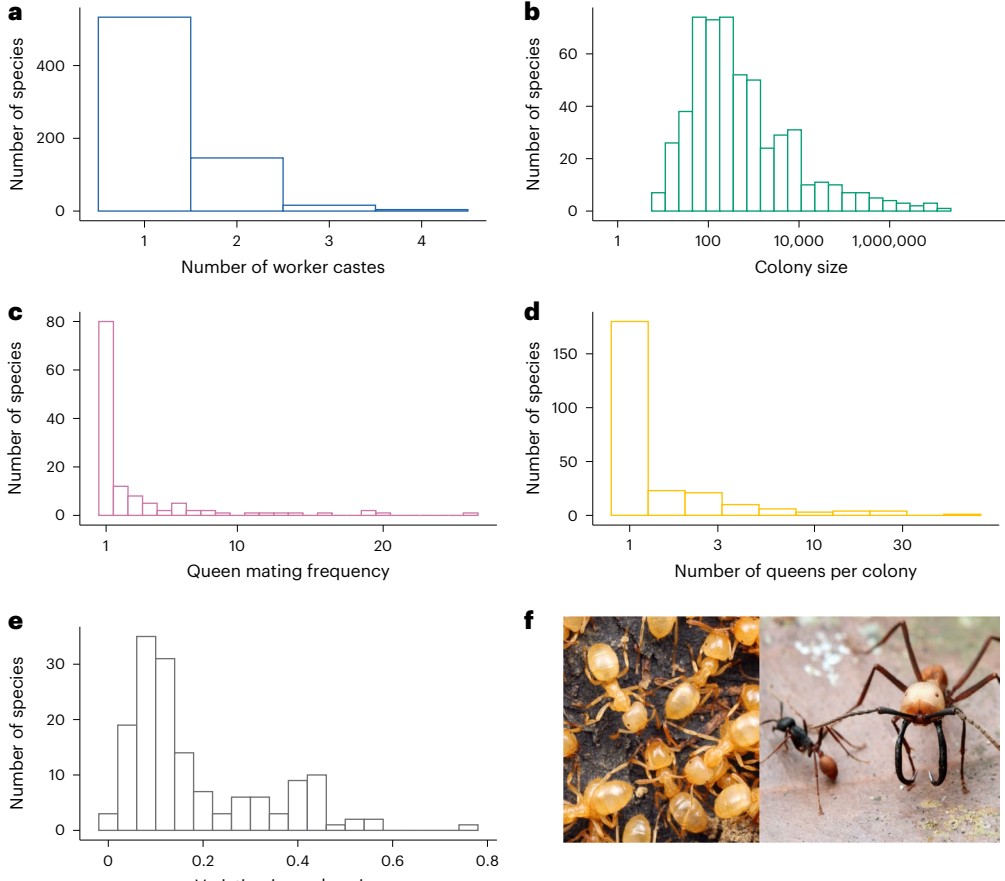

**Fig. 1 | Distribution of colony characteristics across ant species. a**, Number of worker castes ($n_{species}$ = 699). **b**, Colony size ($n_{species}$ = 541). **c**, Queen mating frequency ($n_{species}$ = 126). **d**, Number of queens per colony ($n_{species}$ = 252). **e**, Variation in worker size quantified using the CV for worker head width ($n_{species}$ = 152). **f**, Ants vary in their number of physical worker castes. Left: *Lasius claviger*, a species characterized by having just a single physical worker caste. Right: Two distinct physical worker castes of the same species, *Eciton burchelli*, representing one of the most extreme cases of morphological caste variation (photo by Alex Wild (https://www.alexanderwild.com/)). Axes for colony size and number of queens per colony are plotted on a $\log_{10}$ scale.

reduce conflict within the colony and favour the evolution of multiple worker castes[6,19]. Alternatively, the opposite prediction could also be made in hymenopteran species, where multiple mating by the queen selects for workers to remove (police) the eggs laid by other workers[20]. Policing could reduce conflict, leading to a positive correlation between the number of castes and queen mating frequency[18]. The problem of multiple hypotheses is exasperated by the fact that colony size can be correlated with queen mating frequency[13,14,21]. Consequently, the correlation between colony size and number of castes could just be the result of a relationship caused by variation in queen mating frequency.

Second, in superorganismal social insects, the correlation between colony size and number of worker castes is open to alternative causal explanations. For example, while larger colony sizes may favour the evolution of greater division of labour, it could alternatively be that increased division of labour favours the evolution of larger colony sizes. The problem of causal direction becomes even greater when we consider that there are multiple correlated variables. For example, it could be that an increase in colony size favours an increase in queen mating frequency, and it is an increase in queen mating frequency that favours the evolution of a greater number of worker castes. In this case, colony size and the number of castes would be correlated. However, this pattern would lead to the false impression that larger colony size directly favours increased division of labour and misses the involvement of an intermediate factor. Ultimately, distinguishing between the role of different factors requires

phylogenetically based analyses examining the causal direction and order of evolutionary changes[22].

We resolve this problem by performing a series of phylogenetic analyses on the number of physical worker castes in ants. We focus on physical castes which are irreversibly fixed for life and determined during development, rather than behavioural differentiation among workers. This is because extreme specialization, involving mutual dependence, requires differences in size and morphology. Ants provide an excellent opportunity for testing the size–complexity hypothesis because they are a single monophyletic clade in which the number of worker castes varies across species from one to four, and there is considerable information on factors that could influence the evolution of castes[23–27]. We analysed data from 794 species to examine the influence of three possible factors: colony size, queen mating frequency and number of queens per colony. We performed phylogenetic regressions, to determine how these three traits are correlated with the number of castes and each other. This step provided an overview that helped to guide further causal analyses and allowed comparison with previous studies. We then examined the likely causal relationships between number of worker castes, queen mating frequency and colony size with three methods: (1) phylogenetic path analysis[28], (2) transition rate analysis between pairs of traits[29] and (3) ancestral state reconstruction[30]. These analyses allowed us to tease apart the different roles of each variable. Last, to assess the robustness of our results, we also tested whether the same patterns emerged when examining an alternative possible measure of division of labour—variation in worker size[12,13,18,31].

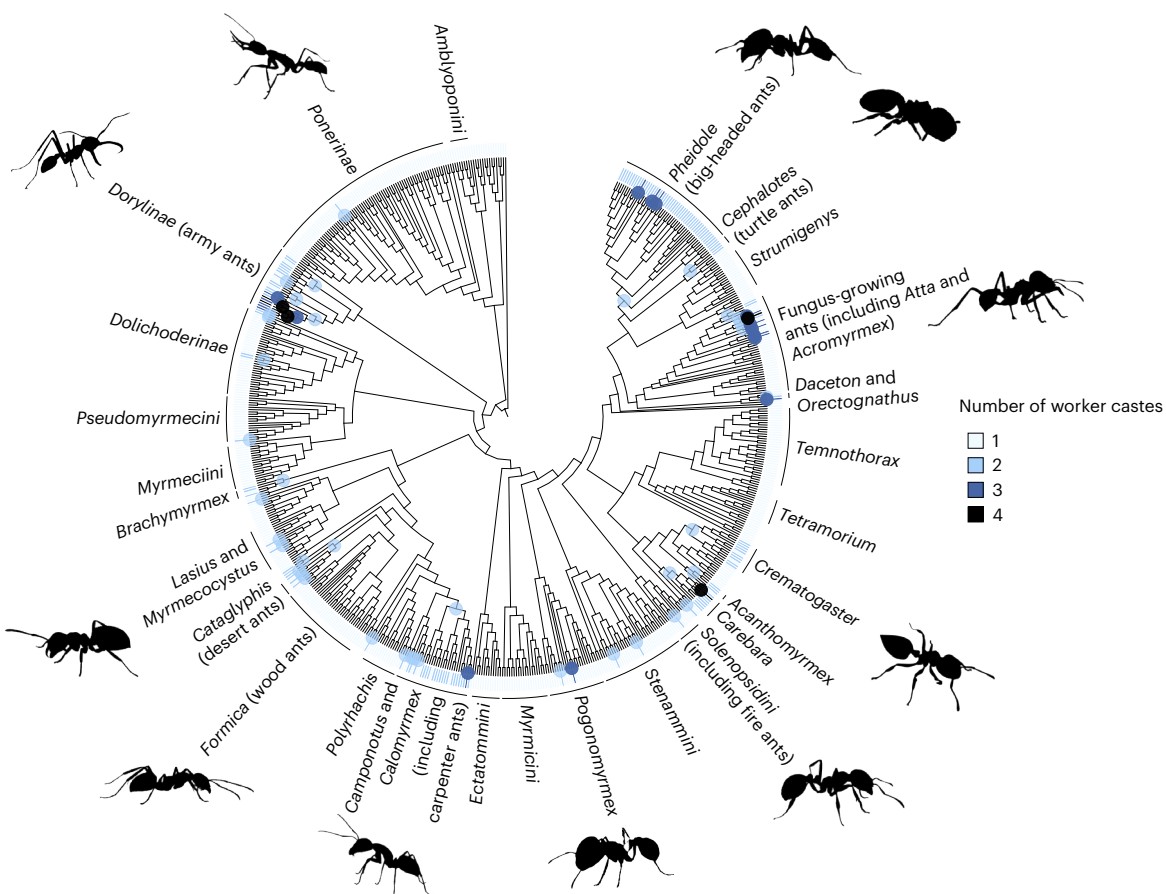

**Fig. 2 | Evolutionary origins of multiple worker castes (worker polymorphism).** The ancestral state at the root of the ant phylogeny is a single physical worker caste. Coloured circles represent evolutionary increases in number of worker castes, with colour corresponding to the number of castes that have evolved. The number of worker castes present in each species is also displayed around the outside of the phylogeny. Branch lengths are not to scale. Ant phylogenetic lineages, of varying levels, are displayed in black text around the outside of the phylogeny. Analyses were repeated over 400 phylogenetic trees, but we display the results of a single representative tree.

## Results

### Summary of ant social traits

Across different ant species, there was considerable variation in the characteristics of a colony. The number of discrete physical worker castes varied from one to four, although 76% of species had only one worker caste (Fig. 1a). Ancestral state reconstruction suggested that polymorphic workers, with more than one caste, had evolved from monomorphic ancestors at least 27 times (Fig. 2 and Supplementary Table 1). Colony sizes varied from 7 to 14,750,000, with a median of 300 (Fig. 1b). The effective number of males that inseminated each queen varied from 1 to 26, with 31% of species mating with a single male, 36% mating with between 1 and 2 males and 33% mating with more than 2 males. The median value for queen mating frequency was 1.1 mates (Fig. 1c). The number of queens per colony varied from 1 to 70, with 63% of species possessing a single queen, 18% possessing between 1 and 2 queens and 19% possessing more than 2 queens. The median number of queens per colony was 1 (Fig. 1d). The dataset was not complete for all traits for each species, and so for each analysis we used the maximum amount of data available. Therefore, sample sizes vary between analyses depending on the variables being analysed, as detailed within each section.

### Phylogenetic correlations

We used Bayesian phylogenetic mixed effect models (BPMMs) to test for correlations between the different ant traits. We found that colony size and queen mating frequency were both positively correlated with number of worker castes (Fig. 3a,b; BPMMs: colony size: $\beta = 0.17$, credible interval (CI) = 0.11 to 0.24, $n_{species} = 436$; queen mating frequency: $\beta = 0.47$, CI = 0.04 to 0.91, $n_{species} = 104$). By contrast, we found no relationship between number of queens per colony and number of worker castes (Fig. 3c; BPMM: $\beta = -0.14$, CI = −0.56 to 0.30, $n_{species} = 192$).

We also found that species with higher queen mating frequencies had larger colony sizes (Fig. 3e; BPMM: $\beta = 1.25$, CI = 0.50 to 2.03, $n_{species} = 109$, $R^2 = 0.08$). The significant correlations between queen mating frequency, colony size and number of worker castes emphasizes the need to carry out analyses that can examine the underlying causality between these variables. By contrast, we found no relationship between number of queens per colony and colony size (Fig. 3f; BPMM: $\beta = 0.07$, CI = −0.37 to 0.49, $n_{species} = 195$, $R^2 = 0.00$). However, we did find a negative relationship, which was close to significance, between number of queens per colony and queen mating frequency (Fig. 3d; BPMM: $\beta = -0.20$, CI = −0.39 to 0.01, $n_{species} = 111$, $R^2 = 0.00$). Based on the results of our correlational analysis, we excluded the number of queens per colony from our causality analysis, except for a potential relationship between number of queens per colony and queen mating frequency, where a negative relationship was also observed previously[25].

### Causality analyses

We then used three different methods to examine the causal relationships that underlly these correlations: (1) phylogenetic path analysis, (2) transition rate analysis and (3) ancestral state reconstruction.

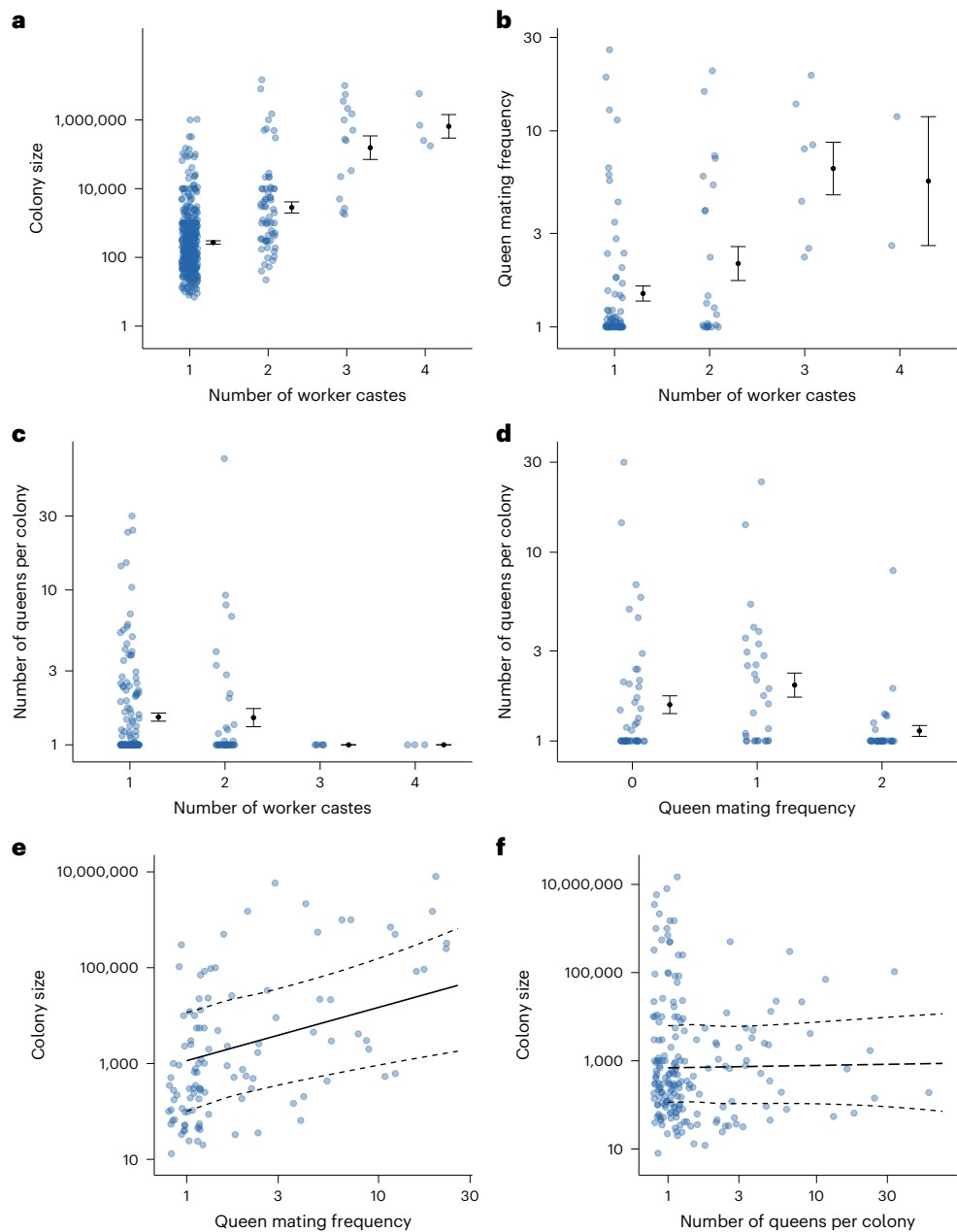

**Fig. 3 | Relationships between colony size, queen mating frequency and number of queens per colony, both among themselves and with number of worker castes. a,b,** Species with larger colony sizes ($n_{species}$ = 436) (**a**) and higher queen mating frequencies ($n_{species}$ = 104) (**b**) had significantly more worker castes. **c,d,f,** We did not find a significant association between number of queens per colony and either number of worker castes ($n_{species}$ = 192) (**c**), queen mating frequency ($n_{species}$ = 111) (**d**) or colony size ($n_{species}$ = 195) (**f**). **e,** Species with higher queen mating frequencies had significantly larger colony sizes ($n_{species}$ = 109). **a–d,** Data show mean ± standard error. **e,f,** Fitted lines are mean regression slopes with 95% CIs from BPMMs using a single phylogenetic tree. Solid regression lines represent significant relationships, while dashed regression lines represent non-significant relationships. Dots represent species averages. Axes for colony size, queen mating frequency and number of queens per colony are plotted on a $log_{10}$ scale.

These forms of analysis required number of worker castes to be modelled as a binary variable, and so we classified species as possessing either a single (monomorphic) or multiple (>1, polymorphic) worker castes.

**Phylogenetic path analysis.** We used phylogenetic path analysis to test four alternative causal models of the relationships between the following: number of worker castes, colony size, queen mating frequency and number of queens per colony (Extended Data Fig. 1). We found strong evidence in favour of the size–complexity hypothesis. The analysis supported a single causal model, where large colony sizes favoured

the evolution of multiple worker castes, and size was the only variable that directly did so (Fig. 4, Extended Data Fig. 2 and Supplementary Tables 2 and 3; $\omega$ (relative weight of support for a model within a set of competing models) = 0.67, $n_{species}$ = 94). We found no support for alternative causal models where colony size is influenced by number of worker castes or where number of worker castes is influenced by queen mating frequency (Supplementary Table 2; $n_{species}$ = 94).

**Transition rate analysis.** We used transition rate analysis as a second method to test the hypothesized causal relationships between number of worker castes and both colony size and queen mating frequency.

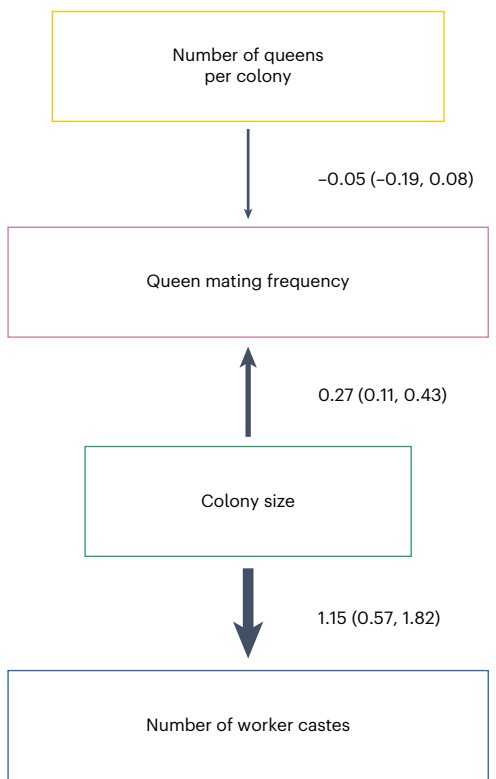

**Fig. 4 | The best-supported causal model identified by phylogenetic path analysis.** Larger colony sizes facilitated the evolution of multiple worker castes. We display the results from the analysis of one MCC consensus tree (Supplementary Tables 2 and 3). However, we repeated the analyses on three additional MCC trees and, in each case, found that the conclusions drawn remained unchanged due to the similarity in the presence, direction and magnitude of the causal relationships identified (Extended Data Fig. 2 and Supplementary Tables 2 and 3). Values represent standardized path coefficients with 95% confidence intervals (path coefficients describe the strength and direction of the relationship between two variables in terms of their correlated variance, after accounting for the effects of other variables in the model). Arrows indicate the direction of the relationship between variables, with heavier lines indicating larger coefficients. Number of worker castes was modelled as a discrete binary variable, while colony size, queen mating frequency and number of queens per colony were modelled as continuous variables.

This method allowed us to test whether the evolution of two traits is correlated, as well as the underlying causal direction, but required all traits to be modelled as discrete binary variables. We classified species as having small or large colony sizes depending upon whether colony size was smaller ($\leq$300) or larger (>300) than the median colony size. We classified species as having low or high mating frequencies depending on whether queens mated more (>2) or less than or equal to ($\leq$2) two times.

We found very strong support for the size–complexity hypothesis, with correlated evolution between colony size and number of worker castes (Supplementary Table 4; Bayes factor (BF) = 20.83, $n_{\text{species}}$ = 436; BFs quantify the relative support for two competing models where BF > 2 offers positive evidence, > 5 provides strong evidence and >10 very strong evidence that the more complex model performs better than the simpler model[29]). Transitions to multiple castes only occurred in species with large colony sizes (Fig. 5). In species with only a single worker caste, transition rates from small colony sizes to large colony sizes were slightly elevated in comparison to species with multiple castes (Fig. 5). By contrast, we found that variation in queen mating frequency does not influence the evolution of number of worker castes, as queen mating frequency evolved independently of the number of

worker castes (Supplementary Table 5; queen mating frequency and worker castes: BF = −1.39, $n_{\text{species}}$ = 104).

**Ancestral state reconstruction.** We used ancestral state reconstruction as a third method to examine the underlying causal relationships and again found support for the size–complexity hypothesis. We found that the ancestors of species that evolved multiple worker castes had colony sizes that were more than 3.5 times larger compared with the ancestors of species that retained one worker caste (Extended Data Fig. 3a; BPMM: ancestral colony size in species with one worker caste, $\beta$ = 281.67, CI = 40.04 to 1,590.90; ancestral colony size in species with multiple worker castes, $\beta$ = 1,025.05, CI = 137.17 to 9,064.82; $P_{\text{single vs multiple}}$ < 0.01, $n_{\text{species}}$ = 436). By contrast, queen mating frequency did not differ significantly between ancestors that evolved multiple worker castes and those that did not (Extended Data Fig. 3b; BPMM: ancestral queen mating frequency in species with one worker caste, $\beta$ = 1.99, CI = 0.95 to 3.90; ancestral queen mating frequency in species with multiple worker castes, $\beta$ = 2.46, CI = 1.10 to 5.55; $P_{\text{single vs multiple}}$ = 0.16, $n_{\text{species}}$ = 104). The phylogenetic signal present in both colony size and queen mating frequency across ants allowed us to estimate ancestral values relatively accurately in these traits (Supplementary Table 6; BPMMs: phylogenetic heritability of colony size = 72.61%, CI = 59.67% to 84.02%; phylogenetic heritability of queen mating frequency = 88.94%, CI = 74.01% to 99.86%).

## Division of labour and worker size

Taken together, our analyses on the number of worker castes provide strong support for the size–complexity hypothesis. Larger colony sizes correlate with and appear to lead to the evolution of more worker castes (Figs. 3–5). However, most ant species do not possess more than one physical worker caste. Despite this, there can be considerable variation in worker size within castes, and there is some evidence that different-sized workers perform different roles even within ant species that have a single physical worker caste[32]. Consequently, an alternative method for testing the size–complexity hypothesis would be to examine the relationship between colony size and variation in worker size, rather than number of castes, as a measure of non-reproductive division of labour[12,13,18,31].

Using data obtained from AntWeb, we examined variation in the head width measurements of 1,064 workers from 152 species and found mixed support for the size–complexity hypothesis (Extended Data Fig. 4). We found a positive correlation between colony size and variation in worker size and that larger colony sizes may favour the evolution of greater variation in worker size (Fig. 6, Extended Data Fig. 5 and Supplementary Tables 7 and 8; BPMM: colony size, $\beta$ = 0.05, CI = 0.03 to 0.08, $n_{\text{species}}$ = 122, $R^2$ = 0.14; path analysis: $n_{\text{species}}$ = 94). However, there was some uncertainty regarding the causal direction of the relationship between colony size and variation in worker size. Among the three supported causal models identified by path analysis, larger colony sizes favoured the evolution of greater variation in worker size in two cases, and greater variation in worker size favoured the evolution of larger colony sizes in the third case (the model fit was compared using the corrected C-statistic Information Criterion (CICc), where models within 2 CICc units of the best model were considered to have equal support; Supplementary Table 7; combined weight for the models supporting the hypothesis that variation in worker size is influenced by colony size: $\omega$ = 0.55). Due to the uncertainty regarding the best-supported model, the confidence intervals for the path coefficients in the average model, linking both colony size and queen mating frequency with variation in worker size, were relatively large and overlapped with zero (Fig. 6b). When examining species with only a single worker caste, we found the contrasting result that greater variation in worker size favours the evolution of larger colony sizes (Fig. 6a, Extended Data Fig. 6 and Supplementary Tables 9 and 10; BPMM: $\beta$ = 0.04, CI = 0.01 to 0.07, $n_{\text{species}}$ = 84, $R^2$ = 0.08; path analysis: $\omega$ = 0.75, $n_{\text{species}}$ = 60).

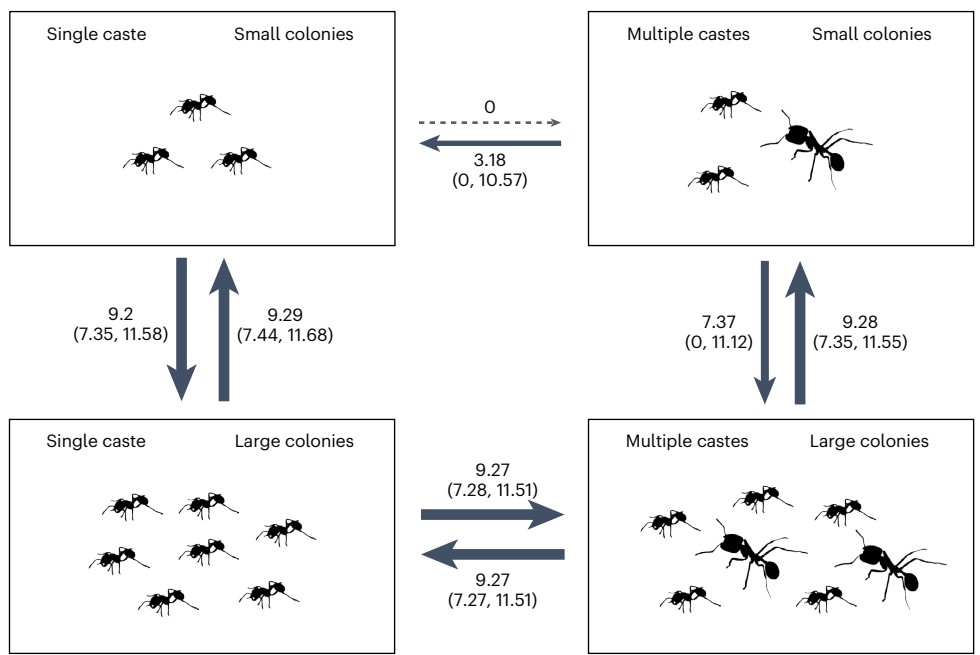

**Fig. 5 | Evolutionary transitions between number of worker castes and colony size.** Multiple worker castes only evolved in lineages with large colony sizes. Values represent the inferred transition rates between number of worker castes and colony size, accompanied by 95% CIs ($n_{species}$ = 436). Transition rates were estimated using rjMCMC implemented in BayesTraits. Each possible state is shown as a box, and the direction of transitions between states are shown with arrows, with higher rates of transition indicated by a heavier line.

By contrast, we did not find support for the hypothesis that variation in worker size is influenced by queen mating frequency. This was the case when analysing either all species or just those with a single worker caste (Fig. 6b, Extended Data Fig. 5–7 and Supplementary Tables 7–11).

## Sensitivity analyses

We examined whether our results were robust to alternative methods of quantifying variation in queen mating frequency and number of queens per colony as categorical rather than continuous variables[21,25,33]. Detailed statistics for each analysis are presented in the supplementary material (Extended Data Figs. 8–10 and Supplementary Tables 11–19). In all cases, the size–complexity hypothesis was still supported, but some of the other relationships depended upon how data were categorized. Our conclusions regarding the phylogenetic correlations were robust, except for the finding that the number of worker castes did not significantly differ between singly mated and facultatively multiply mated species (Supplementary Table 19). The conclusions of our causal analyses were also robust, except when analysing the number of discrete worker castes using phylogenetic path analysis. We identified a single supported model where the evolution of multiple worker castes was favoured by both larger colony sizes and obligate multiple mating by queens. However, the confidence interval for the path coefficient associated with queen mating frequency overlapped with zero (Extended Data Fig. 8 and Supplementary Tables 12 and 13).

## Discussion

We found strong and consistent support for the size–complexity hypothesis, with larger colony sizes appearing to favour the evolution of multiple worker castes (Figs. 3–6). By contrast, we did not find consistent support for the hypothesis that multiple mating favours the evolution of multiple worker castes (Figs. 3, 4 and 6b). We found no evidence to suggest a direct relationship between the evolution of number of worker castes and number of queens per colony (Fig. 3c). Our conclusions were robust to different analysis methods.

Our results suggest that larger colony sizes favour the evolution of multiple worker castes in ants. Previous studies had highlighted a positive correlation between increased division of labour and both cell number in multicellular organisms and colony size in social insects, but these results were open to multiple explanations[7–9,12–14,18,21]. Our results reveal a distinct evolutionary pattern where larger colony sizes tend to evolve before multiple worker castes (Figs. 4 and 5). There are at least two reasons why larger colonies may promote the evolution of multiple castes. First, as colony size increases, the number of tasks that need to be performed also increases[2]. For example, species with larger colonies may experience greater logistical challenges in transporting food or waste within or outside the colony. Second, species with larger colonies are better able to maintain the optimal ratio of the different castes[10,34]. Consequently, these species face a lower risk of losing essential worker functions if workers are lost.

Our analysis also suggests that larger colony sizes favoured the evolution of greater variation in worker size (Fig. 6). However, there is some uncertainty regarding the direction of this relationship, as causality could potentially be reversed. In addition, we found that this result only applied when analysing across species with variable numbers of physical worker castes. When analysing variation in worker size exclusively in species with a single caste, we found that greater variation in worker size preceded the evolution of larger colony sizes. Consequently, while our analyses of variation in worker size support the size–complexity hypothesis, they suggest that colony size is not responsible for the evolution of variation in worker size in species that lack discrete physical worker castes.

We did not find consistent support for the hypothesis that multiple mating favours the evolution of multiple worker castes. There is a positive correlation between both number of worker castes and variation in worker size with queen mating frequency, regardless of whether it is analysed as a continuous or categorical trait (Fig. 3b). However, when analysing the causal relationship between these variables, we found

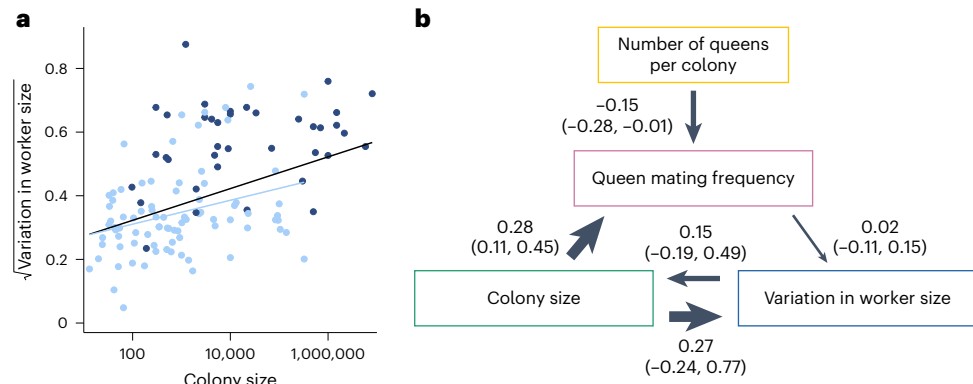

**Fig. 6 | Larger colony sizes may favour the evolution of greater variation in worker size. a**, Species with larger colony sizes possess significantly greater variation in worker size when analysing either of the following: (1) all species for which data were available (all dots and regression line in black, $n_{species}$ = 122) or (2) only species which possess a single worker caste (dots and regression line in light blue, $n_{species}$ = 84). Fitted lines represent mean regression slopes from BPMMs. Dots represent species averages. Colony size is plotted on a $\log_{10}$ scale, while variation in worker size is on a square root scale. **b**, Averaged model from the three supported path analysis models (Supplementary Tables 7 and 8). Larger colony sizes may favour the evolution of greater variation in worker size, but opposite causation remains possible. Analysis performed using all species where data were available for variation in worker size, colony size, number of queens per colony and queen mating frequency ($n_{species}$ = 94). We display the results from the analysis of one MCC consensus tree. We repeated the analyses on three additional MCC trees and, in each case, found that the conclusions drawn remained unchanged due to the similarity in the presence, direction and magnitude of the causal relationships identified (Extended Data Fig. 5 and Supplementary Tables 7 and 8). Values represent standardized path coefficients with 95% confidence intervals. Arrows indicate the direction of the relationship between variables, with heavier lines indicating larger coefficients. Variation in worker size, colony size, queen mating frequency and number of queens per colony were modelled as continuous variables.

that higher mating frequency was not consistently associated with evolutionary transitions from single to multiple worker castes (Figs. 4 and 6b, Extended Data Figs. 8 and 9 and Supplementary Tables 5 and 18). This finding is consistent with the interpretation that queen mating frequency is positively correlated with number of worker castes due to both variables being positively associated with colony size (Figs. 3 and 4). Colony size and queen mating frequency may be positively correlated due to the advantages of increased genetic diversity that result from multiple mating[21,35–41]. If species with larger colony sizes are at greater risk of infection from pathogens, for example, resulting from increased traffic of foragers into the nest, they may benefit more from the protection offered against disease by increased genetic diversity[40–42].

Excluding ants, possessing more than one physical worker caste is extremely rare in the social Hymenoptera. Multiple worker castes appear to be entirely absent in vespine wasps, while in bees, the presence of two castes has only been identified in some stingless bee species[31,43]. For example, *Tetragonisca angustula* possesses two castes—soldiers are both approximately 30% heavier than foragers and have a different morphology[43]. This contrasts with the much larger variation in ants—for example, major workers in *Atta colombica* leafcutter ants can be 60 times heavier than the smallest workers[44]. Why does the causal link between colony size and number of castes that we have identified in ants seemingly not apply in superorganismal species of bees and wasps[31]? Multiple hypotheses have been proposed, suggesting that factors such as the presence of a powerful sting or body size constraints imposed by having winged workers could have impeded the evolution of multiple worker castes in bees and wasps[6,24,45]. However, the phylogenetic power available to carry out statistical tests of these hypotheses is low. Other ways to test the size–complexity hypothesis include examining behavioural variation in bees and wasps or the physical caste variation in termites[6,31,46–48].

To conclude, our results provide strong support for the size–complexity hypothesis in ants, suggesting that larger colony sizes have favoured the evolution of greater division of labour (more castes and greater variation in size). The generality of this hypothesis could be tested by carrying out similar analyses on other forms of division of labour, such as the number of cell types in multicellular organisms[9,49,50] or the difference between queens and workers in superorganismal social insects[48,51].

## Methods

To test whether colony size, queen mating frequency or number of queens per colony explains variation in non-reproductive division of labour in ants, we performed a large-scale phylogenetic comparative analysis. We first tested which factors correlate with number of worker castes and then used three different methods to examine the causal relationships that underly these correlations: phylogenetic path analysis, transition rate analysis and ancestral state reconstruction. We examined causality in these three different ways as it also allowed us to test that our results are robust to different methods of analysis. We also examined whether our results were robust to alternative methods of quantifying variation in queen mating frequency and number of queens per colony. All the species used in the analysis are listed in the supplementary materials (Supplementary Table 20), and we follow the taxonomic classification of ants listed in Bolton's Catalogue available from AntCat.org[52]. When analysed as continuous variables, colony size, queen mating frequency and number of queens per colony were $\log_{10}$ transformed for all analyses, while variation in worker size was square root transformed.

### Data collection

We collected data on five ant traits: (1) the number of discrete physical worker castes, (2) colony size (number of workers in mature colonies), (3) effective queen mating frequency[21,53] (estimated number of mates weighted by the proportion of offspring sired by each male; referred to as 'queen mating frequency'), (4) observed number of queens per colony and (5) variation in worker size. We started by gathering data from major reviews, comparative studies and books and then performed a topic search of the primary literature using Web of Science[24–27]. We also incorporated data collected by the Global Ant Genomics Alliance consortium[54]. In total, we collected data for 794 species from 160 different genera.

We collected data on queen mating frequency and number of queens per colony by searching papers published between 1 January 2008

and 2 May 2020. Among these papers, we included data from Hughes et al. [25], which summarizes the available data on these variables in the eusocial Hymenoptera. We assumed that all relevant data published before 1 January 2008 had been captured by Hughes et al. [25]. We used the following keywords: 'Ant' AND (monandr* OR monogyn* OR polyandr* OR polygyn* OR effective-mating-frequenc* OR mating-frequenc* OR paternity-frequenc* OR mating-system* OR sociogenetic-structure*). We chose to use estimates of effective mating frequency, rather than number of copulations, because it more accurately reflects the genetic relatedness among workers in a colony, considering the potential influence of biased sperm use by queens that have mated multiple times[21,55]. When multiple estimates of queen mating frequency were available, we calculated species-level averages for queen mating frequency using the harmonic mean. We used the harmonic mean because when mating frequency is low, increases in mating frequency will have greater effects on intra-colony relatedness. Therefore, harmonic means, which assign greater weight to small numbers, better represent the genetic effects of multiple mating on colony-wide relatedness than arithmetic means[56]. We collected data on the average observed number of queens per colony, rather than the effective number of queens per colony[57] (the estimated number of reproductive queens weighted by the respective contribution of each queen to the production of offspring) because this was the only available type of continuous data with a sufficient sample size for a large-scale phylogenetic comparative analysis. When multiple estimates of number of queens per colony were available, we calculated species-level averages. For queen mating frequency and number of queens per colony, we also collected categorical data with three factor levels for each trait from the papers identified in our literature search. These factor levels include obligate single mating (monandry), facultative multiple mating (facultative polyandry) where some monandrous queens still occur in the population and obligate multiple mating (obligate polyandry) where mature colonies are never monandrous. Similarly, for number of queens per colony, these levels were obligate single queen (monogyny), facultative multiple queens (facultative polygyny) and obligate multiple queens (obligate polygyny).

Species were classed as having one to four physical worker castes based on the following criteria: Species were classed as having a single worker caste if they showed limited variation in size or monophasic allometric scaling. Species were classified as having more than one caste if there was variation in body size with non-allometric scaling, where the number of castes depended on the number of different scaling relationships between body parts. When morphometric data were absent, the number of castes was determined using direct estimates provided in the literature. We searched for published data up to and including 13 July 2020, and we used the following keywords: 'Ant' AND (worker polymorph* OR worker monomorph* OR (morphometric AND worker* AND caste*) OR subcaste* OR sub-caste* OR worker dimorph* OR major-worker* OR minor-worker* OR worker AND allometr*). We then performed two additional searches to collect missing data for number of worker castes. First, we performed a species-specific literature search for instances where data on queen mating frequency or number of queens per colony was available for a given species but number of castes was not. Our search query included the species name along with the terms '(worker polymorph* OR worker monomorph* OR (morphometric AND worker* AND caste*) OR subcaste* OR sub-caste* OR worker dimorph* OR major-worker* OR minor-worker* OR worker AND allometr* OR soldier* OR replete*)'. Second, we performed a search at the genus level for genera where no caste data had been collected. The search query used the same terms as mentioned above, in combination with the genus name.

To collect data on colony size, we performed a literature search focussing on species for which we already possessed data on queen mating frequency or number of queens per colony. Using Google Scholar, we searched for the following keywords in combination with the species name 'colony size OR colony collection OR worker number'. We adopted this approach based on the methodology outlined in

Dornhaus et al. [14]. Google Scholar was chosen as it indexes the entire text of research papers. Notably, details regarding colony size are often not the primary focus of papers, resulting in brief mentions within the methods or results sections. Therefore, it is less likely to be recovered by search engines that rely solely on keywords, titles and abstracts to retrieve such information. When multiple measurements existed for a given species, we calculated arithmetic means weighted by sample size.

We collected data on variation in worker size by measuring the width of ant heads[12,13,18,31]. We quantified the relative variation in worker size for each species by calculating the coefficient of variation (CV), which is achieved by dividing the standard deviation of worker head width by its mean value ($CV_{worker size} = \frac{\text{worker head width standard deviation}}{\text{worker head width mean}}$). To obtain images of the species for which we already possessed data on colony size and queen mating frequency, we searched for scaled images on the AntWeb online database (antweb.org)[58]. We downloaded front-view images of the ants' heads and measured at the widest point excluding the eyes using the image-processing software ImageJ v1.53i[59]. In total, we measured 1,064 worker ant heads (Extended Data Fig. 4). The number of workers measured per species ranged from 2 to 33, with a mean of 7.

We excluded 112 species from our dataset for the analysis based on their distinct life history traits. These traits represent evolutionarily derived elaborations of the ancestral full-sibling-colony state at the transition to superorganismal colonies at the root of the ant clade. They included (1) species that formed supercolonies (vast networks of connected nests); (2) social parasites, lacking some or all of the worker castes (although temporary social parasites which only establish new colonies with the assistance of a host species were included in the analysis); (3) those that can reproduce parthenogenetically to produce queens and/or workers; (4) those that reproduce via gamergates (mated workers that reproduce sexually) following the evolutionary loss of the original queen caste; or (5) those that use interlineage hybridization for genetic caste determination (Supplementary Table 21). We gathered the information from papers that had summarized the available data on these five traits[60–64]. We excluded these species from the analysis as they represent secondary reductions of complexity in social organization and are likely experiencing different selection pressures for either the evolution of worker castes, colony size, queen mating frequency or number of queens per colony (Supplementary Table 22). This decision to remove species characterized by evolutionarily derived social systems improved our ability to detect which variables could influence the evolution of the number of worker castes.

## Phylogeny

To control for shared evolutionary history in our comparative analyses, we used a posterior sample of 400 phylogenetic trees produced by Economo et al. [65], which included 14,594 species and of which 731 overlapped with the species in our dataset[65]. However, as time-calibrated, molecular phylogenies are only available for ants at the genus level, the species topology within genera varied widely across the sample of 400 trees. Therefore, we took steps to account for phylogenetic uncertainty in each of our analyses (see each section below for details).

## Statistical analyses

All analyses were performed in R v4.2.2 apart from transition rate models that were conducted in Bayestraits V4[29,66].

**Bayesian phylogenetic models.** We fitted BPMMs with Markov chain Monte Carlo (MCMC) estimation using the MCMCglmm package v2.34[67]. Models were run for a minimum of 1,100,000 iterations, with a burn-in of 100,000 and thinning interval of 1,000. However, models were run for as long as necessary to obtain posterior effective sample sizes of at least 300 for all parameters. Overall, the majority of estimates had an effective sample size of at least 1,000. To ensure model convergence,

we used the coda package v0.19-4 to calculate the degree of autocorrelation between successive iterations in each chain. We fitted each model independently two times and used Gelman and Rubin's convergence test to compare within- and between-chain variance[68,69]. We modelled the number of worker castes as a discrete variable for our regression analyses and used a Poisson error distribution with a log-link function. We modelled colony size, queen mating frequency and number of queens per colony as Gaussian traits. The prior settings used for each analysis are specified in the supplementary R code. To select priors for random effects, we first checked model convergence using inverse-Wishart priors ($V = 1$, $v = 0.002$). However, in situations where the MCMC chain showed poor mixing properties, particularly in cases involving discrete response variables, we examined two different parameter expanded priors: the Fisher prior ($V = 1$, $v = 1$, $\alpha.\mu = 0$, $\alpha.V = 1{,}000$) and the $\chi^2$ prior ($V = 1$, $v = 1{,}000$, $\alpha.\mu = 0$, $\alpha.V = 1$). We specified an inverse-Wishart prior for residual variances ($V = 1$, $v = 0.002$)[70]. For fixed effects, we used the default priors in MCMCglmm. We report parameter estimates from models as posterior modes along with 95% lower and upper CIs. For BPMMs with a Gaussian error distribution, we also calculated marginal $R^2$ values and report the median value for each analysis from a distribution of $R^2$ values estimated across a sample of 400 trees[71]. The reported $P$ values, which assess differences between levels, such as species with a single worker caste versus species with multiple worker castes, represent the number of iterations where one level is greater than the other level divided by the total number of iterations.

### Correlational analyses
**Estimating phylogenetic correlations using MCMCglmm.** We used BPMMs to test for correlations between four traits: number of worker castes, colony size, queen mating frequency and number of queens per colony. In addition, we investigated the correlations between variation in worker size and colony size, queen mating frequency and number of queens per colony. When analysing variation in worker size, our analysis was split into two parts. The first part examined correlations across all species for which worker size data were available, while the second part specifically examined species that have a single worker caste. To account for phylogenetic uncertainty in our BPMMs, we repeated each analysis 400 times, each time with a different tree, and combined the posterior samples produced from each tree before parameter estimation. Model convergence was assessed as described in the section 'Statistical analyses'.

### Causality analyses: examining colony size and levels of polyandry that preceded the evolution of multiple worker castes
**Phylogenetic path analysis analysing number of discrete physical worker castes.** To test how the evolution of number of worker castes was influenced by colony size and queen mating frequency, we used phylogenetic path analysis[28,72]. This method compares alternative models of the causal relationships between traits, disentangling direct from indirect effects. After removing species with incomplete data for all variables, we analysed the data for a total of 94 species. By comparing four different causal models, we tested between the following possibilities: (1) variation in colony size directly influenced the evolution of number of worker castes, but queen mating frequency did not; (2) variation in number of worker castes influenced the evolution of colony size, which then predicted queen mating frequency; (3) variation in both colony size and queen mating frequency directly influenced the evolution of number of worker castes; and (4) variation in queen mating frequency directly influenced the evolution of number of worker castes, but colony size did not (list of full models in Extended Data Fig. 1). In each of the causal models tested, we assumed that the evolution of queen mating frequency was influenced by variation in both the number of queens per colony and colony size. We made these assumptions because (1) it has been suggested that the evolution of queen

mating frequency has been influenced by variation in the number of queens per colony[21,25] and (2) larger colony sizes may select for multiple mating[40,41]. Larger colonies may be at greater risk of infection from pathogens, and multiple mating may offer protection against disease because it increases genetic diversity within colonies[37,40–42]. However, multiple mating is a costly trait, and therefore, we hypothesized that larger colony sizes may evolve first and then select for multiple mating if the benefits of multiple mating outweigh the costs[73]. We excluded the relationship between the number of queens per colony and either the number of castes or colony size in our models as we found no correlation between these variables in our regression analysis (section 'Phylogenetic correlations').

Based on the causal relationships presented in the alternative path analysis models, we generated conditional independence statements that can be formulated and tested as a set of phylogenetic generalized linear models. For each model, we calculated Fisher's C statistic and conducted the d-sep test[28]. $P$ values less than 0.05 indicate that a proposed candidate model should be rejected. Next, we compared the fit of different models using CICc scores, where the lowest score represents the best candidate model[74]. Models that have $\Delta$CICc < 2 were considered to have equal support. We used a model-averaging approach to estimate path coefficients that assigned weights to causal links based on the CICc weight ($\omega$) of the supported models ($\Delta$CICc < 2). When a path was absent in a model, we assumed its coefficients and variance to be zero for the purpose of model averaging. We transformed the number of castes into a binary variable (single worker caste/multiple worker castes) for the analysis as the phylopath package does not support discrete count data as response variables. We then used a binomial family for binary response variables when modelling number of worker castes. To the best of our knowledge, there is currently no established method for combining results obtained across different trees to account for phylogenetic uncertainty in frequentist analyses. Therefore, we performed our analysis four times, each time using a different tree, from a sample of four maximum clade credibility (MCC) trees produced by Economo et al. [65]. Each MCC tree was constructed from a sample of 100 trees. We then compared the results from each of the four analyses to determine whether our results remained consistent irrespective of the phylogeny that was used. In the main text, we present the results from a single tree as we found that the conclusions drawn across trees remained unchanged due to the similarity in the presence, direction and magnitude of the causal relationships identified. We present the results using the alternative MCC trees in the supplementary material (Extended Data Fig. 2 and Supplementary Tables 2 and 3). All phylogenetic path analyses were carried out using phylopath v1.1.3 in R[75].

**Phylogenetic path analysis analysing variation in worker size.** We repeated our analysis, examining variation in worker size instead of number of worker castes. We used head width as a measure of worker size and modelled it as continuous variable. Models were constructed using Pagel's $\lambda$[76] for the associated error structure, and we compared the same four causal models that were used in the previous path analysis (Extended Data Fig. 1). To investigate whether differences in the number of worker castes were driving the results when analysing worker size, we performed the analysis in two different ways: (1) with the full set of species for which we had data available for all four traits ($n_{species} = 94$) and (2) limiting our analysis to only species that possess a single worker caste (monomorphic; $n_{species} = 60$). To account for phylogenetic uncertainty, we repeated each analysis using the same four MCC trees used in the previous path analysis. In the main text, we present the results from a single tree as we found that the conclusions drawn across trees remained unchanged due to the similarity in the presence, direction and magnitude of the causal relationships identified. We present the results for both analyses using the alternative MCC trees in the supplementary material (Extended Data Figs. 5 and 6 and Supplementary Tables 7–10).

**Testing models of dependent versus independent evolution and estimating evolutionary transition rates using BayesTraits.** We tested for correlated evolution between number of worker castes and both colony size and queen mating frequency. We used the Discrete module with reverse jump MCMC estimation implemented in BayesTraits V4. This method requires pairs of binary traits, and so we transformed all traits into binary variables. We classified species as having either a single worker caste or multiple worker castes, to focus on the evolution of multiple castes. We classified species as having queens that mated with less than or equal to two males (≤2 males) or more than two males (>2 males), because (1) the influence of larger numbers of matings on relatedness is diminishing; (2) an effective mating frequency value of 2 leads to the average level of relatedness among workers in the colony being midway between the maximal and minimal values of 0.75 and 0.25 that are possible with a single queen; and (3) mating with 2 males is the threshold at which worker policing is favoured[20]. We classified species as having either small or large colonies depending upon whether colony size was smaller (≤300) or larger (>300) than the median colony size. To assess the sensitivity of our results to different colony size classification thresholds, we repeated our analysis, but divided species into classes based on the 40th/60th and 60th/40th quantile boundaries. Across these thresholds, we obtained the same result that large colony size was necessary for the evolution of multiple worker castes. Therefore, we present the results of the 40th/60th and 60th/40th only in the supplementary material (Supplementary Tables 23 and 24). In all cases, we tested the fit of the independent model of evolution against the dependent model of evolution using BFs (2 × (log(likelihood of the dependent model) − log(likelihood of independent model))). We estimated the marginal likelihood of both the independent model and the dependent model using a stepping-stone sampler[77]. As recommended in the Bayestraits manual, we scaled tree branch lengths by a factor of 0.001. This helps prevent the estimated transition rates from becoming very small, which can make the values hard to estimate or search for. We accounted for phylogenetic uncertainty in our analysis by resampling each iteration from the posterior distribution of 400 trees to estimate parameters.

To reduce uncertainty over prior selection, we used a hyper prior approach to seed the mean and variance of an exponential prior[29]. The values for the hyper priors were drawn from a uniform distribution ranging from 0 to 100, based on the estimated range of transition rates determined by using analyses with maximum likelihood estimation. In addition, we investigated the sensitivity of our models to the selection of priors by running models with gamma priors initialized using hyper priors. We found that the conclusions drawn between models using different priors remained the same. Therefore, we only present the results from the models using the gamma priors in the supplementary materials (Supplementary Tables 25 and 26). All models were run for at least 11,000,000 iterations with 1,000,000 iterations of burn-in and a thinning interval of 5,000. However, some models were run for 110,000,000 with 10,000,000 iterations of burn-in and a thinning interval of 50,000 iterations to improve chain convergence and mixing. Model convergence was assessed using the same approach described in 'Statistical analyses'.

We identified correlated evolution between colony size and number of worker castes. Therefore, using the transition rates estimated by the model, we examined whether transitions to multiple worker castes were influenced by small or large colony sizes, as well as whether transitions to large colony sizes were influenced by number of worker castes. We found that queen mating frequency and number of worker castes evolved independently. Therefore, we did not explore the potential causal relationship between these traits. To determine the likelihood of transitions occurring, we examined the proportion of models visited by the reverse jump MCMC algorithm in which the rates were set to zero (Supplementary Tables 4, 5, 18 and 23–26).

**Ancestral state reconstructions.** We used ancestral state reconstruction to examine how colony size and queen mating frequency differed between the ancestors of lineages with a single caste and lineages with multiple castes[30]. First, we reconstructed the ancestral number of worker castes (single/multiple castes) to identify transitions between single and multiple worker castes. Second, we tested whether species that evolved multiple worker castes had greater estimated colony sizes and queen mating frequencies. To account for potential variation in the rate of caste evolution across the ant phylogeny, we used a hidden Markov model technique called hidden rate models (HRMs) to reconstruct the ancestral numbers of castes[78]. This method enabled us to incorporate heterogeneity in the loss and gain rates of our reconstruction. We used the R package corHMM v2.8 to examine HRMs with one to three rate classes and generated both equal rates and all-rates-different models of evolution for each rate class[78]. To determine the best HRM model from the candidate set, we examined corrected Akaike Information Criteria (AICc) values for each model across 400 trees using all available species with data on the number of castes. We selected the equal rates model with two rate categories because it had the lowest AICc value over 57.5% of the 400 trees. We then ran the equal rates model with two rate categories for each analysis, examining the relationship between colony size and number of worker castes and queen mating frequency and number of worker castes. Next, we classified each node in the phylogeny depending on which state (single/multiple castes) had the highest likelihood, resulting in four possible classification categories for each node: (1) single caste with all descendants having a single caste, (2) single caste with at least one descendant having multiple castes, (3) multiple castes with all descendants having multiple castes and (4) multiple castes with at least one descendant having a single caste.

The nodal classifications were entered as an explanatory variable in a BPMM and treated as a fixed factor with four levels. Depending on whether we were analysing the relationship between number of worker castes and either colony size or queen mating frequency, we specified either colony size or queen mating frequency as the response variable and used a phylogenetic covariance matrix that was linked to ancestral nodes as a random effect. We removed the global intercept and estimated the colony size and queen mating frequency values before the evolutionary origin of multiple worker castes (comparison of classification 1 versus 2). To account for phylogenetic uncertainty in our analysis, we repeated this process 400 times, once for each tree from our sample of 400 phylogenetic trees. We then combined the resulting posterior samples across the models to calculate parameter estimates. Accurate estimation of ancestral colony size and queen mating frequency values required phylogenetic signal in these traits. Using BPMMs, we estimated the amount of variation in each trait that can be attributed to shared ancestry between species, calculated as phylogenetic heritability (phylo $h^2 = \left( \frac{V_A}{V_A + V_R} \right) \times 100$, where $V_A$ = phylogenetic variance, $V_R$ = residual variance). For each trait, we repeated the analysis 400 times, each time with a different tree, and combined the posterior samples of the variance components before phylogenetic heritability was estimated.

In addition, we estimated the ancestral number of discrete worker castes (one to four worker castes) using HRMs. We ran each model over a sample of 400 trees and used the same model selection approach for HRMs as described above. The symmetric model with 2 rate classes performed the best over 58.5% of trees. We then estimated the frequency of transitions between each number of castes, classifying each node based on the state with the highest likelihood estimate.

**Sensitivity analysis**

To assess the robustness of our findings to different methods of quantifying queen mating frequency and number of queens per colony, we performed a sensitivity analysis. We repeated our phylogenetic correlations analysis, treating queen mating frequency and number

of queens per colony as categorical variables with three factor levels. This approach was chosen based on evidence indicating that these categorizations may reflect discrete reproductive strategies that may have evolved for different reasons and align with previous studies[21,25,33,79,80]. They were categorized as follows: obligate single mating (monandry), facultative multiple mating (facultative polyandry) and obligate multiple mating (obligate polyandry); and obligate single queen (monogyny), facultative multiple queens (facultative polygyny) and obligate multiple queens (obligate polygyny)[79,80].

We also repeated our phylogenetic path analysis and transition rate analysis. However, we transformed queen mating frequency and number of queens per colony into discrete binary variables, as these analytical methods cannot accept categorical variables with more than two factor levels. Species were classified as being obligately polyandrous or not, and obligately polygynous or not. We combined monandry with facultative polyandry, and monogyny with facultative polygyny to avoid excluding a large proportion of species from our analysis. We combined these categories because species that are obligately polyandrous or polygynous have made an evolutionary transition where mature colonies can no longer possess a monandrous or monogynous queen. By contrast, species that are facultatively polyandrous or polygynous may still show monandry or monogyny. Detailed statistics for each analysis are provided in the supplementary material (Extended Data Figs. 8–10 and Supplementary Tables 11–19).

### Reporting summary

Further information on research design is available in the Nature Portfolio Reporting Summary linked to this article.

## Data availability

All data are provided in Supplementary Table 20 and are made available, along with their associated references, via the public repository, Oxford University Research Archive (ORA) at https://ora.ox.ac.uk/objects/uuid:453278a3-61b9-4144-b08a-98d9379d5ce8 (ref. 81).

## Code availability

R code for analyses is available via GitHub at https://github.com/LouisBell-Roberts/Larger_colony_sizes_favoured_the_evolution_of_more_worker_castes_in_ants.

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

## Acknowledgements

We thank R. Bonifacii, J. Green, T. Scott, A. Dewar and M. Liu for their helpful comments, M. Liu for producing ant silhouettes, C. Cornwallis for advice on performing ancestral state reconstruction and M. Brindle for advice on performing transition rate analysis. We thank the Global Ant Genomics Alliance consortium for making their data on ant life history traits available to us, and the Biotechnology and Biological Sciences Research Council (BB/M011224/1: L.B.-R.) and European Research Council (834164: S.A.W. and J.F.R.T.) for funding.

## Author contributions

Conceptualization: S.A.W., L.R. and G.D.A.W. Data collection: L.B.-R., J.F.R.T. and L.R. Methodology: L.B.-R., J.F.R.T., S.A.W., G.D.A.W. and P.A.D. Investigation: L.B.-R., J.F.R.T., S.A.W. and G.D.A.W. Visualization: L.B.-R., J.F.R.T. and S.A.W. Funding acquisition: S.A.W. Supervision: S.A.W. and G.D.A.W. Writing—original draft: L.B.-R., J.F.R.T. and S.A.W. Writing—review and editing: L.B.-R., J.F.R.T., S.A.W., G.D.A.W., P.A.D. and L.R.

## Competing interests

The authors declare no competing interests.

## Additional information

**Extended data** is available for this paper at https://doi.org/10.1038/s41559-024-02512-7.

**Correspondence and requests for materials** should be addressed to Louis Bell-Roberts.

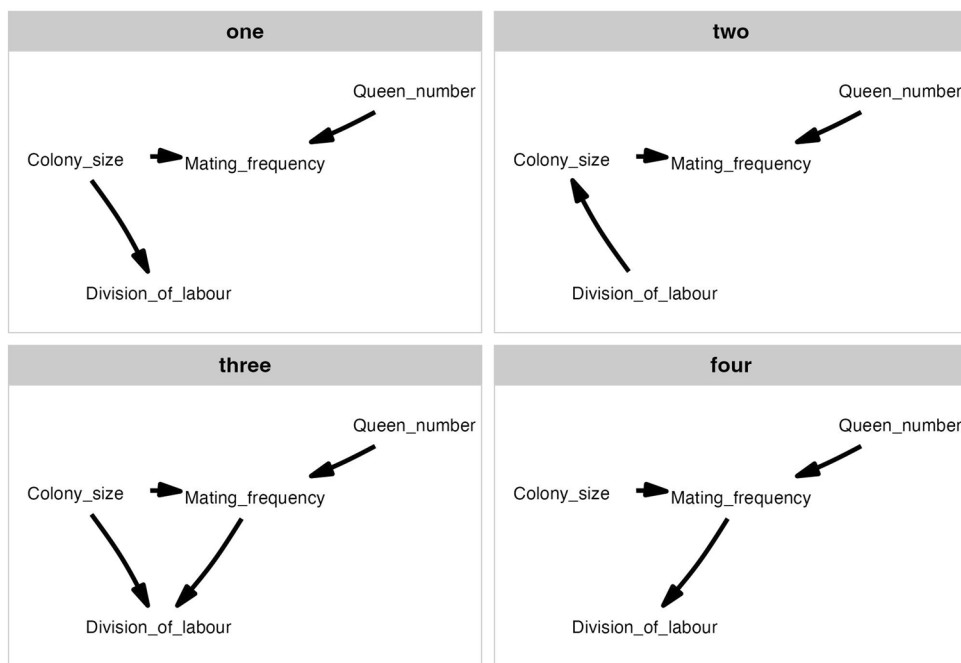

**Extended Data Fig. 1 | The full set of potential causal models tested in the phylogenetic path analyses illustrated using directed acyclic graphs (DAGs).** In different path analyses, division of labour was modelled either as the number of discrete physical worker castes (Fig. 4) or as variation in worker size (Fig. 6b).

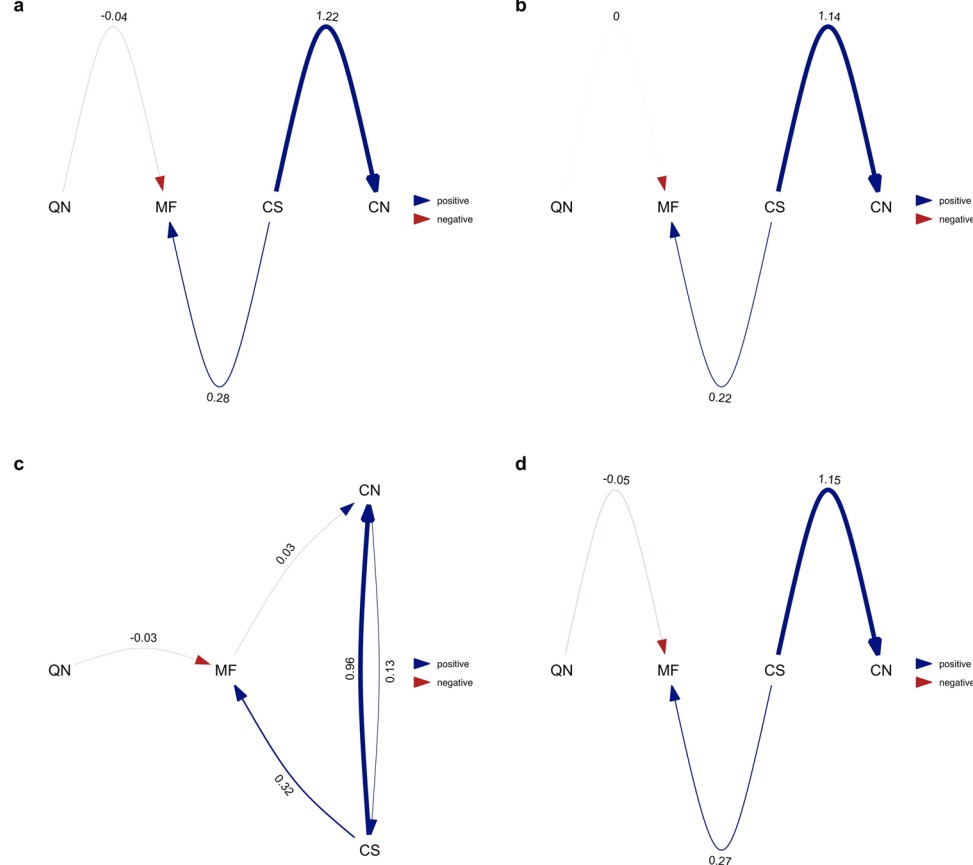

**Extended Data Fig. 2 | Average of the best-performing models where division of labour analysed as the number of physical worker castes (single caste/ multiple castes).** Analysis repeated over four different MCC ant trees from Economo et al. [65]: **a**) NC uniform stem **b**) NC uniform crown **c**) FBD stem **d**) FBD crown. Values represent standardised average coefficients. CN = number of worker castes; CS = colony size; MF = queen mating frequency; QN = number of queens per colony.

**a**

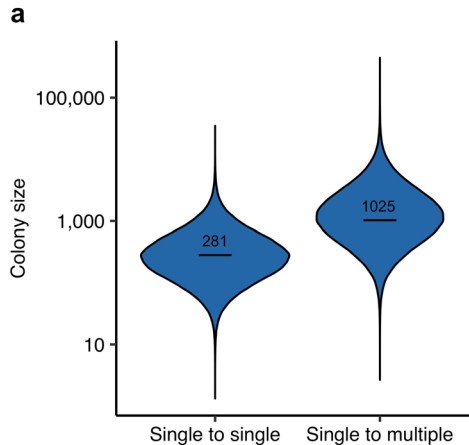

**b**

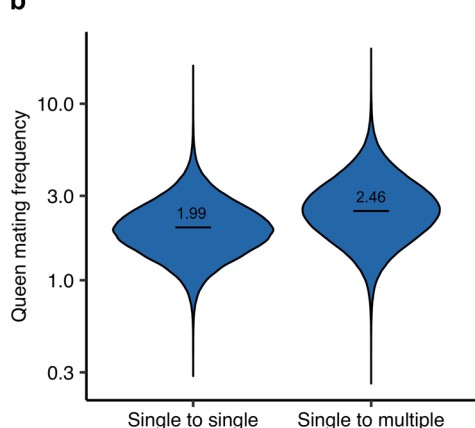

**Extended Data Fig. 3 | Ancestral state reconstruction: larger colony sizes favour the evolution of multiple worker castes.** (**a**) The posterior distribution from a BPMM estimating colony size in ancestors with a single worker caste that led to descendants with a single caste (single to single) or multiple castes (single to multiple). Colony size is larger in the ancestors of lineages that evolve multiple castes, $P_{single\,vs\,multiple} = 0.003$ ($n_{species} = 436$). (**b**) The posterior distribution from

a BPMM estimating queen mating frequency in ancestors with a single worker caste that led to descendants with a single caste (single to single) or multiple castes (single to multiple). Queen mating frequency was not significantly different in the ancestors of lineages that evolved multiple castes and the ancestors of lineages that retained a single caste, $P_{single\,vs\,multiple} = 0.16$ ($n_{species} = 104$).

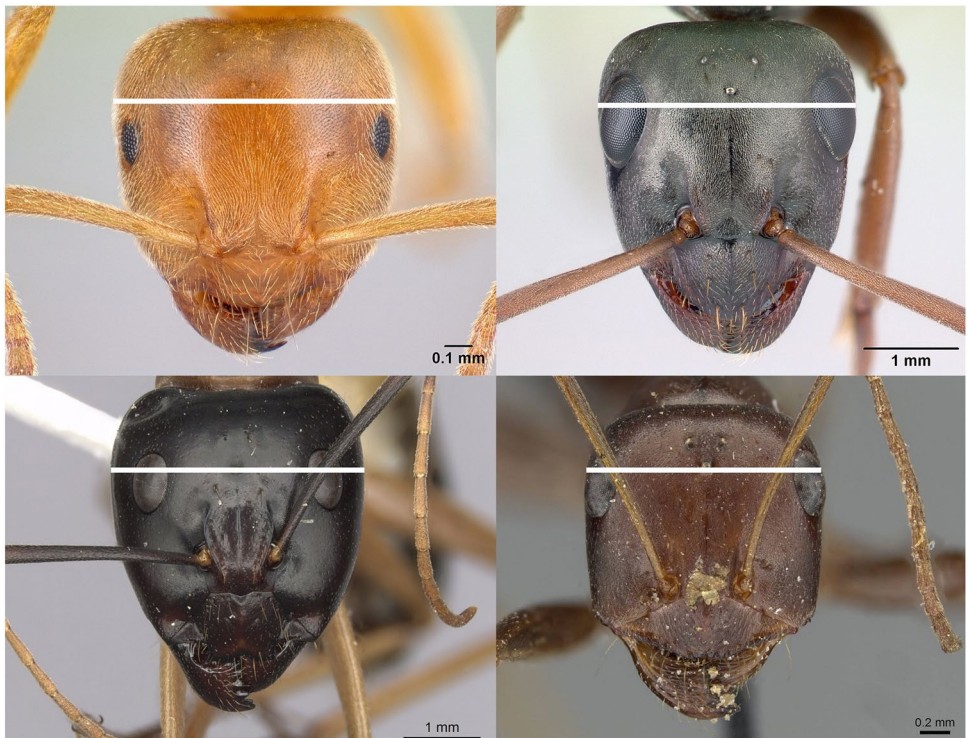

**Extended Data Fig. 4 | Measurement of ant head width.** We downloaded front-view images of ant heads from www.antweb.org and measured at the widest point excluding the eyes (horizontal white line) using the image-processing software ImageJ. © California Academy of Sciences. Clockwise from top left, images by Erin Prado, Erin Prado, Will Ericson, and Z. Lieberman. Specimen codes: CASENT0179923, CASENT0179924, CASENT0280193, CASENT0911107. All images licensed under CC BY 4.0. https://creativecommons.org/licenses/by/4.0/.

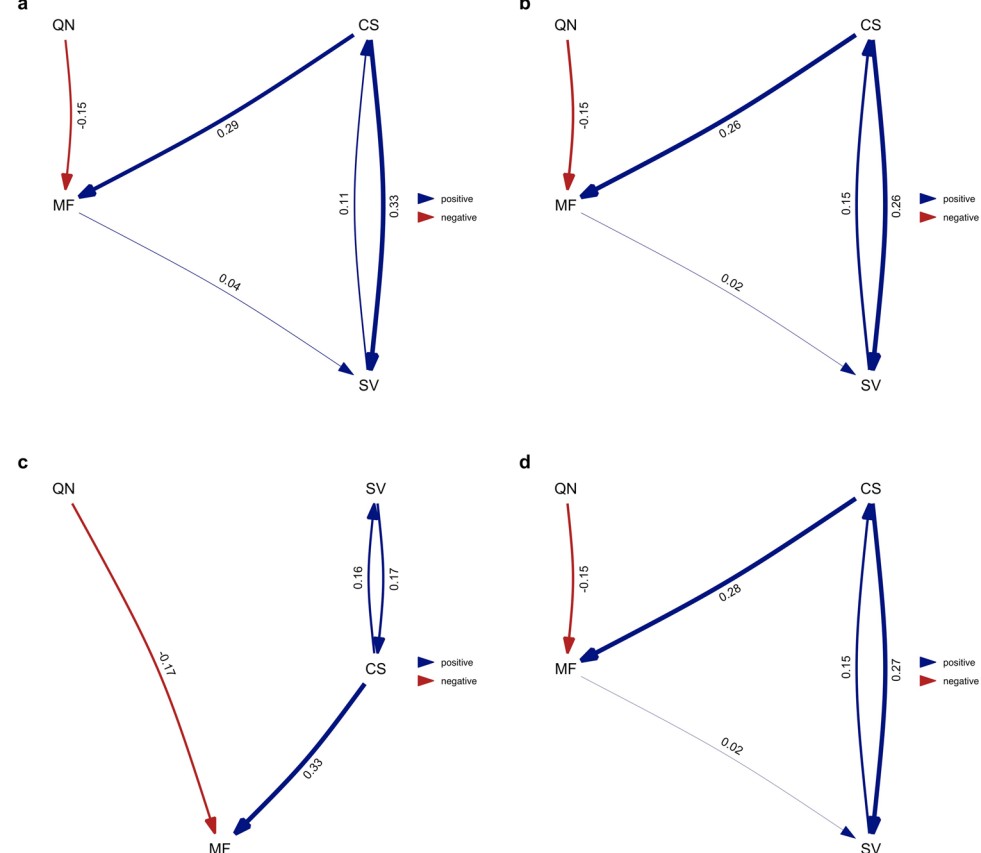

**Extended Data Fig. 5 | Average of the best-performing models where division of labour analysed as variation in worker size.** Analysis performed over all species for which data on all four traits was available. Analysis repeated over four different MCC ant trees from Economo et al. [65] **a)** NC uniform stem **b)** NC uniform crown **c)** FBD stem **d)** FBD crown. Values represent standardised average coefficients. SV = variation in worker size (degree of worker polymorphism); CS = colony size; MF = queen mating frequency; QN = number of queens per colony.

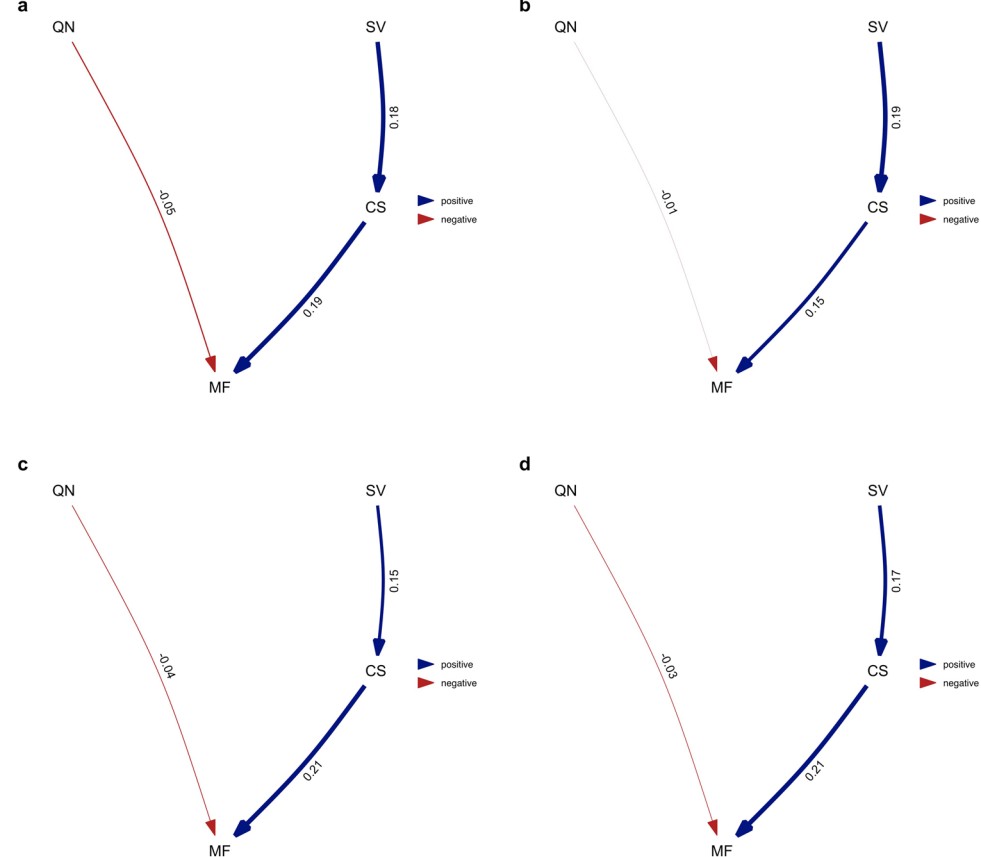

**Extended Data Fig. 6 | Average of the best-performing models where division of labour analysed as variation in worker size and only using species with a single physical worker caste.** Analysis repeated over four different MCC ant trees from Economo et al. [65]: **a)** NC uniform stem **b)** NC uniform crown **c)** FBD stem **d)** FBD crown. Values represent standardised average coefficients. SV = variation in worker size (degree of worker polymorphism); CS = colony size; MF = queen mating frequency; QN = number of queens per colony.

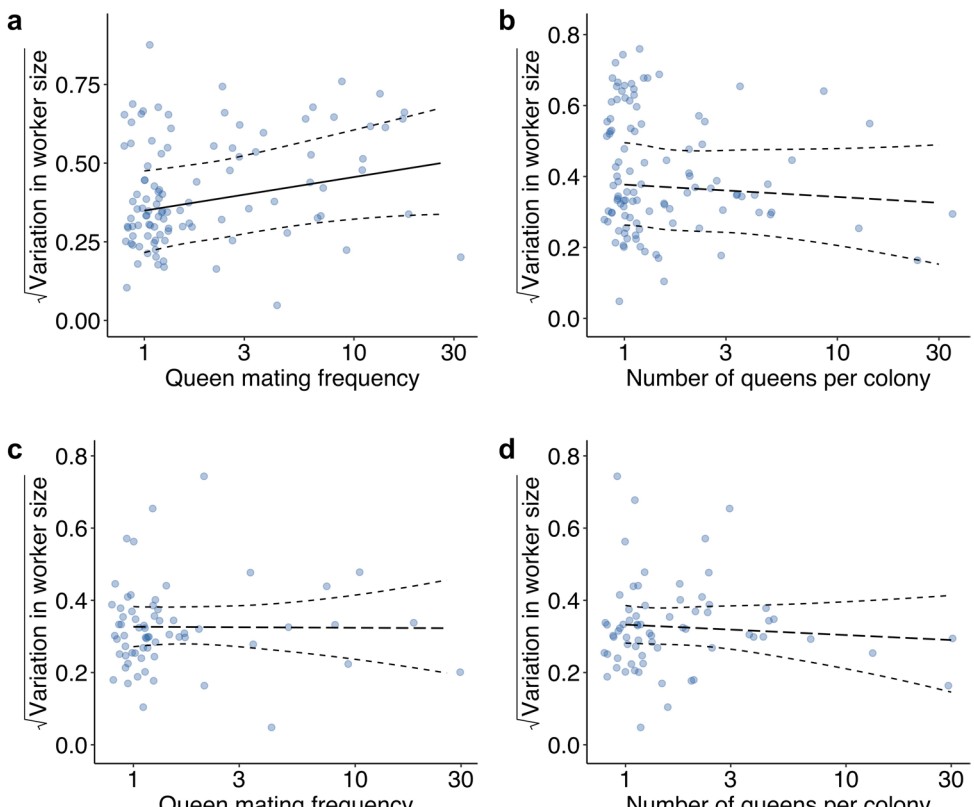

**Extended Data Fig. 7 | Relationships between queen mating frequency and number of queens per colony with variation in worker size.** Species with higher queen mating frequencies exhibited significantly greater variation in worker size (**a**, BPMM: β = 0.12, CI = 0.01 to 0.23, $n_{species}$ = 105). We did not find a significant association between number of queens per colony and variation in worker size (**b**, BPMM: β = −0.03, CI = −0.12 to 0.07, $n_{species}$ = 106). When reanalysing our data so that we only included species with a single worker caste (**c-d**), we found no association between either queen mating frequency or number of queens per colony with variation in worker size (c, BPMMs: queen mating frequency: β = −0.00, CI = −0.10 to 0.10, $n_{species}$ = 65; d, queen number: β = −0.02, CI = −0.12 to 0.07, $n_{species}$ = 68). Fitted lines are mean regression slopes with 95% CIs from BPMMs using a single phylogenetic tree. Solid regression lines represent significant relationships, while dashed regression lines represent non-significant relationships. Dots represent species averages. Axes for queen mating frequency and number of queens per colony are plotted on a $log_{10}$ scale. Axes for variation in worker size are on a square root scale.

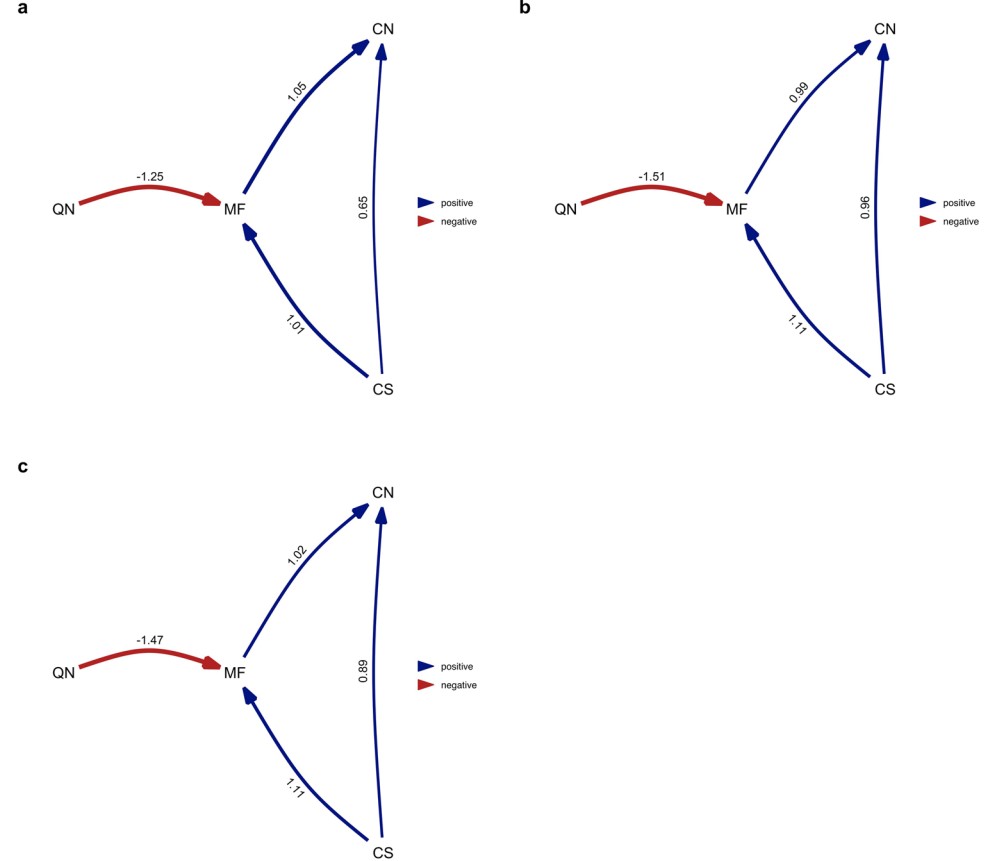

**Extended Data Fig. 8 | Average of the best-performing models where division of labour analysed as the number of physical worker castes (single caste/multiple castes).** Queen mating frequency (obligately monandrous & facultatively polyandrous or obligately polyandrous) and number of queens per colony (obligately monogynous & facultatively polygynous or obligately polygynous) were analysed as binary traits. Analysis repeated over four different MCC ant trees from Economo et al. [65]: **a**) NC uniform stem **b**) NC uniform crown **c**) FBD stem **d**) FBD crown. However, when using the FBD stem tree for the analysis, the model did not converge, and we exclude this result. Values represent standardised average coefficients. CN = number of worker castes; CS = colony size; MF = queen mating frequency; QN = number of queens per colony.

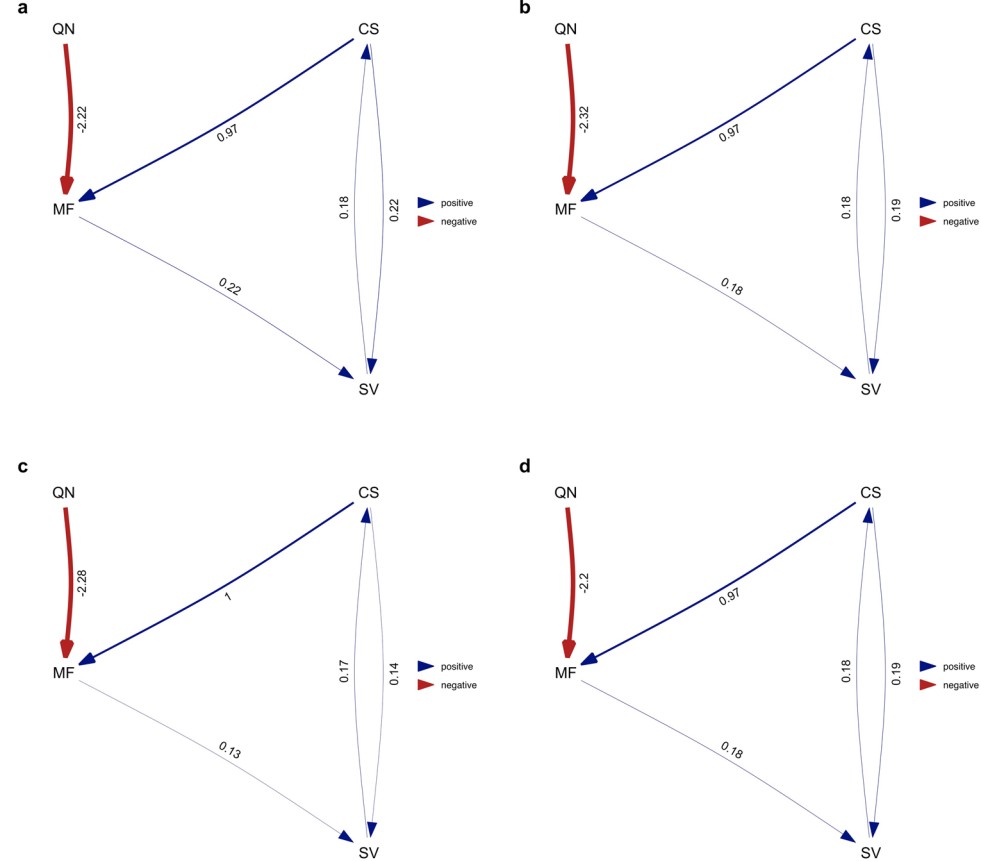

**Extended Data Fig. 9 | Average of the best-performing models where division of labour analysed as variation in worker size.** Analysis performed over all species for which data on all four traits was available. Queen mating frequency (obligately monandrous & facultatively polyandrous or obligately polyandrous) and number of queens per colony (obligately monogynous & facultatively polygynous or obligately polygynous) were analysed as binary traits. Analysis repeated over four different MCC ant trees from Economo et al. [65] **a**) NC uniform stem **b**) NC uniform crown **c**) FBD stem **d**) FBD crown. Values represent standardised average coefficients. SV = variation in worker size (degree of worker polymorphism); CS = colony size; MF = queen mating frequency; QN = number of queens per colony.

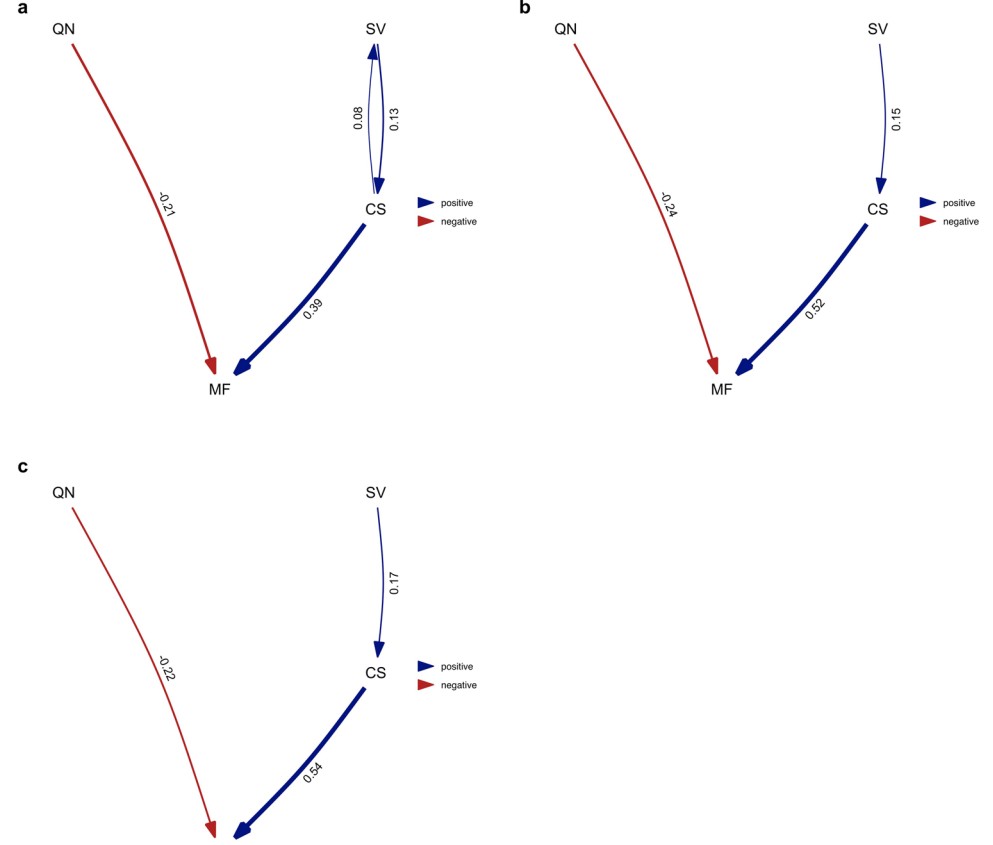

**Extended Data Fig. 10 | Average of the best-performing models where division of labour analysed as variation in worker size and only using species with a single physical worker caste.** Queen mating frequency (obligately monandrous & facultatively polyandrous or obligately polyandrous) and number of queens per colony (obligately monogynous & facultatively polygynous or obligately polygynous) were analysed as binary traits. Analysis repeated over four different MCC ant trees from Economo et al. [65]: **a**) NC uniform stem **b**) NC uniform crown **c**) FBD stem **d**) FBD crown. However, when using the NCuniform stem tree for the analysis, the model did not converge, and we exclude this result. Values represent standardised average coefficients. SV = variation in worker size (degree of worker polymorphism); CS = colony size; MF = queen mating frequency; QN = number of queens per colony.

# Reporting Summary

## Statistics

For all statistical analyses, confirm that the following items are present in the figure legend, table legend, main text, or Methods section.

| n/a | Confirmed | |
|---|---|---|
| ☐ | ☒ | The exact sample size (*n*) for each experimental group/condition, given as a discrete number and unit of measurement |
| ☐ | ☒ | A statement on whether measurements were taken from distinct samples or whether the same sample was measured repeatedly |
| ☐ | ☒ | The statistical test(s) used AND whether they are one- or two-sided<br>*Only common tests should be described solely by name; describe more complex techniques in the Methods section.* |
| ☐ | ☒ | A description of all covariates tested |
| ☐ | ☒ | A description of any assumptions or corrections, such as tests of normality and adjustment for multiple comparisons |
| ☐ | ☒ | A full description of the statistical parameters including central tendency (e.g. means) or other basic estimates (e.g. regression coefficient) AND variation (e.g. standard deviation) or associated estimates of uncertainty (e.g. confidence intervals) |
| ☐ | ☒ | For null hypothesis testing, the test statistic (e.g. *F*, *t*, *r*) with confidence intervals, effect sizes, degrees of freedom and *P* value noted<br>*Give P values as exact values whenever suitable.* |
| ☐ | ☒ | For Bayesian analysis, information on the choice of priors and Markov chain Monte Carlo settings |
| ☒ | ☐ | For hierarchical and complex designs, identification of the appropriate level for tests and full reporting of outcomes |
| ☐ | ☒ | Estimates of effect sizes (e.g. Cohen's *d*, Pearson's *r*), indicating how they were calculated |

*Our web collection on statistics for biologists contains articles on many of the points above.*

## Software and code

Policy information about availability of computer code

| | |
|---|---|
| Data collection | ImageJ v1.53 used to measure ant head widths. |
| Data analysis | All data analysis in BayesTraits V4 or R v4.2.2. Packages used: MCMCglmm v2.34; coda v0.19-4; ape 5.7-1; dplyr v1.1.2; phylopath v1.1.3; corHMM v2.8. R code for analyses can be found at: https://github.com/LouisBell-Roberts/Larger_colony_sizes_favoured_the_evolution_of_more_worker_castes_in_ants. |

For manuscripts utilizing custom algorithms or software that are central to the research but not yet described in published literature, software must be made available to editors and reviewers. We strongly encourage code deposition in a community repository (e.g. GitHub). See the Nature Portfolio guidelines for submitting code & software for further information.

## Data

Policy information about availability of data

All manuscripts must include a data availability statement. This statement should provide the following information, where applicable:
- Accession codes, unique identifiers, or web links for publicly available datasets
- A description of any restrictions on data availability
- For clinical datasets or third party data, please ensure that the statement adheres to our policy

All data are provided in Supplementary Table 20 and will be made available, along with full citations of their associated references, in the public repository Dryad: https://datadryad.org/stash/share/UTQz5aoAgPXrwAJihu-Kln43Jpo30q6514MeYoB-5Xc.

# Research involving human participants, their data, or biological material

Policy information about studies with human participants or human data. See also policy information about sex, gender (identity/presentation), and sexual orientation and race, ethnicity and racism.

| | |
|---|---|
| Reporting on sex and gender | NA |
| Reporting on race, ethnicity, or other socially relevant groupings | NA |
| Population characteristics | NA |
| Recruitment | NA |
| Ethics oversight | NA |

Note that full information on the approval of the study protocol must also be provided in the manuscript.

# Field-specific reporting

Please select the one below that is the best fit for your research. If you are not sure, read the appropriate sections before making your selection.

☐ Life sciences       ☐ Behavioural & social sciences       ☒ Ecological, evolutionary & environmental sciences

For a reference copy of the document with all sections, see nature.com/documents/nr-reporting-summary-flat.pdf

# Ecological, evolutionary & environmental sciences study design

All studies must disclose on these points even when the disclosure is negative.

| | |
|---|---|
| Study description | We find strong support for the size-complexity hypothesis, with larger colony sizes appearing to favour the evolution of greater division of labour, with more worker castes and greater variation in worker size. In contrast, we did not find consistent support for alternative hypotheses for variation in division of labour being explained by either queen mating frequency or number of queens per colony.<br>Using phylogenetic comparative analyses of data on 794 species of ants from 160 different genera we show that:<br>• Colony size and queen mating frequency are both positively correlated with division of labour. However, queen number is not correlated with any other variable.<br>• We reveal a distinct evolutionary pattern where large colony sizes tend to evolve before multiple worker castes.<br>• While queen mating frequency is correlated with greater division of labour, multiple mating does not consistently evolve before the evolution of multiple worker castes. |
| Research sample | We analyse published data on 794 species of ants. Full details of the species studied are given in Supplementary Table 20. |
| Sampling strategy | We gathered data from major reviews, comparative studies and books for queen mating frequency, number of queens, number of worker castes and colony size (Blanchard & Moreau, 2017; Hughes et al., 2008; Hölldobler & Wilson, 1990; Burchill & Moreau, 2016). Data was also made available to us by the Global Ant Genomics Alliance (GAGA) consortium. We then searched published literature to retrieve all studies detailing queen mating frequency, the number of queens and the number of worker castes. We then collected available data on colony size focusing on species where data was already present for queen mating frequency or number of queens. We collected data on variation in worker size for species that we already possessed data for queen mating frequency and the number of queens. Full details on how literature was found are provided in the Methods. |
| Data collection | We searched published literature using the search engine Web of Science. For queen mating frequency and the number of queens we used the following key words "Ant" AND (monandr* OR monogyn* OR polyandr* OR polygyn* OR effective-mating-frequenc* OR mating-frequenc* OR paternity-frequenc* OR mating-system* OR sociogenetic-structure*). For the number of worker castes we used "Ant" AND (worker polymorph* OR worker monomorph* OR (morphometric AND worker* AND caste*) OR subcaste* OR sub-caste* OR worker dimorph* OR major-worker* OR minor-worker* OR worker AND allometr*). For colony size we used the search engine Google Scholar and searched the key words [species name] AND 'colony size OR colony collection OR worker number'. We collected data on variation in worker size by measuring ant head widths from photos available at AntWeb. |
| Timing and spatial scale | Published literature were searched during 2020-21. |
| Data exclusions | The species excluded are supplied in Supplementary Table 21 and detailed reasons for species exclusions are presented in Supplementary Table 22. We excluded 112 species in our dataset from the analysis based on their distinct life history traits, including: (i) species that formed supercolonies (vast networks of connected nests), (ii) were social parasites, lacking some or all of the worker castes (although temporary social parasites which only establish new colonies with the assistance of a host species were included in the analysis), (iii) that can reproduce parthenogenetically to produce queens or workers, (iv) that reproduce via gamergates (mated workers that reproduce sexually), or (v) that use interlineage hybridisation for genetic caste determination. We excluded these species from the analysis as they represent secondary reductions of complexity in social organisation and are likely experiencing |

| Reproducibility | different selection pressures for either the evolution of worker castes, colony size, queen mating frequency or number of queens per colony. This decision improves our ability to detect which variables could influence the evolution of the number of worker castes. |
| --- | --- |
| Reproducibility | All data analyses were performed in the open source software BayesTraits V4 or R v4.2.2. Packages used: MCMCglmm v2.34; coda v0.19-4; ape 5.7-1; dplyr v1.1.2; phylopath v1.1.3; corHMM v2.8. Reproducible R scripts for analyses can be found at: https://github.com/LouisBell-Roberts/Larger_colony_sizes_favoured_the_evolution_of_more_worker_castes_in_ants. |
| Randomization | NA. The study was a comparative analysis of published data. We did not make groups. |
| Blinding | NA. The study was a comparative analysis where we collected all data found in published papers for traits of interest. We analysed all species that were not excluded for the reasons detailed in the Supplementary Tables 21 and 22. |

Did the study involve field work? ☐ Yes ☒ No

# Reporting for specific materials, systems and methods

We require information from authors about some types of materials, experimental systems and methods used in many studies. Here, indicate whether each material, system or method listed is relevant to your study. If you are not sure if a list item applies to your research, read the appropriate section before selecting a response.

## Materials & experimental systems

| n/a | Involved in the study |
| --- | --- |
| ☒ | Antibodies |
| ☒ | Eukaryotic cell lines |
| ☒ | Palaeontology and archaeology |
| ☐ | ☒ Animals and other organisms |
| ☒ | Clinical data |
| ☒ | Dual use research of concern |
| ☒ | Plants |

## Methods

| n/a | Involved in the study |
| --- | --- |
| ☒ | ChIP-seq |
| ☒ | Flow cytometry |
| ☒ | MRI-based neuroimaging |

## Animals and other research organisms

Policy information about studies involving animals; ARRIVE guidelines recommended for reporting animal research, and Sex and Gender in Research

| Laboratory animals | The study did not involve laboratory animals. |
| --- | --- |
| Wild animals | The study involved published accounts of wild animals. The full details of the species studied are given in Supplementary table 20 and the associated references. |
| Reporting on sex | Findings only apply to females. All of the worker caste are female in ants. |
| Field-collected samples | Did not involve samples collected from the field. |
| Ethics oversight | No ethical approval or guidance required. The study only involved published accounts of wild animals. |

Note that full information on the approval of the study protocol must also be provided in the manuscript.

## Plants

| Seed stocks | NA |
| --- | --- |
| Novel plant genotypes | NA |
| Authentication | NA |

