## [Peer Review File · Nature Ecology & Evolution]

Peer Review Information

Journal: Nature Ecology & Evolution

Manuscript Title: Larger colony sizes favoured the evolution of more worker castes in ants

Corresponding author name(s): Louis Bell-Roberts

Editorial Notes:

Reviewer Comments & Decisions:

Decision Letter, initial version:

29th September 2023

Dear Professor West,

Thank you very much for your enquiry about submitting your manuscript "Testing the size-complexity hypothesis in ants" to Nature Ecology & Evolution. It sounds interesting and we would be happy to send it out for peer review as an Article. Please submit the full paper using the link below:

<https://mts-natecolevol.nature.com/cgi-bin/main.plex?el=A4Cn5Sp7A5IOj1J7A9ftdhmbHxpMe9xMZ2GcGIUSgZ>

If you have any questions, please feel free to contact me.

Yours sincerely,

[REDACTED]

Decision Letter, first revision:

27th November 2023

Dear Mr Bell-Roberts,

Your manuscript entitled "Testing the size-complexity hypothesis in ants" has now been seen by 3 reviewers, whose comments are attached. The reviewers have raised a number of concerns which will need to be addressed before we can offer publication in Nature Ecology & Evolution. We will therefore need to see your responses to the criticisms raised and to some editorial concerns, along with a revised manuscript, before we can reach a final decision regarding publication.

We therefore invite you to revise your manuscript taking into account all reviewer and editor comments. Please highlight all changes in the manuscript text file.

We are committed to providing a fair and constructive peer-review process. Do not hesitate to contact us if there are specific requests from the reviewers that you believe are technically impossible or

2unlikely to yield a meaningful outcome.

* If you have not done so already please begin to revise your manuscript so that it conforms to our Article format instructions at <http://www.nature.com/natecolevol/info/final-submission>. Refer also to any guidelines provided in this letter.

[REDACTED]

Nature Ecology & Evolution is committed to improving transparency in authorship. As part of our efforts in this direction, we are now requesting that all authors identified as 'corresponding author' on published papers create and link their Open Researcher and Contributor Identifier (ORCID) with their account on the Manuscript Tracking System (MTS), prior to acceptance. ORCID helps the scientific community achieve unambiguous attribution of all scholarly contributions. You can create and link your ORCID from the home page of the MTS by clicking on 'Modify my Springer Nature account'. For more information please visit www.springernature.com/orcid.

Yours sincerely,

[REDACTED]

Reviewer expertise:

Reviewer #1: Social evolution, hymenoptera

Reviewer #2: Phylogenetic path analysis

Reviewer #3: Eusociality, phylogenetic path analysis

Reviewers' comments:

Reviewer #1 (Remarks to the Author):

This manuscript is based on a large scale comparative analysis to unravel the selection forces that have shaped the global adaptive radiation of the ants. Earlier attempts are available, but they were all much smaller in scale and unable to separate the causal drivers from the response variables. Using state-of-the-art comparative methods for independent contrasts and causal path analysis, this study succeeds in establishing that convergent increases in colony size were the key causal factor to select for secondary diversification among the somatic worker and soldier individuals of ant colonies, both in analyses of discrete additional castes (up to four castes in total) and of continuous size variation within these castes. This is a major achievement that will likely stand as the 'last word' on this important question for many years.

Our major comments concern two key aspects of the study: the contextual phrasing (mostly in the Abstract and Introduction) and the robustness of the dataset.

Major comment 1: Conceptual phrasing

The authors use terms like 'groups' and 'colonies' interchangeably, which is both confusing and – in our opinion – conceptually incorrect. We are aware that 'group' language has been used repeatedly (ultimately based on Bourke, 2011 I believe), but for ants that is misleading because it wrongly suggests that what Darwin called 'neuter' castes arose by group selection, i.e. by pleometrotic coming together of multiple queens at colony founding. Darwin knew better than that and used 'family selection' on the parents (which he implicitly assumed were monogamous) as hypothetical explanation. This conjecture was later confirmed by the formal monogamy hypothesis and the test by Hughes et al. (Science, 2008), which was convincing in spite of the unhelpful all-inclusive eusociality terminology. Darwin thus solved in outline the problem of obligate altruism, later reinforced by August Weismann, Julian Huxley and William Morton Wheeler who formalized this by acknowledging that the colonies of ants, honeybees, stingless bees, bumblebees, yellowjacket wasps and higher termites are higher-level 'organismal', or 'superorganismal'. In contrast, a century later Hamilton solved the problem of facultative, condition-dependent altruism in open societies, focusing primarily on wasps and humans. Importantly, he also derived the key mathematical equation ($br > c$) specifying the simultaneous necessary and sufficient conditions for the dynamics of kin selection and inclusive fitness maximization. As it appeared, Hamilton's rule can also be used to formalize Darwin's neuter caste explanation, in which case the rule partitions in a necessary condition of relatedness-equivalence and

3a lasting $b > c$ sufficiency condition, which may then produce irreversible major transitions to higher-level organismality – the hymenopteran neuter-caste colonies as we know them from ants, crown-group corbiculate bees (honeybees, stingless bees, bumblebees) and vespine yellowjacket wasps.

The implication is, as worked out in your first reference, that condition-dependent individual altruism was never ancestral to obligate collective neuter-altruism – both evolved as sister lineages from solitary ancestry. Calling everything a 'group' muddies that logic by pretending that social systems owing to 'coming together' and 'staying together' are comparable. Merging these two, as done for example in broad-brush undifferentiated terms like eusociality and multicellularity, hides the crucial major transitions and the key differences between open societies and foundationally closed (super) organisms (some drifting later in the colony life cycle does not challenge this distinction). The available comparative evidence consistently supports this interpretation (see Chapter 6 in ref 1). This brings us to our point pertaining to your manuscript. The size complexity-hypothesis was originally developed without making the distinction between loose (coming together) societies and firm (core-family-based) organismal colonies. The early testing of this hypothesis happened across societies and organismal colonies, which was inappropriate for the obvious comparative-analysis reason that there are separate independent contrasts within open insect societies (which have small colonies on average) and within the lineages with organismal (and on average larger) colonies. Your study is burdened by that legacy unless you make explicit that you have avoided this problem by focusing on a single monophyletic clade (the ants) with exclusively organismal colonies that originated from a single ancestral ant with full-sibling colonies. It would thus be helpful if you avoid unwarranted suggestions of parallels between ant castes and division of labor in e.g. humans, wasps and microbes, which never evolved neuter castes or bodies with collectively altruistic somaticized cells. The appropriate analogy comparison is with the somatic metazoan bodies of solitary animals, i.e. between body size of solitary insects and colony size in ants (not group size as in line 17 and 31). Things are also not 'reliant on the 'group' but more specifically reliant on 'cell members of somatic bodies' or on 'individual ant members of organismal colonies'. Also, superorganisms are not societies (line 29), at least not at their full-sibling MTE-origin. In this light, please revise the opening sentence of the Abstract and the first paragraph of the Introduction, and reconsider your use of 'group' and 'society' throughout the manuscript.

Major comment 2: The importance of coherence in comparative data across studies

For this review, Goujie Zhang's postdoc Joel Vizueta and I joined forces to compare your raw data with a similar dataset that we compiled since 2021 to be used as background matrix for the comparative analysis of 163 high-quality genomes across the global ant genera (the GAGA project: <https://db.cngb.org/antbase/project>, which is nearing completion and submission in spring 2024). We believe that both studies would be well-served by coordinating to make these datasets as complete and aligned as they possibly can be. We have uploaded a separate file to report our comparisons and the conclusions drawn from this exercise. Fortunately, there is a reassuringly high similarity between our independent efforts, but there are a number of discrepancies that we propose Louis and Joel correspond about, to mutually validate and to fill in some more missing data and thus increase the statistical power both in your and our analyses.

An independent coherence concern has been the way in which continuous data are boiled down to dichotomies that statistical analyses can handle. We note that your analyses are based on rather arbitrary cut-offs for colony size, queen-insemination (extent of polyandry) and queen number per

4colony (extent of polygyny). You are as aware as we are of the unavoidable inaccuracies of comparative literature data for such variables, but we feel that maintaining the continuous character of the polyandry and polygyny variables may have removed (in spite of your transformations) some of the contrasts that might have caused these variables to be of no or minor significance. Previous reviews (Boomsma & Ratnieks, 1996; Hughes et al, 2008; Boomsma et al. 2009 - a book chapter cited in Boomsma 2022) have used 3 distinct categories for each of these social variables that we feel might well be more efficient in capturing the contrasts between ancestral monogyny+monoandry and evolutionarily derived polyandry or polygyny because they likely represent natural kinds (to use language by Thomas Kuhn).

These categories are: 1. The maintenance of the ancestral state (single mother+single father), 2. The evolution of facultative polyandry or polygyny, with as key criterion that mature monandrous and monogynous colonies still occur in the same population (hence this evolutionary change has remained principally reversible), and 3. The evolution of obligate polyandry or polygyny, with as key criterion that mature colonies are never monandrous or monogynous anymore (hence an irreversible transition to an evolutionary derived social system via either of the two factors that significantly increase colony-level genetic diversity. In this context, facultative polyandry/polygyny hardly makes a difference in average colony-level genetic diversity and seems very unlikely to have a connection with disease defense because that should have induced further selection for obligate polyandry/polygyny – we do not know of any such facultative to obligate transition).

In the interest of maximizing comparability with previous studies and validating your overall conclusions, we would like to see you repeat your analyses with these natural kind categorizations. We know that some of your analyses cannot handle 3 categories, but facultative and obligate polyandry evolved independently and almost certainly for different reasons (see Boomsma, 2013; Beyond promiscuity ... Phil Trans). We strongly suspect this is also true for facultative and obligate polygyny because there are multiple examples (the fire ant is one of them) where the same ant species has populations with strict monogyny next to populations with obligate polygyny and without intermediates. Following this line of thought, your contrasts in the analyses that require binary traits would then be: single vs facultative and single vs obligate, and if the first of these contrasts does not deliver (likely, because facultative remains reversible) then also single+facultative vs obligate. Your way of analyzing mating frequency data may not be too far from the latter contrast, but the way in which you have categorized polygyny (lines 399-402) seems to effectively blur any effective contrast – and this is the variable that never gives anything in path analysis. We thus believe that a validation of this kind is necessary to make your conclusions robust beyond reasonable doubt. If these validations confirm that neither obligate polygyny nor obligate polyandry have ever been a driving causal force that would be great but, as already indicated, your current way of handling these variables (particularly polygyny) may have removed contrasts that might have made it too easy for colony size to be the only causal variable.

In addition, the analyses including caste number seem not to distinguish between monomorphic workers and continuous worker polymorphism (variation in body size with allometric scaling within a monomorphic worker caste). This is addressed later by analyzing worker size variation separately, but that dataset only covers 117 species, a huge reduction compared to your discrete caste number analysis (608 species). It is unclear to us whether these two analyses were really nested (in the sense of addressing continuous size variation within each of the 1-4 discrete castes), and if it was could it then not have been done in a single analysis with two levels of variation?

Minor comments:

1. Please insert a few explicit lines saying that your study confirmed that monogyny and monandry were the (monophyletic) ancestral states when the ant MTE evolved. I cannot imagine that your analyses would fail to show this with high confidence. Such an update conclusion, relative to Hughes et al. (2008), would be really valuable because his analyses used broad-brush eusociality language and did therefore not specifically address the origins of the MTEs to organismal colonies. It somehow seems odd that you have done this for establishing that a single worker caste is ancestral (line 681) but not for a single queen or queen-mate being ancestral.
2. Line 23. Consider supplementing this sentence with something like: 'support for the size-complexity hypothesis in a monophyletic clade of social insects that emanated from a single major transition to permanently caste-differentiated colonies.'
3. Line 26. 'facilitating evolutionary transitions' is rather 'sitting on the fence'. Some DoL may evolve in open societies of cells or multicellular individuals that will then change social interactions (e.g. yeasts), but as long as these tendencies are variable and based on condition-dependent cooperation such developments remain potentially reversible and will not induce transitions comparable to neuter-caste colonies and somatic cell differentiation in metazoan bodies. Note that condition-dependent (coming together) sociality does not preclude the evolution of some cell-type polymorphism (see Fisher, Shik & Boomsma, 2020 for a comparative analysis). The key point appears to be that not all cells are irreversibly committed to being either soma or germline as is the norm in the bilaterian Metazoa.
4. Line 56. Consider skipping ref 23, which had an irritating error for *Sericomyrmex* – see our separate file on data set assessments (ref 23 is also mentioned on line 303)
5. Line 74. Behavioral castes is a contradiction in terms that we owe to E.O. Wilson's tendency to capture everything in smooth 'eusociality'-gradients. His behavioral castes are nothing but adult reproductive role differentiation (i.e. reproductive skew). There is no relationship between adult reproductive skew and pre-imaginal caste differentiation (the original concept of caste in insects), i.e. with the key-trait that convergently evolved in the (super)organismal lineages only.
6. Line 101. The median value of 1.1 inseminations per queen underlines how arbitrary cut-offs can be. You obviously did not chose this median value, but used > 2 and ≤ 2 (line 199). However, this remains arbitrary as well, because far from all (in fact almost no) ants with polyandry >2 evolved worker policing in the way known from honeybees (one of your arguments on line 524). Nonetheless, we believe that using this > 2 cut-off may well be highly correlated with the absence of mature colonies with singly-inseminated queens, but formal validation of this suspicion seems to be called for.
7. The tree of Figure 2 is very nice but could the bars in the four shades of blue be made to stand out in a more pronounced way?
8. Line 146. Make explicit that the relationship between polygyny and polyandry was expected and observed to be negative.

69. Line 184. Given there were only three further MCC trees, consider giving all four rather than just one path-diagram to illustrate in the main paper how coherent these alternative results were? See also lines 618-619.

10. Line 197. In contrast to polyandry and polygyny, a colony-size cut-off between small and large will always be arbitrary because colony size remains an essentially continuous variable. We note that you have tried to validate this cutoff by juggling with different quantile boundaries (line 528). After considerable deliberation, the GAGA project ended up using the categorizations 'small (0-100)', 'medium (100-1000 workers)', large (>1000 workers), with unicoloniality as an additional separate category. In contrast to polyandry and polygyny, there is no way of saying a-priori that one kind of partitioning colony size variation is better than another, so this remains an empirical issue. However, because the causal effect of colony size is the main result of your study, we recommend that you apply a range of small-large boundary values to see whether polygyny and polyandry remain insignificant across this range while colony size always remains causal.

11. Line 207. Not clear why polyandry and number of worker castes should be negatively related. Seems to contradict what you have on line 229.

12. Line 247-250. I remember that Francis Ratnieks always argued that task-partitioning (TP) was something fundamentally different than division of labour (DoL), which may mean that TP cannot be ancestral to caste-based DoL, as you implicitly suggest here. Perhaps check those original pioneering papers on TP if you feel strongly about making this point.

13. Line 258-260. With this 2 versus 1 result, is it then reasonable to conclude that all you can say is that things are correlated and that you were unable to assess causality?

14. Line 309-310. Write caste phenotypes rather than just types. We do not think the microbial references given here (11, 39) are directly relevant for DoL in organismal ant colonies. A better analogy would be something like bodily versus colonial homeostasis – a term that has gone a bit out of fashion but we cannot think of a better one.

15. Line 321. See my earlier comments on the likely non-relationship between TP and DoL

16. 'influenced'? passive voice

17. 333-336. Your references about disease susceptibility here are a mixed bag of grooming-like defences in open societies of primates and much more deeply coordinated behaviors in organismal colonies of honeybees. These two are essentially incomparable, as was recently compellingly argued by Sylvia Cremer (see Figure 6.5 in reference 1 and the related text sections). Also here there is no such thing as a single group-size gradient. Neither can you take for granted that disease pressure increases proportionally with colony size across the superorganismal lineages. It makes a huge difference for disease pressure whether foragers have wings and operate in overlapping 3D territories (like honeybees) or are wingless so they operate in non-overlapping 2D territories (like ants).

18. Line 346. The late Christian Peeters offered a compelling argument for winglessness having released the worker caste of ants from a number of body-size constraints that continued to apply in bees and wasps where workers have universally retained wings (Peeters, 2012). In that light the difference that you highlight here may not be that surprising.

19. Line 359. You mean multicellular organisms here I think, not societies. The difference between queens and workers does not seem relevant because your paper is about differentiation of the metazoan soma-analogues (workers/soldiers) not about reproductive DoL (as you confirm a few lines further down).

20. Line 364. Make explicit that you did a large-scale phylogenetic analysis.

21. Line 379. Write 'queens per colony', also in other places, just to be precise.

22. Line 394. Also cite ref 24 here. It was the first to make the point that number of copulations are rather irrelevant. In many ways that early review carved out the essential predictor variables (polyandry, polygyny, colony size) about whose causal significance your paper tries to offer the definite answer.

23. Line 453. Add here something like: 'social parasites, usually lacking some or all of the worker castes'. Write and/or at the end of the line?

24. Line 457. Your arguments for excluding these cases are compelling, but make more explicit that all these cases are evolutionarily derived elaborations of the ancestral full-sibling-colony state and that it is the ensuing secondary reductions of complexity in social organization that prompted you to exclude these lineages.

25. Line 546 (also other places). Can you define somewhere what you mean when you write 'qualitatively similar'? We assume it must be something like 'Similar enough not to lead to different conclusions because direction and magnitude of (partial) correlations remained essentially the same and the (non)causative position of covariables did not change either.'?

26. Line 571. Ref 24 was the first to make this point, but the first to suggest that colony size should select for multiple queen mating was Blaine Cole (we believe 1983 – cited in the other reviews)

27. Line 572-574. See our earlier comment on disease pressure.

28. Line 576. See Boomsma (2013) Beyond promiscuity ... (Phil Trans) for arguments why obligate and facultative polyandry almost certainly have different evolutionary reasons.

29. Line 629. 'influenced' (past tense)

See also our separately uploaded file: Assessment of the dataset of Bell-Roberts.docx

Copenhagen, 22 November 2023 Koos Boomsma and Joel VizuetaAssessment of the dataset of Bell-Roberts et al.

The dataset is among the largest ever analyzed for ants, which is a major strength. However, there is a several-decades long tradition of comparative data analysis in ants, so it is important that the variables used stay as comparable as possible, both backward in history and laterally, relative to currently ongoing work using the same or similar data. The GAGA project <https://db.cngb.org/antbase/project> is such an ongoing program, the main results of which are expected to be submitted for publication in spring 2024. We therefore compared the dataset used in the present study with the GAGA data to test the similarity of the two independent surveys, as a validation of both our own and the Oxford-group efforts.

We believe that reaching maximal correspondence between the two types of analysis is important because: 1. The B-R study will stand for decades as the last word on the main selection forces that have shaped adaptive radiation in the ants. 2. We imagine that the GAGA papers expected in 2024 will have similar impact, albeit not quite as long lasting given how fast new and better genomes will increasingly become available. GAGA will produce a considerable number of convergent signatures of selection (or lack thereof) emanating from variation in colony size, caste number and worker size variation, polyandry and polygyny. The field will therefore be well-served if, when we refer to your present study, there is no doubt that we made maximal efforts to work with the same data matrix of observed variation in these key social variables.

Prelims

The B-R study includes 62 of the 163 species for which the GAGA project obtained high-quality (chromosome-level for 16 species) genome sequences. The overlap would have been higher had the B-R study not excluded (for valid reasons) about 20 GAGA species because they have gamergates, clonal reproduction or supercolonies (supp. table 14). Overall, the B-R study covers ca. five times as many species as GAGA, and relied completely on literature data while GAGA also had direct information available from collectors. This implies that for 33 out of the 62 overlapping species GAGA has entries that the B-R study lacks and which we are happy to make available in exchange for an acknowledgement if B-R et al. are interested to include them (see list 2 below). In addition, we noticed that some data available from the Romiguier et al. 2022 paper in *Current Biology* - colony sizes and worker polymorphism records for 65 ant species - are missing in the B-R dataset. We therefore recommend you have a look also at the Romiguier data to see whether they could provide some additional entries for species that GAGA did not cover.

Regarding the data entries that we both have, most of them are the same, which is reassuring for both analyses, but we found 7 cases with discrepancies in several variables (see list 1 below). Six of these seem incorrect in the B-R data base, while we concluded that one (*Messor barbarus*) was wrong (for the number of worker castes) in our data table. In another case (*Myrmica scabrinodis*) we had missed facultative polyandry.

Categorization within predictor and response variables:

- For colony size, B-R used the average number of workers in a mature colony while we used both average and maximum number of workers, as well as logarithmic range-classes. Comparing your and our colony-size estimates, we found that most were in agreement, but there were some cases with minor discrepancies that we recommend you have a look at (see list 1 below).
- For queen mating, B-R et al. used queen mating frequency based on harmonic means, while we used three categories: obligate single mating, facultative multiple mating, obligate multiple mating. When we compared the entries, the raw-data correspondence was strikingly good, but the B-R study lacks entries for four of the overlapping GAGA species. Again, you are welcome to use these data. Further, as we explained in the review we maintain that our

10

ESS
: is

categorization is more likely to cut the overall variation up in evolutionary meaningful kinds, because facultative and obligate polyandry almost certainly evolved independently from strictly single mating, – the former for enhanced diversity reasons and the latter primarily for correcting insufficient first inseminations (see Boomsma, 2013 <https://royalsocietypublishing.org/doi/full/10.1098/rstb.2012.0050>). The rationale of our categorization (also used by Boomsma & Ratnieks, 1996, Hughes et al, 2008, and in a book chapter overlooked by B-R et al. (pdf attached)) is that facultative multiple insemination never abandoned single insemination so that substantial variation across populations remains, sometimes approaching pure single mating (see <https://royalsocietypublishing.org/doi/full/10.1098/rstb.2012.0050>). In contrast, obligate polyandry represents an irreversible mating system transition where singly-mated queens are never found in mature colonies (observations of a single monandrous founding-colony should be ignored when other species in the same genus are always obligatorily polyandrous, because such an incipient colony can be assumed not to survive until maturity).

- Regarding queen number, comparison with the GAGA data revealed only two cases with discrepancies (see list 1 below), but the B-R dataset lacks a number of entries as GAGA has 30 more entries in the 62 shared species. We are unsure where this difference comes from. Adding data from the attached book-chapter pdf (2009) might help and once more we offer that you can use the data from our GAGA dataset as well. As mentioned in the review, we categorized queen number in similar categories as queen-mate number (single queen, facultatively polygynous, obligately polygynous), as we did earlier in the 2009 book chapter cited above. It is our fieldwork experience that many ant species have forms of facultative polygyny (likely underestimated when this form of polygyny is rare), but this does not challenge the conclusion that monogyny and independent monogamous colony founding remain the default in those species – additional queens may then sometimes be picked up when colonies are mature (the odd newly inseminated daughter queen; the odd alien intruder that fails to be recognized as such). But the difference with obligate polygyny is quite striking because that means one never finds a mature colony in the field with just a single queen. Also here, it seems likely that the two forms of polygyny evolved independently and in response to different selection forces as in the case of queen-mate number. To validate the robustness of your conclusions, we thus recommend that you also run your analyses with our categorization of queen number which implies the same two or three contrasts as for queen-mate number.
- Regarding the number of discrete castes, there is again good correspondence and once more we offer that you can use the seven extra entries that we have in the GAGA data set (see list 2 below). We struggled with the same categorization issues as you did, but reached slightly different categories and we did not track worker head-widths systematically (a very nice addition). Thus we ended up having just three categories: monomorphic workers, continuous worker polymorphism and discretely polymorphic castes. As we expressed in the review, we are unsure whether your two ways of analyzing worker size variation represent two levels throughout, but we note that you only had worker size variation data for 117 species out of the 608 with caste number, and that only 24 of these species overlapped with the GAGA data set. In case you did not distinguish between monomorphic and continuous worker polymorphism in the analyses of single worker caste colonies, we would recommend to include a separate category for continuous worker polymorphism in your first analysis to validate the robustness of your conclusions.

For making these comparisons across data sets Joel created a combined table with the columns highlighted in green being our GAGA data (added when available for the species or genus), and the fields highlighted in red are discrepancies found with your data:

<https://docs.google.com/spreadsheets/d/1jC7c9uEvGoAX-1dRJJdjiyS2fRDfaaB/edit?usp=sharing&oid=118398902960620366424&rtopf=true&sd=true>

11

ESS
: IS

Here are the detailed species-specific cases:

List 1 - Discrepancies:

- *Acromyrmex*: You have four *Acromyrmex* species with obligate monogyny, while we have two of these (*A. octospinosus* and *A. subterraneus*) as facultative polygynous. Also, there is a difference in colony size in some of these *Acromyrmex* species of almost one order of magnitude: e.g. *A. echinator* (you have 4,3 and we have 5,1).

- *Hypoponera opacior*: You have estimates of 50 workers per colony while we have an average of 120 with up to 600 workers (Collector; Markus H. Rüger, 2007). The reference is a thesis from the University of Munich: https://edoc.ub.uni-muenchen.de/8416/1/Rueger_Markus_H.pdf.

- *Leptothorax acervorum*: You have a colony size estimate of 45 workers similar to our initial estimate, but we updated this to 150 based on Seifert (2018).

- *Sericomyrmex mayri*: You have monogynous for this species based on Mehdiabadi & Schultz, 2010; but we have facultative polygyny based on more recent collections (Ana Jesovnik and Ted Schultz, who gave us 1-3 for the number of queens per colony). Note there has also been ambiguity about another *Sericomyrmex* species (*amabilis*). You cite (ref 23) the paper by Murakami et al concluding that this species is facultatively polyandrous. However, we later showed (with Murakami as co-author) that this result is erroneous and that for this species the higher genetic diversity was due to occasional polygyny (consistent with the *S. mayri* update), while all queens were singly inseminated. The better reference is therefore: <https://www.ncbi.nlm.nih.gov/pmc/articles/PMC1691065/pdf/12184823.pdf>. Btw, this attine ant study shows that obligate multiple insemination evolved from strictly single insemination without 'intermediate' branches with facultative polyandry. The only time facultative multiple mating was found in attine ants was for an *Acromyrmex* inquiline social parasite, which represents a secondary reduction from obligate polyandry (Sumner et al., 2000).

- *Temnothorax*: For the cases where we have the same species, the colony size estimates are underestimated in your table compared to ours which for some species were validated with Seifert's 2018 book. Examples: For *T. longispinosus* you have an average of 24 workers while we have 85 with up to 1000 in our table; for *T. nylanderi* you have 54 while we have 85 with up to 500; for *T. unifasciatus* you have 67 while we have 134 with up to 500).

- *Typhlomyrmex rogenhoferi*: For colony size you have 300 workers (Brown, 1965) while we have 2000 (from Benoît Guenard and Nate Sanders, personal communication).

- *Messor barbarus*. You have that there is one worker caste, but we had this species noted down as also having soldiers (i.e. having discrete worker polymorphism). We realized that our entry was wrong and corrected it based on your reference (Bernadou et al. 2016: <https://doi.org/10.1242/jeb.141556>). However, this species would still have continuous worker polymorphism.

Cases of continuous worker polymorphism:

12

- *Cataglyphis aenescens*: You have one worker caste, but according to our information this species has continuous worker polymorphism.

ESS
: IS

- All *Formica* species: You write they have a single worker caste, but our information (validated with Heikki) is that they all have continuous worker polymorphism.

- *Lasius flavus*: Also here we have continuous worker polymorphism.

- *Megaponera analis*: We have continuous worker polymorphism also here, based on information from the collector (Erik Frank), who has a publication under submission, while you have one worker caste without further information on continuous worker size variation.

- *Mystrium*: You have one worker caste while we have continuous worker polymorphism for this species (genus).

- *Paraponera clavata*: you have one worker caste, while we have continuous worker polymorphism (Breed and Harrison 1988 <https://www.jstor.org/stable/25085003?seq=5>).

List 2 - Missing data in your table that we have in the GAGA data set:

Most of these are in agreement with data you have for other species in the same genus, but you do not have entries for the following species:

- *Aphaenogaster subterraneus*: colonies with 2000 and up to 4000 workers.
- *Camponotus japonicus*: Facultative polygyny and colonies with 1500-2000 workers.
- *Camponotus singularis*: Monogyne and colonies of 1000 workers.
- *Carebara diversa* is facultative polygynous.
- *Cataglyphis aenescens* is monogyne.
- *Crematogaster osakensis* workers are monomorphic (one worker caste).
- *Discothyrea kamiteta* is facultative polygynous, workers are monomorphic.
- *Hypoponera opacior*: facultative polygynous.
- *Iberoformica subrufa*: obligately polygynous.
- *Kalathomyrmex emeryi*: monogynous.
- *Leptanilla*: monogynous.
- *Leptogenys diminuta*: monogynous.
- *Liomotopum microcephalum* is facultative polygynous, and colonies have up til millions of workers (Seifert 2018).
- *Manica rubida*: colony size 500, up to 1000 workers; facultative polygynous.
- *Megaponera analis*: monoandrous (from collector, paper in review).
- *Messor capitatus*: colonies of 3500-7000 workers. Monogynous.
- *Messor barbarus*: monogynous.
- *Mycetomoellerius zeteki*: monoandry and monogyny.
- *Mycetosoritis hartmanni*: monoandry and monogyny.
- *Myrmecia pilosula*: monomorphic workers.
- *Myrmecina graminicola*: colonies with on average 45 workers with up to 300 (you have 27 on average). Monoandrous and facultative polygynous.
- *Mystrium camillae*: facultative polygyny and continuous worker polymorphism.
- *Ochetellus glaber*: obligately polygynous.
- *Odontomachus brunneus* is monogynous.
- *Paraponera clavata* is monogynous.
- *Paratrachymyrmex cornetzi*: monoandrous and monogynous.
- *Polyrhachis illaudata* is monogynous.
- *Pseudomyrmex spinicola*: monogynous and polymorphic workers.
- *Stenamma debile*: facultative polygynous; colonies of 56 +/-40 workers.
- *Stictoponera bicolor*: monogynous and colonies of 480 workers.
- *Strumigenys mutica*: monomorphic workers.
- *Trachymyrmex septentrionalis*: monoandrous.

- *Typhlomyrmex rogenhoferi*: monogynous and continuous worker polymorphism.

Copenhagen, 22 November 2023

Joel Vizueta and Koos Boomsma

Reviewer #2 (Remarks to the Author):

With interest have I read the manuscript by Bell-Roberts et al., which investigates the size-complexity hypothesis in ants. They have performed various phylogenetic comparative analyses to relate the colony size of ant species ("size") with the number of worker castes ("complexity"). They find, like others before them, that species with larger colonies tend to have more worker castes. They add to this finding by excluding alternative explanations of this correlation related the number and mating behavior of queens. I found the paper to be clear and easy to read, and the analytic approach to be sound. The three separate approaches to disentangle the order and causality of effects are a strength of the work. Overall, I only have relatively minor comments. Nonetheless, the presentation of results can be improved in several places, which I outline below.

A general weak point of the presentation of this work is the treatment of effect size and uncertainty. The text should do a much better job of interpreting the effect sizes (and their credible intervals), instead of the binary significant/non-significant language currently used. This is especially prudent given the wide range of sample sizes used in different analyses. In addition, the figures do not give the reader any idea of the uncertainty of the estimates, even though the methods employed do give the authors the opportunity to do so. In particular, figures 3, 4 and 6 would be more informative with the inclusion of credible/confidence intervals or standard errors.

The introduction starts very broad, but then quickly seems to only consider multi-cellularity and social insects. In particular, the use physical castes stands out to me. I don't follow why mutual dependence requires differences in size and morphology (lines 74:76). Don't e.g. meerkats, Seychelles warblers etc. have mutual dependence? I don't recall that they have physical castes. Given that this is the only argument provided for using physical castes, and no reference is provided, this warrants a better explanation.

The authors exclude a lot of species based on several criteria. As I'm not an ant expert, I can't evaluate these criteria very well, but it is important that exclusion criteria are mentioned in the main text, as it should be apparent to which situations these results may be expected to generalize.

Line-by-line comments:

Title: The title would be more informative if it would make clear how the paper advances on the other papers that have tested the size-complexity hypothesis in ants.

19: Reporting 758 species here is technically accurate, but rather misleading. It suggests a level of power and generality that isn't borne out in the manuscript because of the large amount of missing data. I did not find a single analysis that reaches that sample size.

85: cite reference 75 here, or the original paper in Evolution.

183: MCC stands for maximum clade credibility (no extra C for consensus).

15253-266: This section overstates the support for the size-complexity hypothesis. The path analysis is suggestive but ambiguous. There is some support for both directions of effect between colony size and variation in worker size. The alternative hypothesis of queen mating frequency causing variation in worker size, in turn causing colony size should not be excluded. Again, a discussion of the relevant effect sizes and uncertainties would greatly help the reader here. It may be possible to achieve more power for this analysis by using phylogenetic trait imputation, recently made available in Thorson & Van der Bijl 2023, JEB 36 (10): 1357-1364.

449: The range of 1-31 is odd, since you can't calculate the standard deviation of a single data point.

498: This is phrased ambiguously, since the fraction of iterations where one level is greater than OR less than the other level is always 1.

515:552 While I can see why this is listed under the correlational analyses, it is confusing since this doesn't match the presentation in the main text, where only the BPMMs are presented in that section.

Tables S2 and S3: please report all models, not just the ones within 2 CICc.

I encourage the authors to present their supplemental tables in a more accessible format than the proprietary xlsx.

Reviewer #3 (Remarks to the Author):

Key results

The MS tested in ants whether increasing colony size is correlated and has a causal relationship to a stronger division of labor among workers. They quantified the division of labor in two ways: (1) physical worker caste and (2) variation in worker size. Both (1) and (2) are found to correlate with colony size in phylogenetically controlled regression. Phylogenetic path analysis that tested different causal directions between colony size and the division of labor also resulted in a consistent best support model where colony size causally affects the division of labor using both (1) and (2). Using (2) revealed additional causal relationships that were not fully explored. Using only (1), the authors also use transition rate analysis to show that transitions from 1 to >1 physical castes only occurs in species with large (>300) colony sizes. The analysis was robust to using alternative ways to categorize large/small colony sizes. Finally, analyzing the reconstructed ancestral states also shows that ancestors of species with multiple castes had larger colony sizes compared to ancestors of species with only one caste.

Validity

The data collection and analysis were done with care and using the current best methods. However, additional analyses could be done to improve the breadth of the conclusions.

Originality and significance

The study is original and gets at the problem in previous studies that are mostly based on the

16correlation between colony size and the number of worker castes. The study is significant for both the study of animal social evolution and to evolutionary biology in general because the use of path analysis and transition modeling can reveal the direction of casualty in evolution, which has been a limit of a field that is mostly based on correlation analysis and which experimental validation is almost impossible in most taxa.

Data & methodology

I have two concerns with how data are coded/analyzed.

First, I'm most uncomfortable with coding the number of queens as binary (one vs. >1 queen) (line 135). While the authors have provided some justifications for doing so (line 405), I don't see why they must code the number of queens as binary, and that they absolutely cannot use the reported values of the number of queens in the analysis. See * below for more details. An additional analysis that treats the number of queens as a continuous variable will help to resolve whether the lack of relationship in Fig 3e is due to losing power in doing an ANOVA vs. a regression analysis.

Second, the using of two ways (no. of physical worker caste and variation in worker size) to quantify worker division of labor is a valuable aspect of this study. However, some of the important conclusions (see ^ below) will be better supported if additional analysis is done using variation in worker size with ONLY species that have a single worker caste. An ancestral state reconstruction should also be done using the variation in worker size. This could hopefully also resolve the presence of additional causal relationships in the path analysis when the variation in worker size.

More detailed comments are below:

Line 36: It'll be helpful to provide some context for the readers on why it's the leading explanation. The later part of the paragraph cites a few studies in cell and insect. But how do these studies make the size-complexity the leading explanation?

Line 52: A single queen can also mate multiple times, so should "queen" be "queen(s)"?

Line 55-56: Please be specific. Is the number of queens also correlated with group size?

Line 65-67: Can you rephrase this sentence without all the "one /another variables"? It doesn't help with reading.

Line 68-69: I'd say "this pattern leads to the false impression that larger group directly favors increased division of labor and misses the involvement of an intermediate factor."

* Line 135: Please justify why the number of queens is reported (Fig 2) and analyzed (Fig 3c) as a discrete variable and not a continuous variable. This is very important since you removed the number of queens in the causality analyses based on not finding a significant relationship between the number of queens and other variables. What if you found a significant relationship when a continuous variable of queen number is used?

* Line 405: I see the argument that the observed number of queens could be an overestimate of the effective number of queens. But then why can't the idea of a harmonic mean be applied here as you have for queen mating frequency? And while it may be an overestimate, why is coding it as binary better than keeping it as a continuous variable? I'm worried that when using the binary variable, the analysis may lose power to detect what could be in there.

Line 169: This section appears to be overly brief in comparison to other pieces in the result section - although the result is very important. The competing models and the verdicts of testing only 4 models should be explained here instead of in the method section.

17

Line 175: Please at least report the omega values here to show how the best model compares with the competing models.

^ Line 238, 313-321: This is a great additional analysis, but I wonder whether the result is still mainly driven by the drastic CV in species with different physical worker castes. The present analysis only supports the statement in line 315-317 "These results indicate that our findings are robust to analysing division of labour as either the number of physical worker castes or as variation in worker size". However, the analysis is not enough to conclude that "This applies not only to species with different numbers of physical worker castes, but also to species that have just a single worker caste" (314-315) nor "positive correlation between colony size and variation in worker size is not just the result of variation across species with different numbers of physical worker castes." (317-318). To support these statements, additional analyses with only species that have a single physical worker caste should be done. And if the pattern still holds, then it will directly support the statements above - which is very important in understanding how colony size affects the division of labor.

Line 275: Can you explain why that when analyzing the variation in worker size, there are two additional arrows (variation in worker size  colony size & queen mating frequency  variation in worker size). Analytically, I understand that it's coming from the 2nd best-supported model, but in terms of the evolution of worker caste, what does it mean? An ancestral state analysis is not used for the variation of worker size, but it may allow you to tease out what happened. Perhaps in parts of the phylogeny, you see some support for the second-best model. And it may even be related to only those species that have a single physical caste. Rather than opening that for interpretation by readers, I believe it will improve the paper to dig in more about why additional, despite weaker, causal relationships exist when a different way of quantifying caste is used.

Line 321: Any citations to size-based task partitioning?

Line 326: " the number of queens " is really single vs. multiple queen according to the analysis.

Line 409-427: Since two methods were used to determine the number of worker castes (morphometric & literature search), did you do a data validation using species with morphometric to test whether the literature search yielded the same result as the morphometric classification?

Line 496: I don't see any reports of p values for BPMMs in the result section.

Line :

Line 556: The result section reported the path analysis first. Please move this part of the method before the BayesTrait.

Line 563: (ii) is your model four in Extended Data Fig. 1. Please fix this.

Line 606: This part is missing a subtitle.

*****END*****

Author Rebuttal, first revision:

Response to reviewers

*Manuscript number:* NATECOLEVOL-23092284A

*Title:* Larger colony sizes favoured more worker castes in ants

We thank all reviewers for their insightful and constructive comments. We believe that the
revised paper is stronger as a result of peer review.

In this document, we use **black text** for the reviewers' comments, and **blue text** for our
response.

Reviewer expertise:

Reviewer #1: Social evolution, hymenoptera

Reviewer #2: Phylogenetic path analysis

Reviewer #3: Eusociality, phylogenetic path analysis

Reviewers' comments:

Reviewer #1 (Remarks to the Author):

This manuscript is based on a large scale comparative analysis to unravel the selection forces
that have shaped the global adaptive radiation of the ants. Earlier attempts are available, but
they were all much smaller in scale and unable to separate the causal drivers from the
response variables. Using state-of-the-art comparative methods for independent contrasts and
causal path analysis, this study succeeds in establishing that convergent increases in
colony size were the key causal factor to select for secondary diversification among the
somatic worker and soldier individuals of ant colonies, both in analyses of discrete additional
castes (up to four castes in total) and of continuous size variation within these castes. This is
a major achievement that will likely stand as the 'last word' on this important question for
many years.

Our major comments concern two key aspects of the study: the contextual phrasing (mostly
in the Abstract and Introduction) and the robustness of the dataset.

Major comment 1: Conceptual phrasing

The authors use terms like 'groups' and 'colonies' interchangeably, which is both confusing
and – in our opinion – conceptually incorrect. We are aware that 'group' language has been
used repeatedly (ultimately based on Bourke, 2011 I believe), but for ants that is misleading
because it wrongly suggests that what Darwin called 'neuter' castes arose by group selection,
i.e. by pleometrotic coming together of multiple queens at colony founding. Darwin knew
better than that and used 'family selection' on the parents (which he implicitly assumed were
monogamous) as hypothetical explanation. This conjecture was later confirmed by the formal

19

ess

is

43 monogamy hypothesis and the test by Hughes et al. (Science, 2008), which was convincing in
spite of the unhelpful all-inclusive eusociality terminology. Darwin thus solved in outline the
problem of obligate altruism, later reinforced by August Weismann, Julian Huxley and
William Morton Wheeler who formalized this by acknowledging that the colonies of ants,
honeybees, stingless bees, bumblebees, yellowjacket wasps and higher termites are higher-
level ‘organismal’, or ‘superorganismal’. In contrast, a century later Hamilton solved the
problem of facultative, condition-dependent altruism in open societies, focusing primarily on
wasps and humans. Importantly, he also derived the key mathematical equation ($br > c$)
specifying the simultaneous necessary and sufficient conditions for the dynamics of kin
selection and inclusive fitness maximization. As it appeared, Hamilton’s rule can also be used
to formalize Darwin’s neuter caste explanation, in which case the rule partitions in a
necessary condition of relatedness-equivalence and a lasting $b > c$ sufficiency condition, which
may then produce irreversible major transitions to higher-level organismality – the
hymenopteran neuter-caste colonies as we know them from ants, crown-group corbiculate
bees (honeybees, stingless bees, bumblebees) and vespine yellowjacket wasps.

The implication is, as worked out in your first reference, that condition-dependent individual
altruism was never ancestral to obligate collective neuter-altruism – both evolved as sister
lineages from solitary ancestry. Calling everything a ‘group’ muddies that logic by
pretending that social systems owing to ‘coming together’ and ‘staying together’ are
comparable. Merging these two, as done for example in broad-brush undifferentiated terms
like eusociality and multicellularity, hides the crucial major transitions and the key
differences between open societies and foundationally closed (super) organisms (some
drifting later in the colony life cycle does not challenge this distinction). The available
comparative evidence consistently supports this interpretation (see Chapter 6 in ref 1).
This brings us to our point pertaining to your manuscript. The size complexity-hypothesis
was originally developed without making the distinction between loose (coming together)
societies and firm (core-family-based) organismal colonies. The early testing of this
hypothesis happened across societies and organismal colonies, which was inappropriate for
the obvious comparative-analysis reason that there are separate independent contrasts within
open insect societies (which have small colonies on average) and within the lineages with
organismal (and on average larger) colonies. Your study is burdened by that legacy unless
you make explicit that you have avoided this problem by focusing on a single monophyletic
clade (the ants) with exclusively organismal colonies that originated from a single ancestral
ant with full-sibling colonies. It would thus be helpful if you avoid unwarranted suggestions
of parallels between ant castes and division of labor in e.g. humans, wasps and microbes,
which never evolved neuter castes or bodies with collectively altruistic somaticized cells. The
appropriate analogy comparison is with the somatic metazoan bodies of solitary animals, i.e.
between body size of solitary insects and colony size in ants (not group size as in line 17 and
31). Things are also not ‘reliant on the ‘group’ but more specifically reliant on ‘cell members
of somatic bodies’ or on ‘individual ant members of organismal colonies’. Also,
superorganisms are not societies (line 29), at least not at their full-sibling MTE-origin. In this
light, please revise the opening sentence of the Abstract and the first paragraph of the
ntroduction and reconsider your use of ‘group’ and ‘society’ throughout the manuscript.

20

ess
is

87 In light of the reviewer's comments, we have revised our use of the terms 'group' and
88 'society' throughout the manuscript.

Major comment 2: The importance of coherence in comparative data across studies

For this review, Goujie Zhang's postdoc Joel Vizueta and I joined forces to compare your
raw data with a similar dataset that we compiled since 2021 to be used as background matrix
for the comparative analysis of 163 high-quality genomes across the global ant genera (the
GAGA project: <https://db.cngb.org/antbase/project>, which is nearing completion and
submission in spring 2024). We believe that both studies would be well-served by
coordinating to make these datasets as complete and aligned as they possibly can be. We
have uploaded a separate file to report our comparisons and the conclusions drawn from this
exercise. Fortunately, there is a reassuringly high similarity between our independent efforts,
but there are a number of discrepancies that we propose Louis and Joel correspond about, to
mutually validate and to fill in some more missing data and thus increase the statistical power
both in your and our analyses.

We would like to thank the reviewer for making their data available for use in our analysis
and which has contributed to improving our statistical power. We will include an
acknowledgement for their help in the paper. We respond individually to each species
highlighted by the reviewer in the supplementary material regarding the data.

An independent coherence concern has been the way in which continuous data are boiled
down to dichotomies that statistical analyses can handle. We note that your analyses are
based on rather arbitrary cut-offs for colony size, queen-insemination (extent of polyandry)
and queen number per colony (extent of polygyny). You are as aware as we are of the
unavoidable inaccuracies of comparative literature data for such variables, but we feel that
maintaining the continuous character of the polyandry and polygyny variables may have
removed (in spite of your transformations) some of the contrasts that might have caused these
variables to be of no or minor significance. Previous reviews (Boomsma & Ratnieks, 1996;
Hughes et al, 2008; Boomsma et al. 2009 - a book chapter cited in Boomsma 2022) have used
3 distinct categories for each of these social variables that we feel might well be more
efficient in capturing the contrasts between ancestral monogyny+monoandry and
evolutionarily derived polyandry or polygyny because they likely represent natural kinds (to
use language by Thomas Kuhn).

These categories are: 1. The maintenance of the ancestral state (single mother+single father),
2. The evolution of facultative polyandry or polygyny, with as key criterion that mature
monandrous and monogynous colonies still occur in the same population (hence this
evolutionary change has remained principally reversible), and 3. The evolution of obligate
polyandry or polygyny, with as key criterion that mature colonies are never monandrous or
monogynous anymore (hence an irreversible transition to an evolutionary derived social
system via either of the two factors that significantly increase colony-level genetic diversity.
In this context, facultative polyandry/polygyny hardly makes a difference in average colony-
level genetic diversity and seems very unlikely to have a connection with disease defense
because that should have induced further selection for obligate polyandry/polygyny – we do
not know of any such facultative to obligate transition).

21

ess
is

131 In the interest of maximizing comparability with previous studies and validating your overall
conclusions, we would like to see you repeat your analyses with these natural kind
categorizations. We know that some of your analyses cannot handle 3 categories, but
facultative and obligate polyandry evolved independently and almost certainly for different
reasons (see Boomsma, 2013; Beyond promiscuity ... Phil Trans). We strongly suspect this is
also true for facultative and obligate polygyny because there are multiple examples (the fire
ant is one of them) where the same ant species has populations with strict monogyny next to
populations with obligate polygyny and without intermediates. Following this line of thought,
your contrasts in the analyses that require binary traits would then be: single vs facultative
and single vs obligate, and if the first of these contrasts does not deliver (likely, because
facultative remains reversible) then also single+facultative vs obligate. Your way of
analyzing mating frequency data may not be too far from the latter contrast, but the way in
which you have categorized polygyny (lines 399-402) seems to effectively blur any effective
contrast – and this is the variable that never gives anything in path analysis. We thus believe
that a validation of this kind is necessary to make your conclusions robust beyond reasonable
doubt. If these validations confirm that neither obligate polygyny nor obligate polyandry have
ever been a driving causal force that would be great but, as already indicated, your current
way of handling these variables (particularly polygyny) may have removed contrasts that
might have made it too easy for colony size to be the only causal variable.

As requested, we reanalyse our data, wherever possible, using the method suggested by the
reviewer to classify species into the following categories: monandry/facultative
polyandry/obligate polyandry + monogyny/ facultative monogyny/obligate polygyny. We
also use their suggested method for analyses that require the use of binary variables.

While we recognise the value in repeating our analyses using the method suggested by the
reviewer, we still feel that it is important to analyse our data in its untransformed, continuous
form, which the data was originally collected in. This approach allows us to analyse the data
with the maximum amount of resolution available and try to prevent any loss due to it being
transformed into a categorical variable with three levels. We also feel that analysing the data
in a continuous form will best represent any changes in colony-level relatedness that result
from variation in queen mating frequency or number of queens per colony. Therefore, we
include in the manuscript, results from analyses where queen mating frequency and queen
number are analysed first as continuous variables, and then as categorical variables
(reviewer's suggested method). This should improve the manuscript as it stands as a good test
of how robust our results are to different classification systems. We also retain an analysis
where queen mating frequency is transformed into a binary variable based on a threshold
value of 2, as this is the predicted number of mates at which worker policing is predicted to
evolve.

While we endeavoured to repeat all analyses using the reviewer's classification method for
mating frequency and queen number, it is not possible to analyse queen mating frequency as
a categorical/binary variable for the method used in our ancestral state reconstruction causal
analysis. This is because this method directly compares estimates of queen mating frequency
(continuous trait) at nodes in the phylogeny where transitions to multiple castes occur vs

22

ess
is

175 nodes where a single caste is retained. Therefore, a continuous estimate for the number of
176 times that a queen mates is required to compare which value is larger, or whether values are
177 not significantly different. This approach would not be possible if queen mating frequency
were analysed as a categorical or binary variable.

We agree with the reviewer that the threshold used to transform colony size into a binary
variable for some of our analyses is somewhat arbitrary. Therefore, we perform a sensitivity
analysis to check whether our results are robust to using different thresholds to classify
species as possessing small or large colony sizes.

In addition, the analyses including caste number seem not to distinguish between
monomorphic workers and continuous worker polymorphism (variation in body size with
allometric scaling within a monomorphic worker caste). This is addressed later by analyzing
worker size variation separately, but that dataset only covers 117 species, a huge reduction
compared to your discrete caste number analysis (608 species). It is unclear to us whether
these two analyses were really nested (in the sense of addressing continuous size variation
within each of the 1-4 discrete castes), and if it was could it then not have been done in a
single analysis with two levels of variation?

We agree with the reviewer that the analysis of caste number is not able to distinguish
between species with monomorphic workers and species with continuous worker
polymorphism. However, we believe that it's best not to combine a 'continuous worker
polymorphism' category into our measure of discrete worker castes, as this variable is
naturally 'count' data and should not be combined with categorical data. Considering this
limitation, we perform a separate analysis of worker size variation which should capture
worker caste variation across species, including those which have continuous worker
polymorphism. However, it's important to maintain our analysis of the number of discrete
worker castes in the manuscript as a larger amount of data is available for this variable.

Minor comments:

1. Please insert a few explicit lines saying that your study confirmed that monogyny and
monandry were the (monophyletic) ancestral states when the ant MTE evolved. I cannot
imagine that your analyses would fail to show this with high confidence. Such an update
conclusion, relative to Hughes et al. (2008), would be really valuable because his analyses
used broad-brush eusociality language and did therefore not specifically address the origins
of the MTEs to organismal colonies. It somehow seems odd that you have done this for
establishing that a single worker caste is ancestral (line 681) but not for a single queen or
queen-mate being ancestral.

The aim of our study is to identify the variables associated with, and potentially responsible
for determining the number of worker castes. Therefore, life history data for species that are
not informative for testing this hypothesis are not in the analysis. For example, those species
that possess gamergates or those with clonal reproduction. As a result, there is not sufficient
data among the species being analysed to robustly test whether monogyny and monandry

23

ess

is

219 were the ancestral states when the ant MTE evolved. This limitation stems from the
220 significant number of species excluded from our analysis from basal ant subfamilies (e.g.
Ponerinae), primarily due to the presence of gamergates.

2. Line 23. Consider supplementing this sentence with something like: ‘support for the size-
complexity hypothesis in a monophyletic clade of social insects that emanated from a single
major transition to permanently caste-differentiated colonies.’
To address this concern, we make the point that ants are a monophyletic clade in the 5th
paragraph of the introduction.

3. Line 26. ‘facilitating evolutionary transitions’ is rather ‘sitting on the fence’. Some DoL
may evolve in open societies of cells or multicellular individuals that will then change social
interactions (e.g. yeasts), but as long as these tendencies are variable and based on condition-
dependent cooperation such developments remain potentially reversible and will not induce
transitions comparable to neuter-caste colonies and somatic cell differentiation in metazoan
bodies. Note that condition-dependent (coming together) sociality does not preclude the
evolution of some cell-type polymorphism (see Fisher, Shik & Boomsma, 2020 for a
comparative analysis). The key point appears to be that not all cells are irreversibly
committed to being either soma or germline as is the norm in the bilaterian Metazoa.
We feel that the current phrasing best-captures our point while keeping the opening sentence
of the first paragraph simple, which is important for clarity.

4. Line 56. Consider skipping ref 23, which had an irritating error for Sericomymex – see
our separate file on data set assessments (ref 23 is also mentioned on line 303)
Citation removed.

5. Line 74. Behavioral castes is a contradiction in terms that we owe to E.O. Wilson’s
tendency to capture everything in smooth ‘eusociality’-gradients. His behavioral castes are
nothing but adult reproductive role differentiation (i.e. reproductive skew). There is no
relationship between adult reproductive skew and pre-imaginal caste differentiation (the
original concept of caste in insects), i.e. with the key-trait that convergently evolved in the
(super)organismal lineages only.
‘Behavioural castes’ is removed.

6. Line 101. The median value of 1.1 inseminations per queen underlines how arbitrary cut-
offs can be. You obviously did not chose this median value, but used > 2 and ≤ 2 (line 199).
However, this remains arbitrary as well, because far from all (in fact almost no) ants with
polyandry > 2 evolved worker policing in the way known from honeybees (one of your
arguments on line 524). Nonetheless, we believe that using this > 2 cut-off may well be
highly correlated with the absence of mature colonies with singly-inseminated queens, but
formal validation of this suspicion seems to be called for.
As mentioned above, we repeat our analyses using the suggested categories to assess the
robustness of our results: monandry/facultative polyandry/obligate polyandry + monogyny/
facultative monogyny/obligate polygyny. For the appropriate analyses, we also retain our

original method of transforming queen mating frequency into a binary variable based on a
threshold value of 2, because worker policing has not been studied in a large proportion of
ant species, and 2 mates is the predicted value at which selection would favour it to evolve. In
monogynous colonies, 2 mates is also the value at which relatedness among workers in the
colony is midway between the maximal value of 0.75 and the minimal value of 0.25.

7. The tree of Figure 2 is very nice but could the bars in the four shades of blue be made to
stand out in a more pronounced way?

We have made the shades of blue more contrasting and the highlighted transition points
larger to aid interpretation.

8. Line 146. Make explicit that the relationship between polygyny and polyandry was
expected and observed to be negative.

Amended so that the relationship is explicitly stated to be negative, and we highlight that this
negative result has been observed previously.

9. Line 184. Given there were only three further MCC trees, consider giving all four rather
than just one path-diagram to illustrate in the main paper how coherent these alternative
results were? See also lines 618-619.

As pointed out by the reviewer, 3 out of the 4 trees give the same result. For the 4th tree, we
still obtain similar results to the other trees. This result being that: 1. the positive effect of
colony size on the number of worker castes is much larger than any of the other relationships
that stem to/from the number of worker castes; 2. The strength of the relationship between
mating frequency and number of castes is negligibly small and confidence interval overlaps
with zero. Therefore, given the similarity of the result for the one tree that differs, we feel
that the manuscript is best served by presenting the results from all four trees in the
supplementary material, as not much additional information is given by presenting them all in
the main text.

10. Line 197. In contrast to polyandry and polygyny, a colony-size cut-off between small and
large will always be arbitrary because colony size remains an essentially continuous variable.

We note that you have tried to validate this cutoff by juggling with different quantile
boundaries (line 528). After considerable deliberation, the GAGA project ended up using the
categorizations ‘small (0-100)’, ‘medium (100-1000 workers)’, large (>1000 workers), with
unicoloniality as an additional separate category. In contrast to polyandry and polygyny,
there is no way of saying a-priori that one kind of partitioning colony size variation is better
than another, so this remains an empirical issue. However, because the causal effect of colony
size is the main result of your study, we recommend that you apply a range of small-large
boundary values to see whether polygyny and polyandry remain insignificant across this
range while colony size always remains causal.

25

We agree with the reviewer that the cut-off values used to transform colony size into a binary
variable are arbitrary. This is why we perform a sensitivity analysis applying different cut-off
thresholds when colony size is analysed as a binary variable in the transition rate analysis. In
all other analyses, colony size is allowed to remain as a continuous variable, avoiding this

ess

is

307 issue. In analyses that incorporate queen mating frequency, number of queens per colony and
308 colony size simultaneously (path analysis), colony size can be analysed as a continuous
variable. Therefore, a range of binary cut-offs is not required to test whether the number of
queens or mating frequency have a significant effect on the evolution of the number of castes
across a range of colony size thresholds.

11. Line 207. Not clear why polyandry and number of worker castes should be negatively
related. Seems to contradict what you have on line 229.

The negative value obtained for the Bayes Factor (BF) does not indicate a negative
relationship between queen mating frequency and the number of worker castes. Instead, it
indicates that a model where queen mating frequency and the number of worker castes evolve
independently of one another performs better than a model where the evolution of each of the
two variables depends on the state of the other variable (i.e. correlated evolution). I have
amended the manuscript to clarify this issue.

12. Line 247-250. I remember that Francis Ratnieks always argued that task-partitioning (TP)
was something fundamentally different than division of labour (DoL), which may mean that
TP cannot be ancestral to caste-based DoL, as you implicitly suggest here. Perhaps check
those original pioneering papers on TP if you feel strongly about making this point.

Thank you for bringing this to our attention. We mistakenly referred to 'size-based task
partitioning' when discussing size-based division of labour in species with a single discrete
worker caste. We have amended the manuscript accordingly.

13. Line 258-260. With this 2 versus 1 result, is it then reasonable to conclude that all you
can say is that things are correlated and that you were unable to assess causality?

The 2 versus 1 outcome identifies that there is greater support for the hypothesis suggesting
colony size is the primary causal factor driving the evolution of worker size variation.

However, we acknowledge the need to highlight the uncertainty associated with this finding
and we revise the phrasing in the manuscript.

14. Line 309-310. Write caste phenotypes rather than just types. We do not think the
microbial references given here (11, 39) are directly relevant for DoL in organismal ant
colonies. A better analogy would be something like bodily versus colonial homeostasis – a
term that has gone a bit out of fashion but we cannot think of a better one.

We have amended 'types' to 'caste phenotypes' in the manuscript. However, we still feel that
references 11 and 39 are appropriate. While these papers do not explicitly model
multicellular organisms or superorganisms, they still demonstrate that larger groups are better
able to maintain the optimal ratio of phenotypes performing each task – this should apply
regardless of whether a major transition has occurred or not.

15. Line 321. See my earlier comments on the likely non-relationship between TP and DoL

We have amended as mentioned above.

26

ess

is

16. 'influenced'? passive voice

All appropriate instances changed to the passive voice, excluding the section where the
alternative causal models are proposed for the path analysis where describing the models is
clearer when using the active voice.

17. 333-336. Your references about disease susceptibility here are a mixed bag of grooming-
like defences in open societies of primates and much more deeply coordinated behaviors in
organismal colonies of honeybees. These two are essentially incomparable, as was recently
compellingly argued by Sylvia Cremer (see Figure 6.5 in reference 1 and the related text
sections). Also, here there is no such thing as a single group-size gradient. Neither can you
take for granted that disease pressure increases proportionally with colony size across the
superorganismal lineages. It makes a huge difference for disease pressure whether foragers
have wings and operate in overlapping 3D territories (like honeybees) or are wingless so they
operate in non-overlapping 2D territories (like ants).

We have removed references that mix defences found in open societies and coordinated
behaviours found in superorganisms. We rephrase the paragraph suggesting only that disease
pressure might increase with colony size. We do not think that our newly edited paragraph
suggests that comparisons should be made across superorganismal lineages regarding colony
size and parasite pressure.

18. Line 346. The late Christian Peeters offered a compelling argument for winglessness
having released the worker caste of ants from a number of body-size constraints that
continued to apply in bees and wasps where workers have universally retained wings
(Peeters, 2012). In that light the difference that you highlight here may not be that surprising.
We amend the manuscript to include the Molet, Wheeler & Peeters, 2012 citation and to
include the hypothesis that they propose.

19. Line 359. You mean multicellular organisms here I think, not societies. The difference
between queens and workers does not seem relevant because your paper is about
differentiation of the metazoan soma-analogues (workers/soldiers) not about reproductive
DoL (as you confirm a few lines further down).

Thank you, we have amended societies to organism. However, we do feel that queen-worker
dimorphism (another form of division of labour) is a relevant trait that future studies could
focus on, as the size-complexity hypothesis predicts that queens and workers will specialise
in their roles as colony size increases. Therefore, we might expect greater queen-worker
dimorphism in ant species that have larger colony sizes.

20. Line 364. Make explicit that you did a large-scale phylogenetic analysis.

We have amended this.

21. Line 379. Write 'queens per colony', also in other places, just to be precise.

We have amended this.

27

ess

is

394 22. Line 394. Also cite ref 24 here. It was the first to make the point that number of
395 copulations are rather irrelevant. In many ways that early review carved out the essential
predictor variables (polyandry, polygyny, colony size) about whose causal significance your
paper tries to offer the definite answer.

We have amended this.

23. Line 453. Add here something like: ‘social parasites, usually lacking some or all of the
worker castes’. Write and/or at the end of the line?

We have amended this.

24. Line 457. Your arguments for excluding these cases are compelling, but make more
explicit that all these cases are evolutionarily derived elaborations of the ancestral full-
sibling-colony state and that it is the ensuing secondary reductions of complexity in social
organization that prompted you to exclude these lineages.

We have amended this.

25. Line 546 (also other places). Can you define somewhere what you mean when you write
‘qualitatively similar’? We assume it must be something like ‘Similar enough not to lead to
different conclusions because direction and magnitude of (partial) correlations remained
essentially the same and the (non)causative position of covariables did not change either.’?

We clarify the language used to present our finding that the conclusions drawn across trees
remained unchanged due to the similarity in the presence, direction and magnitude of the
causal relationships identified.

26. Line 571. Ref 24 was the first to make this point, but the first to suggest that colony size
should select for multiple queen mating was Blaine Cole (we believe 1983 – cited in the
other reviews)

We have included Ref 24. We include references for Loope et al., 2014 and Schmid-Hempel,
1998 as these relate to the relationship between colony size, mating frequency and pathogen
pressure which is the hypothesis that is discussed, while Cole, 1983 refers to sperm limitation
instead.

27. Line 572-574. See our earlier comment on disease pressure.

We remove references that mix defences found in open societies and coordinated behaviours
found in superorganisms as was requested previously. However, we feel that the language
used for lines 572-574 suggests only that disease pressure might increase with colony size
rather than taking this for granted as was mentioned in the reviewer’s previous comment.

28. Line 576. See Boomsma (2013) Beyond promiscuity ... (Phil Trans) for arguments why
obligate and facultative polyandry almost certainly have different evolutionary reasons.

We recognise the point made by the reviewer, and as mentioned above, we reanalyse our data
with queen mating frequency as a categorical variable (monandrous/facultative
polyandrous/obligate polyandrous) wherever possible. However, we feel that it’s still
important to include analyses in the manuscript where queen mating frequency is a

28

ess
is

438 continuous variable. In our dataset, there is variation in effective queen mating frequency in
species classified as facultatively polyandrous (range = 1.04-3.4; mean = 1.5, species = 33),
indicating that mating frequency could have a considerable effect on genetic diversity and
intra-colony relatedness in facultatively polyandrous species. Therefore, it's still possible that
facultative polyandry could evolve for its influence on genetic diversity, and effective queen
mating frequency should also be analysed as a continuous variable in our manuscript.

29. Line 629. 'influenced' (past tense)

We have amended this.

See also our separately uploaded file: Assessment of the dataset of Bell-Roberts.docx

Copenhagen, 22 November 2023 Koos Boomsma and Joel Vizúeta

Reviewer #2 (Remarks to the Author):

With interest have I read the manuscript by Bell-Roberts et al., which investigates the size-
complexity hypothesis in ants. They have performed various phylogenetic comparative
analyses to relate the colony size of ant species ("size") with the number of worker castes
("complexity"). They find, like others before them, that species with larger colonies tend to
have more worker castes. They add to this finding by excluding alternative explanations of
this correlation related the number and mating behavior of queens. I found the paper to be
clear and easy to read, and the analytic approach to be sound. The three separate approaches
to disentangle the order and causality of effects are a strength of the work. Overall, I only
have relatively minor comments. Nonetheless, the presentation of results can be improved in
several places, which I outline below.

Thanks! We're glad that you found our paper interesting.

A general weak point of the presentation of this work is the treatment of effect size and
uncertainty. The text should do a much better job of interpreting the effect sizes (and their
credible intervals), instead of the binary significant/non-significant language currently used.
This is especially prudent given the wide range of sample sizes used in different analyses. In
addition, the figures do not give the reader any idea of the uncertainty of the estimates, even
though the methods employed do give the authors the opportunity to do so. In particular,
figures 3, 4 and 6 would be more informative with the inclusion of credible/confidence
intervals or standard errors.

We amend figures 3-6 to include either SE, confidence intervals or credible intervals,
excluding Figure 6a, where credible intervals were largely overlapping for the two regression
lines and made the figure messy and potentially confusing.

We have improved our manuscript by incorporating more extensive discussion on the effect
sizes depicted in our key findings. This includes: 1) calculating R^2 values for our regression

29

ess
is

482 analyses that use Gaussian error distributions; 2) a more detailed discussion of our ancestral
state reconstruction causality analysis; and 3) greater detail regarding the magnitude of path
coefficients in relation to confidence intervals, especially in instances where uncertainty is
more pronounced.

The introduction starts very broad, but then quickly seems to only consider multi-cellularity
and social insects. In particular, the use physical castes stands out to me. I don't follow why
mutual dependence requires differences in size and morphology (lines 74:76). Don't e.g.
meerkats, Seychelles warblers etc. have mutual dependence? I don't recall that they have
physical castes. Given that this is the only argument provided for using physical castes, and
no reference is provided, this warrants a better explanation.

We have amended the manuscript to explicitly focus on lineages that have undergone major
evolutionary transitions in individuality (multicellularity and superorganismality) because
suitable comparisons can only be made between groups that have experienced these
transitions. This is because selection can operate at a higher level, allowing the evolution of
group-level adaptations. This is in contrast to species that form social groups (e.g. meerkats)
that have not experienced a major transition.

In our analysis, we focus on physical castes, as opposed to behavioural differentiation among
workers, because we are interested in the evolution of complex life involving extreme
specialisation (outlined in the abstract and 1st paragraph). While it's true that species such as
meerkats are obligately social and rely on a cooperative group for reproduction, roles within
the group are behavioural and can change over the course of a life. Therefore, extreme
specialisation has not evolved. Physical castes in ants permit extreme specialisation, are
determined during development and fixed for life, and involve mutual dependence.

To reflect the points in our response, we have amended the manuscript so that: 1.
comparisons are only made between multicellular organisms and social insect
superorganisms which have both undergone major transitions. This avoids potential
comparisons between cooperative breeders and superorganisms; 2. For lines 74-76, we edit to
make the point that physical castes allow extreme specialisation and are fixed for life, as
opposed to behavioural role differentiation.

The authors exclude a lot of species based on several criteria. As I'm not an ant expert, I can't
evaluate these criteria very well, but it is important that exclusion criteria are mentioned in
the main text, as it should be apparent to which situations these results may be expected to
generalize.

We provide further information detailing why these species are excluded in the main text
which should clarify which situations our results may generalise.

Line-by-line comments:

Title: The title would be more informative if it would make clear how the paper advances on
the other papers that have tested the size-complexity hypothesis in ants.

30

ess
is

526 In light of this point, we alter the title to ‘Larger colony sizes favoured more worker castes in
ants.’ The new title details our main finding, indicating how our paper advances the field in
relation to previous work.

19: Reporting 758 species here is technically accurate, but rather misleading. It suggests a
level of power and generality that isn't borne out in the manuscript because of the large
amount of missing data. I did not find a single analysis that reaches that sample size.
We recognise that data is not complete for all traits for all species in our analysis. Therefore,
in the first paragraph of the results section we highlight that: 1) we used the maximum
amount of data available for each analysis; 2) sample sizes will vary between analyses
depending on the variables being analysed; 3) we detail the number of species used in each
analysis.

85: cite reference 75 here, or the original paper in Evolution.

We have amended this.

183: MCC stands for maximum clade credibility (no extra C for consensus).

We have amended this.

253-266: This section overstates the support for the size-complexity hypothesis. The path
analysis is suggestive but ambiguous. There is some support for both directions of effect
between colony size and variation in worker size. The alternative hypothesis of queen mating
frequency causing variation in worker size, in turn causing colony size should not be
excluded. Again, a discussion of the relevant effect sizes and uncertainties would greatly help
the reader here. It may be possible to achieve more power for this analysis by using
phylogenetic trait imputation, recently made available in Thorson & Van der Bijl 2023, JEB
36 (10): 1357-1364.

We amend this section to not overstate the support for the size-complexity hypothesis,
highlighting that contrasting results are found across different models.

As recommended, we also include discussion of the relationship between mating frequency
and variation in worker size, the effect size and the confidence interval. While one supported
model includes a relationship between queen mating frequency and variation in worker size,
the magnitude of the path coefficient overlaps with zero (detailed in Fig. 6 and Supp Table
5). Taken together, these results suggest that queen mating frequency is not influencing
variation in worker size which we mention within the results section.

While having missing data can introduce bias into an analysis. We are concerned that trait
imputation could potentially introduce greater error into the dataset being analysed and
therefore the results: 1. Despite having considerable sample sizes for our analyses, there is
still a large amount of missing data as a consequence of the very large number of ant species
that exist. Therefore, it may be difficult to draw conclusions from analysing datasets with
large amounts of imputed data. 2. It is likely that data is often not missing at random. For

31

ess
is

570 example, there is a large geographical sampling bias for ant species that have been studied,
therefore, species in certain geographical locations may be more likely to have complete data
than in other regions. Furthermore, differences in life history traits, such as colony size, may
also mean that some species are more likely to have data collected on them than others.

449: The range of 1-31 is odd, since you can't calculate the standard deviation of a single data
point.

This was an error, thank you for spotting.

498: This is phrased ambiguously, since the fraction of iterations where one level is greater
than OR less than the other level is always 1.

We have clarified this.

515:552 While I can see why this is listed under the correlational analyses, it is confusing
since this doesn't match the presentation in the main text, where only the BPMMs are
presented in that section.

We combine both sections and move them to the 'Causality analyses' methods section.

Tables S2 and S3: please report all models, not just the ones within 2 CICc.

We have amended.

I encourage the authors to present their supplemental tables in a more accessible format than
the proprietary xlsx.

xlsx format is needed to have multiple sheets present in the supplementary table file.

However, csv files are made available for our data which is also presented in the
supplementary tables.

Reviewer #3 (Remarks to the Author):

Key results

The MS tested in ants whether increasing colony size is correlated and has a causal
relationship to a stronger division of labor among workers. They quantified the division of
labor in two ways: (1) physical worker caste and (2) variation in worker size. Both (1) and (2)
are found to correlate with colony size in phylogenetically controlled regression.

Phylogenetic path analysis that tested different causal directions between colony size and the
division of labor also resulted in a consistent best support model where colony size causally
affects the division of labor using both (1) and (2). Using (2) revealed additional causal
relationships that were not fully explored. Using only (1), the authors also use transition rate
analysis to show that transitions from 1 to >1 physical castes only occurs in species with
large (>300) colony sizes. The analysis was robust to using alternative ways to categorize
large/small colony sizes. Finally, analyzing the reconstructed ancestral states also shows that
ancestors of species with multiple castes had larger colony sizes compared to ancestors of

32

ess

is

614 species with only one caste.

Validity

The data collection and analysis were done with care and using the current best methods.

However, additional analyses could be done to improve the breadth of the conclusions.

Originality and significance

The study is original and gets at the problem in previous studies that are mostly based on the

correlation between colony size and the number of worker castes. The study is significant for

both the study of animal social evolution and to evolutionary biology in general because the

use of path analysis and transition modeling can reveal the direction of casualty in evolution,

which has been a limit of a field that is mostly based on correlation analysis and which

experimental validation is almost impossible in most taxa.

Data & methodology

I have two concerns with how data are coded/analyzed.

First, I'm most uncomfortable with coding the number of queens as binary (one vs. >1 queen)

(line 135). While the authors have provided some justifications for doing so (line 405), I

don't see why they must code the number of queens as binary, and that they absolutely

cannot use the reported values of the number of queens in the analysis. See * below for more

details. An additional analysis that treats the number of queens as a continuous variable will

help to resolve whether the lack of relationship in Fig 3e is due to losing power in doing an

ANOVA vs. a regression analysis.

We recognise this point and we have repeated our analyses using the number of queens per

colony as a continuous variable, which was the form that the data was originally collected in.

In addition, we perform analyses where queen number is categorised as a discrete categorical

variable (obligately monogynous/facultatively polygynous/obligately polygynous), rather

than a binary variable, as this was requested by another reviewer. Obligately

monogynous/facultatively polygynous/obligately polygynous appear to be discrete

reproductive strategies which may evolve for different reasons, which is one reason why we

also include these analyses. Another reason is that previous studies have also used this

classification with 3 categories, and analysing queen number as a categorical variable with 3

levels will maximise comparability between these studies.

Second, the using of two ways (no. of physical worker caste and variation in worker size) to

quantify worker division of labor is a valuable aspect of this study. However, some of the

important conclusions (see ^ below) will be better supported if additional analysis is done

using variation in worker size with ONLY species that have a single worker caste. An

ancestral state reconstruction should also be done using the variation in worker size. This

could hopefully also resolve the presence of additional causal relationships in the path

analysis with variation in worker size.

We feel that an ancestral state reconstruction may not be an appropriate analysis in this

particular case. This is because this method directly compares estimates of colony size or

queen mating frequency (i.e. a continuous trait) at nodes in the phylogeny where transitions

33

ess

is

658 occur in a discrete categorical trait. For example, species that either have a single worker
caste or multiple worker castes. Therefore, to test whether colony size is greater when
transitions occur in variation in worker size, variation in worker size would need to be
transformed into a discrete variable.

One of the main benefits of analysing species based on variation in worker size is that it is a
continuous variable, where its measurement has no subjective interpretation. This is in
contrast to classifying species based on the number of physical worker castes, where
continuous variation in the workers of a colony can sometimes mean that the number of
worker castes estimated for a species is subjective. However, the benefit of classifying
species based on discrete worker castes is that a lot more data is available. Overall, we feel it
is best to avoid transforming variation in worker size into a categorical variable for analyses,
and instead use number of discrete worker castes, for all analyses that require discrete
variables.

We acknowledge that we transform colony size, which is a continuous variable, into a
discrete variable for some analyses in the manuscript. However, we feel this decision was
justified as we had no alternative option, or alternative variable that could be used to quantify
colony size as a discrete trait. This is not the case when measuring non-reproductive division
of labour in ants where two variables exist – one as a continuous variable, and the other as a
discrete variable.

More detailed comments are below:

Line 36: It'll be helpful to provide some context for the readers on why it's the leading
explanation. The later part of the paragraph cites a few studies in cell and insect. But how do
these studies make the size-complexity the leading explanation?

We have amended to provide greater explanation regarding why it is a leading explanation.

We also rephrase to clarify that it is not the only explanation, but 'one of' the leading
explanations. We feel that the citations in the manuscript demonstrating its empirical support
across different levels of biological organisation support our claim.

Line 52: A single queen can also mate multiple times, so should "queen" be "queen(s)"?

We have amended.

Line 55-56: Please be specific. Is the number of queens also correlated with group size?

We have amended, as a correlation was not expected between number of queens and colony
size.

Line 65-67: Can you rephrase this sentence without all the "one /another variables"? It
doesn't help with reading.

We have rephrased accordingly.

Line 68-69: I'd say "this pattern leads to the false impression that larger group directly favors
increased division of labor and misses the involvement of an intermediate factor."

34

ess

is

702 We have rephrased accordingly.

* Line 135: Please justify why the number of queens is reported (Fig 2) and analyzed (Fig 3c)
as a discrete variable and not a continuous variable. This is very important since you removed
the number of queens in the causality analyses based on not finding a significant relationship
between the number of queens and other variables. What if you found a significant
relationship when a continuous variable of queen number is used?

We have reanalysed our data, treating number of queens per colony as a continuous variable,
to address this concern.

* Line 405: I see the argument that the observed number of queens could be an overestimate
of the effective number of queens. But then why can't the idea of a harmonic mean be applied
here as you have for queen mating frequency? And while it may be an overestimate, why is
coding it as binary better than keeping it as a continuous variable? I'm worried that when
using the binary variable, the analysis may lose power to detect what could be in there.

As mentioned above, we recognise your concern, and reanalyse our data using the observed
number of queens per colony as a continuous variable. We use the arithmetic mean for the
number of reproductive queens per colony as this is often the only data that is available in the
literature.

Line 169: This section appears to be overly brief in comparison to other pieces in the result
section - although the result is very important. The competing models and the verdicts of
testing only 4 models should be explained here instead of in the method section.

As requested, we elaborate further on the results and the unsupported models. However,
restricting the number of points made here emphasises the strong result that supports the size
complexity hypothesis, allowing it to stand out. Describing each of the alternative proposed
causal models verbally takes up a lot of space in the text which is why we feel that it is best
left in the methods section. We also feel that the alternative causal models are best described
visually, and we have a figure that does this in the supplementary material. There is a link to
this figure in the results section.

Line 175: Please at least report the omega values here to show how the best model compares
with the competing models.

We amend to include this.

737 ^ Line 238, 313-321: This is a great additional analysis, but I wonder whether the result is
738 still mainly driven by the drastic CV in species with different physical worker castes. The
739 present analysis only supports the statement in line 315-317 "These results indicate that our
findings are robust to analysing division of labour as either the number of physical worker
castes or as variation in worker size". However, the analysis is not enough to conclude that
"This applies not only to species with different numbers of physical worker castes, but also to
species that have just a single worker caste" (314-315) nor " Our results also suggest that the
positive correlation between colony size and variation in worker size is not just the result of
variation across species with different numbers of physical worker castes." (317-318). To

35

ess
is

746 support these statements, additional analyses with only species that have a single physical
worker caste should be done. And if the pattern still holds, then it will directly support the
statements above - which is very important in understanding how colony size affects the
division of labor.

Since analysing number of queens as a continuous variable, and including additional data
provided by one of the reviewers in the analysis, we have a new result when analysing only
species with a single worker caste. In species with a single worker caste, we find support for
variation in worker size evolving before larger colony size. We think that this change in
result is likely driven by the change in the association between queen number and the other
traits in the analysis as it is now being analysed as a continuous variable. The additional data
could also be influencing the result. Therefore, we remove the conclusions highlighted by the
reviewer (lines: 314-315 & 317-318) as they are no longer relevant to our result. However,
our result that large colony size is important for the evolution of more extreme forms of
variation in worker size/caste number still stands.

Line 275: Can you explain why that when analyzing the variation in worker size, there are
two additional arrows (variation in worker size  colony size & queen mating frequency
variation in worker size). Analytically, I understand that it's coming from the 2nd best-
supported model, but in terms of the evolution of worker caste, what does it mean? An
ancestral state analysis is not used for the variation of worker size, but it may allow you to
tease out what happened. Perhaps in parts of the phylogeny, you see some support for the
second-best model. And it may even be related to only those species that have a single
physical caste. Rather than opening that for interpretation by readers, I believe it will improve
the paper to dig in more about why additional, despite weaker, causal relationships exist
when a different way of quantifying caste is used.

In the average model, the magnitude of the path coefficient for the effect of queen mating
frequency on variation in worker size is very small and the confidence interval overlaps with
zero. This is demonstrated in Fig. 6. When individually examining model 3, where queen
mating frequency influences variation in worker size, the magnitude of the path coefficient is
also low (0.08), and the confidence interval overlaps with zero. In the results section, we link
to the supplementary tables to present these statistics. Overall, the impact of mating
frequency on the evolution of variation in worker size is small when compared to the
influence of colony size. Although our analysis identifies a causal pathway, it suggests that
mating frequency may not significantly contribute to the observed variation in worker size.
We made edits to ensure that the manuscript reflects this conclusion and our reasons for it.

Regarding the positive effect of variation in worker size on colony size, while there is some
uncertainty in the direction of the relationship, overall, there is greater support for colony size
affecting variation in worker size. In the manuscript, we highlight the greater support for
colony size affecting variation in worker size but also discuss the uncertainty.

However, for the reasons explained above, we feel that it's not appropriate to transform
variation in worker size into a categorical binary variable so that we can apply the same
causal analyses that we have applied to the discrete caste number variable. The discrete caste
number variable is better suited to these types of analyses (ancestral state and transition rate)

36

ess
is

790 and analysing variation in worker size with path analysis is an effective test of the robustness
of our results.

Line 321: Any citations to size-based task partitioning?

This line no longer exists in the paper, and therefore a citation is not necessary.

Line 326: " the number of queens " is really single vs. multiple queen according to the
analysis.

We now analyse number of queens per colony as a continuous variable.

Line 409-427: Since two methods were used to determine the number of worker castes
(morphometric & literature search), did you do a data validation using species with
morphometric to test whether the literature search yielded the same result as the
morphometric classification?

When morphometric data was available alongside direct estimates of the number of castes by
other authors, there were only a very limited number of instances where these estimates
differed. In these cases, using our own estimates based on morphometric plots should take
precedence. This is because it allows us to estimate the number of castes based on our
definition, which is best suited for the purpose of testing the questions proposed in this
manuscript. However, when morphometric data is not available, using estimates for the
number of worker castes provided by other authors is necessary to increase our sample size as
this is the largest source of data that is available.

Line 496: I don't see any reports of p values for BPMMs in the result section.

Please see the section 'Ancestral state reconstruction' in the 'Causality analyses' of the
results section where p values are present.

Line 556: The result section reported the path analysis first. Please move this part of the
method before the BayesTrait.

Manuscript edited so that causality analyses in the methods section are now in the order: 1.

Path analysis; 2. Bayestraits; 3. Ancestral state reconstruction. This matches the results
section.

Line 563: (ii) is your model four in Extended Data Fig. 1. Please fix this.

Thank you for spotting, we have amended.

Line 606: This part is missing a subtitle.

We have added a subtitle.

Assessment of the dataset of Bell-Roberts et al.

The dataset is among the largest ever analyzed for ants, which is a major strength. However, there is a several-decades long tradition of comparative data analysis in ants, so it is important that the variables used stay as comparable as possible, both backward in history and laterally, relative to currently ongoing work using the same or similar data. The GAGA project <https://db.cngb.org/antbase/project> is such an ongoing program, the main results of which are expected to be submitted for publication in spring 2024. We therefore compared the dataset used in the present study with the GAGA data to test the similarity of the two independent surveys, as a validation of both our own and the Oxford-group efforts.

We believe that reaching maximal correspondence between the two types of analysis is important because: 1. The B-R study will stand for decades as the last word on the main selection forces that have shaped adaptive radiation in the ants. 2. We imagine that the GAGA papers expected in 2024 will have similar impact, albeit not quite as long lasting given how fast new and better genomes will increasingly become available. GAGA will produce a considerable number of convergent signatures of selection (or lack thereof) emanating from variation in colony size, caste number and worker size variation, polyandry and polygyny. The field will therefore be well-served if, when we refer to your present study, there is no doubt that we made maximal efforts to work with the same data matrix of observed variation in these key social variables.

Prelims

The B-R study includes 62 of the 163 species for which the GAGA project obtained high-quality (chromosome-level for 16 species) genome sequences. The overlap would have been higher had the B-R study not excluded (for valid reasons) about 20 GAGA species because they have gamergates, clonal reproduction or supercolonies (supp. table 14). Overall, the B-R study covers ca. five times as many species as GAGA, and relied completely on literature data while GAGA also had direct information available from collectors. This implies that for 33 out of the 62 overlapping species GAGA has entries that the B-R study lacks and which we are happy to make available in exchange for an acknowledgement if B-R et al. are interested to include them (see list 2 below). In addition, we noticed that some data available from the Romiguier et al. 2022 paper in *Current Biology* - colony sizes and worker polymorphism records for 65 ant species - are missing in the B-R dataset. We therefore recommend you have a look also at the Romiguier data to see whether they could provide some additional entries for species that GAGA did not cover.

As far as I can tell, Romiguier et al., 2022 does not publish the sources that they have obtained their data from, nor whether they are genus or species-level estimates for each given species. The colony size estimates are also only provided on a \log_{10} scale and their measurement of non-reproductive division of labour does not distinguish between the number of discrete worker castes. Therefore, due to the uncertainty associated with this data, I have not included the small number of additional species present in Romiguier et al., 2022 that are absent from our database.

Regarding the data entries that we both have, most of them are the same, which is reassuring for both analyses, but we found 7 cases with discrepancies in several variables (see list 1 below). Six of these seem incorrect in the B-R data base, while we concluded that one (*Messor barbarus*) was wrong (for the number of worker castes) in our data table. In another case (*Myrmica scabrinodis*) we had missed facultative polyandry.

Categorization within predictor and response variables:

- For colony size, B-R used the average number of workers in a mature colony while we used both average and maximum number of workers, as well as logarithmic range-classes. Comparing your and our colony-size estimates, we found that most were in agreement, but

38

ess
is

there were some cases with minor discrepancies that we recommend you have a look at (see list 1 below).

- For queen mating, B-R et al. used queen mating frequency based on harmonic means, while we used three categories: obligate single mating, facultative multiple mating, obligate multiple mating. When we compared the entries, the raw-data correspondence was strikingly good, but the B-R study lacks entries for four of the overlapping GAGA species. Again, you are welcome to use these data. Further, as we explained in the review we maintain that our categorization is more likely to cut the overall variation up in evolutionary meaningful kinds, because facultative and obligate polyandry almost certainly evolved independently from strictly single mating, – the former for enhanced diversity reasons and the latter primarily for correcting insufficient first inseminations (see Boomsma, 2013 <https://royalsocietypublishing.org/doi/full/10.1098/rstb.2012.0050>). The rationale of our categorization (also used by Boomsma & Ratnieks, 1996, Hughes et al, 2008, and in a book chapter overlooked by B-R et al. (pdf attached) is that facultative multiple insemination never abandoned single insemination so that substantial variation across populations remains, sometimes approaching pure single mating (see <https://royalsocietypublishing.org/doi/full/10.1098/rstb.2012.0050>). In contrast, obligate polyandry represents an irreversible mating system transition where singly-mated queens are never found in mature colonies (observations of a single monandrous founding-colony should be ignored when other species in the same genus are always obligatorily polyandrous, because such an incipient colony can be assumed not to survive until maturity).
- Regarding queen number, comparison with the GAGA data revealed only two cases with discrepancies (see list 1 below), but the B-R dataset lacks a number of entries as GAGA has 30 more entries in the 62 shared species. We are unsure where this difference comes from. Adding data from the attached book-chapter pdf (2009) might help and once more we offer that you can use the data from our GAGA dataset as well. As mentioned in the review, we categorized queen number in similar categories as queen-mate number (single queen, facultatively polygynous, obligately polygynous), as we did earlier in the 2009 book chapter cited above. It is our fieldwork experience that many ant species have forms of facultative polygyny (likely underestimated when this form of polygyny is rare), but this does not challenge the conclusion that monogyny and independent monogamous colony founding remain the default in those species – additional queens may then sometimes be picked up when colonies are mature (the odd newly inseminated daughter queen; the odd alien intruder that fails to be recognized as such). But the difference with obligate polygyny is quite striking because that means one never finds a mature colony in the field with just a single queen. Also here, it seems likely that the two forms of polygyny evolved independently and in response to different selection forces as in the case of queen-mate number. To validate the robustness of your conclusions, we thus recommend that you also run your analyses with our categorization of queen number which implies the same two or three contrasts as for queen-mate number.
- Regarding the number of discrete castes, there is again good correspondence and once more we offer that you can use the seven extra entries that we have in the GAGA data set (see list 2 below). We struggled with the same categorization issues as you did, but reached slightly different categories and we did not track worker head-widths systematically (a very nice addition). Thus we ended up having just three categories: monomorphic workers, continuous worker polymorphism and discretely polymorphic castes. As we expressed in the review, we are unsure whether your two ways of analyzing worker size variation represent two levels throughout, but we note that you only had worker size variation data for 117 species out of the 608 with caste number, and that only 24 of these species overlapped with the GAGA data set. In case you did not distinguish between monomorphic and continuous worker polymorphism in the analyses of single worker caste colonies, we would recommend to include a separate category for continuous worker polymorphism in your first analysis to validate the robustness of your conclusions.

We agree with the reviewer that the analysis of caste number is not able to distinguish between species with monomorphic workers and species with continuous worker polymorphism. However, we believe that it's best not to combine a 'continuous worker polymorphism' category into our measure of discrete worker castes, as this variable is naturally 'count' data and should not be combined with categorical data. Considering this limitation, we perform a separate analysis of worker size variation which should capture worker caste variation across species, including those which have continuous worker polymorphism. For the species identified in this document as having continuous worker polymorphism, we have made a new collection of all available data on head size variation using images from AntWeb.

Here are the detailed species-specific cases:

List 1 - Discrepancies:

- *Acromyrmex*: You have four *Acromyrmex* species with obligate monogyny, while we have two of these (*A. octospinosus* and *A. subterraneus*) as facultative polygynous. Also, there is a difference in colony size in some of these *Acromyrmex* species of almost one order of magnitude: e.g. *A. echinator* (you have 4,3 and we have 5,1).

- *A. octospinosus*: While in the GAGA database that was shared with us, it classifies as facultatively polygynous, the sources that it cites classifies as obligate monogynous - Boomsma et al., 2014 "The evolution of multiqueen breeding in eusocial lineages with permanent physically differentiated castes"
- *A. subterraneus* - updated accordingly.
- *A. echinator* - we include the colony size data from the GAGA database in our averaged estimate

- *Hypoponera opacior*: You have estimates of 50 workers per colony while we have an average of 120 with up to 600 workers (Collector; Markus H. Rüger, 2007). The reference is a thesis from the University of Munich: https://edoc.ub.uni-muenchen.de/8416/1/Rueger_Markus_H.pdf

- We searched the document but couldn't find the data source. Without being able to verify, we maintain our initial estimate.

- *Leptothorax acervorum*: You have a colony size estimate of 45 workers similar to our initial estimate, but we updated this to 150 based on Seifert (2018).

- Original estimate maintained as we were unable to access Seifert (2018).

- *Sericomyrmex mayri*: You have monogynous for this species based on Mehdiabadi & Schultz, 2010; but we have facultative polygyny based on more recent collections (Ana Jesovnik and Ted Schultz, who gave us 1-3 for the number of queens per colony).

- Updated according to your recommendation.

Note there has also been ambiguity about another *Sericomyrmex* species (*amabilis*). You cite (ref 23) the paper by Murakami et al concluding that this species is facultatively polyandrous. However, we later showed (with Murakami as co- author) that this result is erroneous and that for this species the higher genetic diversity was due to occasional polygyny (consistent with the *S. mayri* update), while all queens were singly inseminated. The better reference is therefore: <https://www.ncbi.nlm.nih.gov/pmc/articles/PMC1691065/pdf/12184823.pdf>. Btw, this attine ant study

40

ess
is

shows that obligate multiple insemination evolved from strictly single insemination without 'intermediate' branches with facultative polyandry. The only time facultative multiple mating was found in attine ants was for an *Acromyrmex* inquiline social parasite, which represents a secondary reduction from obligate polyandry (Sumner et al., 2000).

- We have updated *Sericomyrmex amabilis* according to your recommendation

- *Temnothorax*: For the cases where we have the same species, the colony size estimates are infra-estimated in your table compared to ours which for some species were validated with Seifert's 2018 book. Examples: For *T. longispinosus* you have an average of 24 workers while we have 85 with up to 1000 in our table; for *T. nylanderi* you have 54 while we have 85 with up to 500; for *T. unifasciatus* you have 67 while we have 134 with up to 500).

- Original estimates were maintained as we were unable to access Seifert (2018).

- *Typhlomyrmex rogenhoferi*: For colony size you have 300 workers (Brown, 1965) while we have 2000 (from Benoit Guenard and Nate Sanders, personal communication).

- We include the colony size data from the GAGA database in our averaged estimate.

- *Messor barbarus*. You have that there is one worker caste, but we had this species noted down as also having soldiers (i.e. having discrete worker polymorphism). We realised that our entry was wrong and corrected it based on your reference (Bernadou et al. 2016: <https://doi.org/10.1242/jeb.141556>). However, this species would still have continuous worker polymorphism.

- We collected additional data on worker head widths for this species and calculated variation in worker size.

Cases of continuous worker polymorphism:

- *Cataglyphis aenescens*: You have one worker caste, but according to our information this species has continuous worker polymorphism.

- All *Formica* species: You write they have a single worker caste, but our information (validated with Heikki) is that they all have continuous worker polymorphism.

- *Lasius flavus*: Also here we have continuous worker polymorphism.

- *Megaponera analis*: We have continuous worker polymorphism also here, based on information from the collector (Erik Frank), who has a publication under submission, while you have one worker caste without further information on continuous worker size variation.

- *Mystridium*: You have one worker caste while we have continuous worker polymorphism for this species (genus).

- *Paraponera clavata*: you have one worker caste, while we have continuous worker polymorphism (Breed and Harrison 1988 <https://www.jstor.org/stable/25085003?seq=5>).

41

As mentioned above, we recognise that there are some species that have one worker caste but also have continuous worker polymorphism. We do not feel that it's appropriate to include an additional category for species with a single worker caste but with worker polymorphism into our analysis of the number of discrete worker castes as it is mixing different types of data: 'count' data and categorical

ess
is

data. However, to account for the presence of species with one worker caste but which also have continuous worker polymorphism, we also perform a second analysis in our paper where we measure the size of ant heads, which should capture this type of variation. We ensure that head size measurements have been collected, where available, for all of the species highlighted in the reviewers' comments as having a single worker caste but with polymorphic workers.

List 2 - Missing data in your table that we have in the GAGA data set:

Most of these are in agreement with data you have for other species in the same genus, but you do not have entries for the following species:

We amend our data according to your recommendation unless otherwise stated next to the species name.

- *Aphaenogaster subterraneus*: colonies with 2000 and up to 4000 workers.
- *Camponotus japonicus*: Facultative polygyny and colonies with 1500-2000 workers.
- *Camponotus singularis*: Monogyne and colonies of 1000 workers.
- *Carebara diversa* is facultative polygynous.
- *Cataglyphis aenescens* is monogyne.
- *Crematogaster osakensis* workers are monomorphic (one worker caste).
- *Discothyrea kamiteta* is facultative polygynous, workers are monomorphic. *Could not verify 'facultative polygyny' based on the citation provided in the GAGA database.*
- *Hypoponera opacior*: facultative polygynous.
- *Iberofornica subrufa*: obligately polygynous.
- *Kalathomyrmex emeryi*: monogynous.
- *Leptanilla*: monogynous. *Could not verify that this species is monogynous based on citation provided in the GAGA database.*
- *Leptogenys diminuta*: monogynous.
- *Liometopum microcephalum* is facultative polygynous, and colonies have up to millions of workers (Seifert 2018). *Could not verify this species for both facultatively polygyny or colony size based on the citations provided in the GAGA database.*
- *Manica rubida*: colony size 500, up to 1000 workers; facultative polygynous. *Could not verify colony size based on the citations provided in the GAGA database.*
- *Megaponera analis*: monoandrous (from collector, paper in review).
- *Messor capitatus*: colonies of 3500-7000 workers. Monogynous.
- *Messor barbarus*: monogynous.
- *Mycetomoellerius zeteki*: monoandry and monogyny.
- *Mycetosoritis hartmanni*: monoandry and monogyny. *Species estimate for mating frequency based on genus estimate and therefore not included in our database.*
- *Myrmecia pilosula*: monomorphic workers.
- *Myrmecina graminicola*: colonies with on average 45 workers with up to 300 (you have 27 on average). Monoandrous and facultative polygynous. *Could not verify queen mating frequency based on citation in the GAGA database.*
- *Mystrium camillae*: facultative polygyny and continuous worker polymorphism.
- *Ochetellus glaber*: obligately polygynous.
- *Odontomachus brunneus* is monogynous.
- *Paraponera clavata* is monogynous.
- *Paratrachymyrmex cornetzi*: monoandrous and monogynous.
- *Polyrhachis illaudata* is monogynous.
- *Pseudomyrmex spinicola*: monogynous and polymorphic workers.
- *Stenamma debile*: facultative polygynous; colonies of 56 +/-40 workers. *Could not verify queen number based on citation in the GAGA database.*
- *Stictoponera bicolor*: monogynous and colonies of 480 workers.

42

ess
is

- *Strumigenys mutica*: monomorphic workers.
- *Trachymyrmex septentrionalis*: monoandrous. Mating frequency estimate not included as it is inferred from other members of the genus rather than directly estimated for this species.
- *Typhlomyrmex rogenhoferi*: monogynous and continuous worker polymorphism.

Copenhagen, 22 November 2023 Joel Vizueta and Koos Boomsma

Incorporating valuable insights from Joel and Koos, and the references and citations provided, we have expanded our analysis by including an additional 36 species. Furthermore, we have added additional data for numerous species that were already part of our analysis but did not have complete data.

Decision Letter, second revision:

Our ref: NATECOLEVOL-23092284B

29th April 2024

Dear Dr. Bell-Roberts,

Thank you for your patience as we've prepared the guidelines for final submission of your Nature Ecology & Evolution manuscript, "Larger colony sizes favoured more worker castes in ants" (NATECOLEVOL-23092284B). Please carefully follow the step-by-step instructions provided in the attached file, and add a response in each row of the table to indicate the changes that you have made. Please also check and comment on any additional marked-up edits we have proposed within the text. Ensuring that each point is addressed will help to ensure that your revised manuscript can be swiftly handed over to our production team.

****We would like to start working on your revised paper, with all of the requested files and forms, as soon as possible (preferably within two weeks). Please get in contact with us immediately if you anticipate it taking more than two weeks to submit these revised files.****

In recognition of the time and expertise our reviewers provide to Nature Ecology & Evolution's editorial process, we would like to formally acknowledge their contribution to the external peer review of your manuscript entitled "Larger colony sizes favoured more worker castes in ants". For those reviewers who give their assent, we will be publishing their names alongside the published article.

Nature Ecology & Evolution offers a Transparent Peer Review option for new original research manuscripts submitted after December 1st, 2019. As part of this initiative, we encourage our authors to support increased transparency into the peer review process by agreeing to have the reviewer comments, author rebuttal letters, and editorial decision letters published as a Supplementary item. When you submit your final files please clearly state in your cover letter whether or not you would like to participate in this initiative. Please note that failure to state your preference will result in delays in accepting your manuscript for publication.

44Cover suggestions

We welcome submissions of artwork for consideration for our cover. For more information, please see our guide for cover artwork.

Nature Ecology & Evolution has now transitioned to a unified Rights Collection system which will allow our Author Services team to quickly and easily collect the rights and permissions required to publish your work. Approximately 10 days after your paper is formally accepted, you will receive an email in providing you with a link to complete the grant of rights. If your paper is eligible for Open Access, our Author Services team will also be in touch regarding any additional information that may be required to arrange payment for your article.

Please note that *Nature Ecology & Evolution* is a Transformative Journal (TJ). Authors may publish their research with us through the traditional subscription access route or make their paper immediately open access through payment of an article-processing charge (APC). Authors will not be required to make a final decision about access to their article until it has been accepted. Find out more about Transformative Journals

Authors may need to take specific actions to achieve compliance with funder and institutional open access mandates. If your research is supported by a funder that requires immediate open access (e.g. according to Plan S principles) then you should select the gold OA route, and we will direct you to the compliant route where possible. For authors selecting the subscription publication route, the journal's standard licensing terms will need to be accepted, including <https://www.nature.com/nature-portfolio/editorial-policies/self-archiving-and-license-to-publish>. Those licensing terms will supersede any other terms that the author or any third party may assert apply to any version of the manuscript.

Please use the following link for uploading these materials:
[REDACTED]

45[REDACTED]

Reviewer #1:

Remarks to the Author:

The authors have addressed our earlier comments or argued that constraints prevented to fully meet our suggestions, so we have mostly some minor cosmetic comments to offer. However, there is one conceptual issue that we would like criticize for the way in which it is currently analyzed and presented. This escaped our attention when we reviewed the first version of the manuscript because it then remained more implicit but we think it is important to get this clarified.

Major comment 1:

The authors used two ways to define aspects of differentiation of the somatic worker caste: 1. The number of discrete worker/soldier castes (1-4) and 2. The continuous variation within worker castes, particularly in the majority of ant species with a single worker caste.

The problem is that in the present text 1 and 2 are considered to form a kind of continuum in which 2 is claimed to be a precursor of 1. In our opinion, there is no empirical evidence for that inference and neither is it conceptually compelling. Our arguments are as follows:

1. Differentiation of physically distinct castes (first between gynes and workers and secondarily within the worker caste) reflects deeply anchored developmental programs that play out across the 3-4 larval stages. These developments towards distinct caste phenotypes are mediated by canalized gene-expression, in turn often driven by juvenile hormone (JH) as master regulator establishing discrete body sizes and potential allometries (for example disproportionally large heads in soldiers). There is no doubt that this kind of caste differentiation is adaptive at the colony (i.e. family) level. In comparison, continuous variation within castes is merely variation in body size (often unimodal or at best weakly bimodal) emerging shortly before pupation. This may be adaptive for dealing with different tasks but part of this variation may also be merely stochastic depending on variation in resources across the seasons or the stages of colony growth. There is no evidence that we know of for these two categories of differentiation being homologous, which would be required for inferring that 2 is a precursor of 1. It may be in some cases, but it may also often not be. For example, the evolution of so-called supersoldiers in some Pheidole ants

(<https://www.science.org/doi/epdf/10.1126/science.1211451>) depends on ancient developmental modules that have no obvious relationship to quantitative size variation within a normal soldier caste. We therefore propose that the authors delete or strongly modify their text on line 275-278 to make this caveat or to discard the entire point. Note that this argument has interesting parallels in the difference between facultative multiple mating and obligate multiple mating as well as the difference between facultative and obligate altruism, which also evolved for different reasons. We have all been brainwashed by eusociality-continuum thinking into implicitly but incorrectly assuming that facultative must be a precursor of the obligate.

2. This relates to the ambiguous results and reconstruction of causation that emerged in the analyses of quantitative variation within castes (in contrast to the much clearer results for qualitative caste differentiation). Given the doubtful adaptive significance of quantitative body size variation among workers outlined above, the ambiguous outcome on causation between colony size and variation in

46worker size (Figure 6) is not surprising. However, we feel that this analysis should have included the number of discrete worker castes to be comparable to Figure 4. This could be achieved by adding worker size variation to the phylogenetic path model instead of replacing the discrete worker caste variable by worker size variation, i.e. by using a model with five instead of four variables. As far as we can see, only such comparison can decide whether variation in worker size is really an independent causal factor or whether such combined analysis will essentially recover Figure 4 with variation in worker size disappearing as independent factor just like queen number did in the earlier analyses. Depending on this outcome, the Results section on worker size variation (ca. line 282-299) needs to be reconsidered, and the Discussion section (364-372) could then address the outcome of this new analyses and summarize the main conclusion about the two alternative forms of variation in the somatic worker caste.

Minor comments:

Line 42: Check ref 10. We think it is one of these papers that assumes that major transitions evolved gradually by facultative altruism somehow going to fixation. There is no evidence for that contention and substantial indirect evidence against it. Consider skipping this reference if our suspicion is correct.

Line 48-49: Write "the ants" instead of "social insects" – your six references here are exclusively or almost only about ants.

Line 54: Is reference 21 an ant paper? If not skip or change the start of the sentence into something like "in ants and other superorganismal social insects".

Line 56: Write "maximal" instead of "maximized" – the latter would suggest (erroneously) that this reflects a process of increasingly higher relatedness without necessarily reaching the maximal possible value.

Line 66: Also here write "in ants and other superorganismal social insects" or just "superorganismal social insects" – none of this applies in the more open types of insect societies.

Line 85: Write "information" instead of "data"?

Line 106: Rephrase into "males that inseminated each queen varied ...".

Line 112: "dataset" instead of "data"?

Line 123: The figure seems only about "distribution" so skip "and evolution" in the first sentence of the legend?

Line 131: "plotted" rather than "represented"?

Figure 2: Some of the genus names around the figure seem poorly aligned.

Line 140: Write "phylogenetic lineages" instead of "taxonomic groups".

Line 141-142: Write "Analyses were repeated ..." ... "a single representative tree"?

Lines 154, 158, 161 (possibly other places as well): Readers would normally expect that R-square values inform about the % of variance explained, but here R-squares are used as some kind of statistical significance parameter. We think this should be explained here because many readers will not pick this up from the methods section (line 560-561).

Line 211: Consider adding at the end of the sentence something like "after adjusting for other potential predictor variables".

Line 229: Delete "is"?

Line 244: "were estimated".

Line 272: Write "ant species" rather than "species"?

Line 275-278: The major comment above suggest you skip these four lines.

Line 280-299: To be reconsidered after doing a combined analyses with 5 factors rather than 4 (see

47major comment above) – also reconsider Figure 6b.

Line 315: Add at the end of this sentence something like “but opposite causation remains possible.”, in order to make the caveat explicit that you really cannot draw conclusions here. Or rephrase in another way if the combined analysis of 5 factors would change the overall interpretation.

Line 336: Make explicit that you now found a positive correlation although not (quite) significant.

Line 370-372: Given our major comment above, we think these lines should reflect the lack of homology between discrete and continuous variation in worker size by skipping the word “initial”, by making the phrasing more explicit, and by allowing space for interpreting the additional analysis that we proposed. The sentence could then be something like: “Consequently, while our results supported the size complexity hypothesis for discrete castes (Figs. 4-5) and found a correlation between colony size and continuous variation in worker size (Fig 6a), they also indicated that higher continuous variation in worker size induced rather than followed increasing colony size in ant species with only a single worker caste.” This could then be supplemented by a next sentence referring to the new analysis, reading like: “An extended path analysis considering both distinct and continuous variation in worker size (in addition to queen number, queen mating frequency and colony size) showed that xxx xxx etc.”

Line 390: Specify “absent in vespine wasps” (polistines, stenogastrines etc. are not superorganismal).

Line 391: The Hölldobler & Wilson book (2009)(ref 52) seems irrelevant here because the Grüter papers discovering these phenomena are from 2012 (ref 51) and 2017 (ref 33).

Line 396: Write “superorganismal bees” rather than “social bees” to avoid any confusion with halictines, allodapines etc.

Line 404-409: In light of the major comment above we suggest you also rewrite this concluding section, particularly reinterpreting the quantitative variation in worker size. On line 408-409, write again “superorganismal” rather than “social” insects because outside the superorganismal lineages there is only continuous variation that affects the odds of becoming a breeder or helper. The fact that many use queen-worker terminology in e.g. halictid bees does not imply these are castes in the pre-imaginal development sense (see major comment above).

Line 429: Also cite ref 23 here.

Line 439: I think this cutoff should also include ref 41 (2009) – it is unclear what “was already contained within our dataset” (line 441) implies.

Line 456: “available type of”?

Line 462: Write “queens” rather than “individuals”. You excluded gamergates and parthenogenetic species (line 510-518) so your dataset should not contain any cases of inseminated workers.

Line 492: Numbered reference for Dornhaus is missing.

Line 517: Consider adding after “sexually)” something like “after the loss of the original queen caste”?

Line 523-524: To make more explicit rephrase into: “This decision to remove species characterized by evolutionarily derived social systems (often of recent origin) improved our ability to detect which variables improved the deeper evolution of the number of worker castes.” The rationale is that parthenogenesis, gamergates, ergatoids and inquilines essentially never created adaptive radiations in the ant tree of life.

Line 563: “greater than”.

Line 757: Specify better by writing “that may have evolved for different reasons and aligns ...”

Line 761: Add 1-2 references about where these categorizations were used before.

Using past tense where appropriate has been implemented but not fully. We picked up some more

48places where we felt it would be formally more correct to use past rather than present sense: line 90 (provided and helped); line 94 (allowed); line 204 (facilitated); line 213 (was); line 214 (were); line 219 (allowed); line 220 (required); line 309 (were); 316 (were); line 325 (were); line 333 (were); line 338 (was); line 364 (favoured); line 369 (preceded); line 400 (was); line 417 (examined and allowed); line 472 (were); line 479 (were); line 573 (were); line 583 (was); line 588 (influenced), line 589 (did); line 590 (influenced and predicted); line 591 (influenced); line 592 (influenced); line 593 (did); line 595 (was); line 597 (has been); line 601 (hypothesized); line 736 (required); line 998 (was).

Copenhagen, 17 April 2024 Koos Boomsma and Joel Vizuela

Reviewer #2:

Remarks to the Author:

I have read the responses from the authors and the updated manuscript. My comments have been satisfyingly addressed, and the manuscript has been considerably improved in response to all reviewers.

Reviewer #3:

Remarks to the Author:

My concerns were fully resolved or addressed by the authors.

Reviewer #4:

Remarks to the Author:

The authors have addressed our earlier comments or argued that constraints prevented to fully meet our suggestions, so we have mostly some minor cosmetic comments to offer. However, there is one conceptual issue that we would like criticize for the way in which it is currently analyzed and presented. This escaped our attention when we reviewed the first version of the manuscript because it then remained more implicit but we think it is important to get this clarified.

Major comment 1:

The authors used two ways to define aspects of differentiation of the somatic worker caste: 1. The number of discrete worker/soldier castes (1-4) and 2. The continuous variation within worker castes, particularly in the majority of ant species with a single worker caste.

The problem is that in the present text 1 and 2 are considered to form a kind of continuum in which 2 is claimed to be a precursor of 1. In our opinion, there is no empirical evidence for that inference and neither is it conceptually compelling. Our arguments are as follows:

1. Differentiation of physically distinct castes (first between gynes and workers and secondarily within the worker caste) reflects deeply anchored developmental programs that play out across the 3-4 larval

49stages. These developments towards distinct caste phenotypes are mediated by canalized gene-expression, in turn often driven by juvenile hormone (JH) as master regulator establishing discrete body sizes and potential allometries (for example disproportionately large heads in soldiers). There is no doubt that this kind of caste differentiation is adaptive at the colony (i.e. family) level. In comparison, continuous variation within castes is merely variation in body size (often unimodal or at best weakly bimodal) emerging shortly before pupation. This may be adaptive for dealing with different tasks but part of this variation may also be merely stochastic depending on variation in resources across the seasons or the stages of colony growth. There is no evidence that we know of for these two categories of differentiation being homologous, which would be required for inferring that 2 is a precursor of 1. It may be in some cases, but it may also often not be. For example, the evolution of so-called supersoldiers in some *Pheidole* ants

(<https://www.science.org/doi/epdf/10.1126/science.1211451>) depends on ancient developmental modules that have no obvious relationship to quantitative size variation within a normal soldier caste. We therefore propose that the authors delete or strongly modify their text on line 275-278 to make this caveat or to discard the entire point. Note that this argument has interesting parallels in the difference between facultative multiple mating and obligate multiple mating as well as the difference between facultative and obligate altruism, which also evolved for different reasons. We have all been brainwashed by eusociality-continuum thinking into implicitly but incorrectly assuming that facultative must be a precursor of the obligate.

2. This relates to the ambiguous results and reconstruction of causation that emerged in the analyses of quantitative variation within castes (in contrast to the much clearer results for qualitative caste differentiation). Given the doubtful adaptive significance of quantitative body size variation among workers outlined above, the ambiguous outcome on causation between colony size and variation in worker size (Figure 6) is not surprising. However, we feel that this analysis should have included the number of discrete worker castes to be comparable to Figure 4. This could be achieved by adding worker size variation to the phylogenetic path model instead of replacing the discrete worker caste variable by worker size variation, i.e. by using a model with five instead of four variables. As far as we can see, only such comparison can decide whether variation in worker size is really an independent causal factor or whether such combined analysis will essentially recover Figure 4 with variation in worker size disappearing as independent factor just like queen number did in the earlier analyses. Depending on this outcome, the Results section on worker size variation (ca. line 282-299) needs to be reconsidered, and the Discussion section (364-372) could then address the outcome of this new analyses and summarize the main conclusion about the two alternative forms of variation in the somatic worker caste.

Minor comments:

Line 42: Check ref 10. We think it is one of these papers that assumes that major transitions evolved gradually by facultative altruism somehow going to fixation. There is no evidence for that contention and substantial indirect evidence against it. Consider skipping this reference if our suspicion is correct.

Line 48-49: Write "the ants" instead of "social insects" – your six references here are exclusively or almost only about ants.

Line 54: Is reference 21 an ant paper? If not skip or change the start of the sentence into something like "in ants and other superorganismal social insects".

Line 56: Write "maximal" instead of "maximized" – the latter would suggest (erroneously) that this reflects a process of increasingly higher relatedness without necessarily reaching the maximal possible

50value.

Line 66: Also here write "in ants and other superorganismal social insects" or just "superorganismal social insects" – none of this applies in the more open types of insect societies.

Line 85: Write "information" instead of "data"?

Line 106: Rephrase into "males that inseminated each queen varied ...".

Line 112: "dataset" instead of "data"?

Line 123: The figure seems only about "distribution" so skip "and evolution" in the first sentence of the legend?

Line 131: "plotted" rather than "represented"?

Figure 2: Some of the genus names around the figure seem poorly aligned.

Line 140: Write "phylogenetic lineages" instead of "taxonomic groups".

Line 141-142: Write "Analyses were repeated ..." ... "a single representative tree"?

Lines 154, 158, 161 (possibly other places as well): Readers would normally expect that R-square values inform about the % of variance explained, but here R-squares are used as some kind of statistical significance parameter. We think this should be explained here because many readers will not pick this up from the methods section (line 560-561).

Line 211: Consider adding at the end of the sentence something like "after adjusting for other potential predictor variables".

Line 229: Delete "is"?

Line 244: "were estimated".

Line 272: Write "ant species" rather than "species"?

Line 275-278: The major comment above suggest you skip these four lines.

Line 280-299: To be reconsidered after doing a combined analyses with 5 factors rather than 4 (see major comment above) – also reconsider Figure 6b.

Line 315: Add at the end of this sentence something like "but opposite causation remains possible.", in order to make the caveat explicit that you really cannot draw conclusions here. Or rephrase in another way if the combined analysis of 5 factors would change the overall interpretation.

Line 336: Make explicit that you now found a positive correlation although not (quite) significant.

Line 370-372: Given our major comment above, we think these lines should reflect the lack of homology between discrete and continuous variation in worker size by skipping the word "initial", by making the phrasing more explicit, and by allowing space for interpreting the additional analysis that we proposed. The sentence could then be something like: "Consequently, while our results supported the size complexity hypothesis for discrete castes (Figs. 4-5) and found a correlation between colony size and continuous variation in worker size (Fig 6a), they also indicated that higher continuous variation in worker size induced rather than followed increasing colony size in ant species with only a single worker caste." This could then be supplemented by a next sentence referring to the new analysis, reading like: "An extended path analysis considering both distinct and continuous variation in worker size (in addition to queen number, queen mating frequency and colony size) showed that xxx xxx etc."

Line 390: Specify "absent in vespine wasps" (polistines, stenogastrines etc. are not superorganismal).

Line 391: The Hölldobler & Wilson book (2009)(ref 52) seems irrelevant here because the Grüter papers discovering these phenomena are from 2012 (ref 51) and 2017 (ref 33).

Line 396: Write "superorganismal bees" rather than "social bees" to avoid any confusion with halictines, allodapines etc.

Line 404-409: In light of the major comment above we suggest you also rewrite this concluding

section, particularly reinterpreting the quantitative variation in worker size. On line 408-409, write again "superorganismal" rather than "social" insects because outside the superorganismal lineages there is only continuous variation that affects the odds of becoming a breeder or helper. The fact that many use queen-worker terminology in e.g. halictid bees does not imply these are castes in the pre-imaginal development sense (see major comment above).

Line 429: Also cite ref 23 here.

Line 439: I think this cutoff should also include ref 41 (2009) – it is unclear what "was already contained within our dataset" (line 441) implies.

Line 456: "available type of"?

Line 462: Write "queens" rather than "individuals". You excluded gamergates and parthenogenetic species (line 510-518) so your dataset should not contain any cases of inseminated workers.

Line 492: Numbered reference for Dornhaus is missing.

Line 517: Consider adding after "sexually" something like "after the loss of the original queen caste"?

Line 523-524: To make more explicit rephrase into: "This decision to remove species characterized by evolutionarily derived social systems (often of recent origin) improved our ability to detect which variables improved the deeper evolution of the number of worker castes." The rationale is that parthenogenesis, gamergates, ergatoids and inquilines essentially never created adaptive radiations in the ant tree of life.

Line 563: "greater than".

Line 757: Specify better by writing "that may have evolved for different reasons and aligns ..."

Line 761: Add 1-2 references about where these categorizations were used before.

Using past tense where appropriate has been implemented but not fully. We picked up some more places where we felt it would be formally more correct to use past rather than present sense: line 90 (provided and helped); line 94 (allowed); line 204 (facilitated); line 213 (was); line 214 (were); line 219 (allowed); line 220 (required); line 309 (were); 316 (were); line 325 (were); line 333 (were); line 338 (was); line 364 (favoured); line 369 (preceded); line 400 (was); line 417 (examined and allowed); line 472 (were); line 479 (were); line 573 (were); line 583 (was); line 588 (influenced), line 589 (did); line 590 (influenced and predicted); line 591 (influenced); line 592 (influenced); line 593 (did); line 595 (was); line 597 (has been); line 601 (hypothesized); line 736 (required); line 998 (was).

Copenhagen, 17 April 2024 Koos Boomsma and Joel Vizueta

Author Rebuttal, second revision:

We thank the reviewers for checking the manuscript again and for their additional constructive comments.

In this document, we use **black text** for the reviewers' comments, and **blue text** for our response.

Reviewer #1 (Remarks to the Author):

The authors have addressed our earlier comments or argued that constraints prevented to fully meet our suggestions, so we have mostly some minor cosmetic comments to offer. However, there is one conceptual issue that we would like criticize for the way in which it is currently analyzed and presented. This escaped our attention when we reviewed the first version of the manuscript because it then remained more implicit but we think it is important to get this clarified.

Major comment 1:

The authors used two ways to define aspects of differentiation of the somatic worker caste: 1. The number of discrete worker/soldier castes (1-4) and 2. The continuous variation within worker castes, particularly in the majority of ant species with a single worker caste.

The problem is that in the present text 1 and 2 are considered to form a kind of continuum in which 2 is claimed to be a precursor of 1. In our opinion, there is no empirical evidence for that inference and neither is it conceptually compelling. Our arguments are as follows:

1. Differentiation of physically distinct castes (first between gynes and workers and secondarily within the worker caste) reflects deeply anchored developmental programs that play out across the 3-4 larval stages. These developments towards distinct caste phenotypes are mediated by canalized gene-expression, in turn often driven by juvenile hormone (JH) as master regulator establishing discrete body sizes and potential allometries (for example disproportionally large heads in soldiers). There is no doubt that this kind of caste differentiation is adaptive at the colony (i.e. family) level. In comparison, continuous variation within castes is merely variation in body size (often unimodal or at best weakly bimodal) emerging shortly before pupation. This may be adaptive for dealing with different tasks but part of this variation may also be merely stochastic depending on variation in resources across the seasons or the stages of colony growth. There is no evidence that we know of for these two categories of differentiation being homologous, which would be required for inferring that 2 is a precursor of 1. It may be in some

53cases, but it may also often not be. For example, the evolution of so-called supersoldiers in some Pheidole ants (<https://www.science.org/doi/epdf/10.1126/science.1211451>) depends on ancient developmental modules that have no obvious relationship to quantitative size variation within a normal soldier caste. We therefore propose that the authors delete or strongly modify their text on line 275-278 to make this caveat or to discard the entire point. Note that this argument has interesting parallels in the difference between facultative multiple mating and obligate multiple mating as well as the difference between facultative and obligate altruism, which also evolved for different reasons. We have all been brainwashed by eusociality-continuum thinking into implicitly but incorrectly assuming that facultative must be a precursor of the obligate.

We have discarded the point made on line 275-278 to reflect the reviewer's comment. We have also edited lines 374-376 to remove the implication that continuous variation in worker size is on an evolutionary continuum with discrete worker castes.

2. This relates to the ambiguous results and reconstruction of causation that emerged in the analyses of quantitative variation within castes (in contrast to the much clearer results for qualitative caste differentiation). Given the doubtful adaptive significance of quantitative body size variation among workers outlined above, the ambiguous outcome on causation between colony size and variation in worker size (Figure 6) is not surprising. However, we feel that this analysis should have included the number of discrete worker castes to be comparable to Figure 4. This could be achieved by adding worker size variation to the phylogenetic path model instead of replacing the discrete worker caste variable by worker size variation, i.e. by using a model with five instead of four variables. As far as we can see, only such comparison can decide whether variation in worker size is really an independent causal factor or whether such combined analysis will essentially recover Figure 4 with variation in worker size disappearing as independent factor just like queen number did in the earlier analyses. Depending on this outcome, the Results section on worker size variation (ca. line 282-299) needs to be reconsidered, and the Discussion section (364-372) could then address the outcome of this new analyses and summarize the main conclusion about the two alternative forms of variation in the somatic worker caste.

We recognise the point made by the reviewers that variation in worker size is not necessarily a precursor for the evolution of discrete worker castes and we have edited the manuscript to reflect this point.

However, we disagree that a path analysis including both measures of non-reproductive division of labour (1. Number of discrete worker castes; 2. Variation in worker size) is a necessary or appropriate analysis to perform. These two variables are alternative methods of quantifying the same thing – worker variation.

Our objective is to identify the causal pathway by which non-reproductive division of labour is related to other ant life history traits – making phylogenetic path analysis a useful tool for achieving this goal.

However, it is not our objective to select between two alternative methods of quantifying worker variation. Therefore, we do not think that the analysis suggested by the reviewers is appropriate. Instead, number of discrete worker castes and variation in worker sizes should only be analysed in separate path analyses, avoiding performing a model selection comparison between these variables. This approach is already taken, where we repeat our analysis examining non-reproductive division of labour quantified in two different ways.

Minor comments:

Line 42: Check ref 10. We think it is one of these papers that assumes that major transitions evolved gradually by facultative altruism somehow going to fixation. There is no evidence for that contention and substantial indirect evidence against it. Consider skipping this reference if our suspicion is correct.

We have removed this citation.

Line 48-49: Write “the ants” instead of “social insects” – your six references here are exclusively or almost only about ants.

We have changed to ‘ants.’

Line 54: Is reference 21 an ant paper? If not skip or change the start of the sentence into something like “in ants and other superorganismal social insects”.

We have removed reference 21 as it is not an ant specific paper.

Line 56: Write “maximal” instead of “maximized” – the latter would suggest (erroneously) that this reflects a process of increasingly higher relatedness without necessarily reaching the maximal possible value.

We have changed to ‘maximal.’

Line 66: Also here write “in ants and other superorganismal social insects” or just “superorganismal social insects” – none of this applies in the more open types of insect societies.

We have changed to ‘superorganismal social insects.’

Line 85: Write “information” instead of “data”?

We have changed to ‘information.’

Line 106: Rephrase into “males that inseminated each queen varied ...”.

We have made the suggested change.

Line 112: “dataset” instead of “data”?

We change to ‘dataset.’

Line 123: The figure seems only about “distribution” so skip “and evolution” in the first sentence of the legend?

We have removed ‘and evolution.’

Line 131: “plotted” rather than “represented”?

We have changed to ‘plotted.’

Figure 2: Some of the genus names around the figure seem poorly aligned.

Silhouettes and genus names have been realigned.

Line 140: Write “phylogenetic lineages” instead of “taxonomic groups”.

Altered to ‘phylogenetic lineages.’

Line 141-142: Write “Analyses were repeated ...” ... “a single representative tree”?

We accept the proposed change.

Lines 154, 158, 161 (possibly other places as well): Readers would normally expect that R-square values inform about the % of variance explained, but here R-squares are used as some kind of statistical significance parameter. We think this should be explained here because many readers will not pick this up from the methods section (line 560-561).

R-squared values should be interpreted in the normal way throughout the manuscript i.e. providing a measure of the proportion of variance explained. Confusion regarding this point may have been caused by

R-squared being reported as ' $R^2 < 0.01$ ' in some cases in the text? In case this is the issue causing confusion, we have altered these sections to report R^2 as ' $R^2 = 0.00$.'

Line 211: Consider adding at the end of the sentence something like “after adjusting for other potential predictor variables”.

Thank you, we have included this point.

Line 229: Delete “is”?

We have deleted ‘is.’

Line 244: “were estimated”.

Thank you, we have changed.

Line 272: Write “ant species” rather than “species”?

We have changed to ‘ant species’ and adjusted the references to reflect this.

Line 275-278: The major comment above suggest you skip these four lines.

These lines are deleted.

Line 280-299: To be reconsidered after doing a combined analyses with 5 factors rather than 4 (see major comment above) – also reconsider Figure 6b.

The text between lines 280-299 does not require changes, as explained in response to the 'major comments' above.

Line 315: Add at the end of this sentence something like “but opposite causation remains possible.”, in order to make the caveat explicit that you really cannot draw conclusions here. Or rephrase in another way if the combined analysis of 5 factors would change the overall interpretation.

We make changes to include the caveat more explicitly.

Line 336: Make explicit that you now found a positive correlation although not (quite) significant.

We already state that “we identified a single supported model where the evolution of multiple worker castes is favoured by both larger colony sizes and obligate multiple mating by queens”. This makes explicit that there’s a positive association between obligate multiple mating and multiple worker castes. However, when analysing the magnitude of the path coefficients and their associated confidence intervals, we feel that it’s not correct to suggest a nearly significant result. For the majority of the trees analysed, the confidence interval overlaps with zero: 0.99 (-0.28-2.18), 1.02 (-0.11-2.29), 1.05 (0.22-2.47). Therefore, we feel the current wording is most appropriate.

Line 370-372: Given our major comment above, we think these lines should reflect the lack of homology between discrete and continuous variation in worker size by skipping the word “initial”, by making the phrasing more explicit, and by allowing space for interpreting the additional analysis that we proposed. The sentence could then be something like: “Consequently, while our results supported the size complexity hypothesis for discrete castes (Figs. 4-5) and found a correlation between colony size and continuous variation in worker size (Fig 6a), they also indicated that higher continuous variation in worker size induced rather than followed increasing colony size in ant species with only a single worker caste.” This could then be supplemented by a next sentence referring to the new analysis, reading like: “An extended path analysis considering both distinct and continuous variation in worker size (in addition to queen number, queen mating frequency and colony size) showed that xxx xxx xxx etc.”

We have removed the word ‘initial’ to avoid implying that continuous variation in worker size is on an evolutionary continuum with discrete worker castes. However, further interpretation of the results is not required as an additional path analysis has not been performed. The reasons for this decision are given above when responding to the ‘major comments.’

Line 390: Specify “absent in vespine wasps” (polistines, stenogastrines etc. are not superorganismal).

We have amended.

Line 391: The Hölldobler & Wilson book (2009)(ref 52) seems irrelevant here because the Grüter papers discovering these phenomena are from 2012 (ref 51) and 2017 (ref 33).

We remove this citation.

Line 396: Write “superorganismal bees” rather than “social bees” to avoid any confusion with halictines, allodapines etc.

We include ‘superorganismal.’

Line 404-409: In light of the major comment above we suggest you also rewrite this concluding section, particularly reinterpreting the quantitative variation in worker size.

Given that our analyses for both discrete worker castes and continuous variation in worker size (analysing across species with varying numbers of physical castes) both support the size-complexity hypothesis, we feel that the current wording of the conclusion is appropriate. We only fail to find a causal link for colony size favouring greater variation in worker size when examining only species with a single physical worker caste.

In addition, no further analyses have been conducted (reasons highlighted above); therefore, they do not need to be considered in the concluding paragraph.

On line 408-409, write again “superorganismal” rather than “social” insects because outside the superorganismal lineages there is only continuous variation that affects the odds of becoming a breeder or helper. The fact that many use queen-worker terminology in e.g. halictid bees does not imply these are castes in the pre-imaginal development sense (see major comment above).

We include ‘superorganismal.’

Line 429: Also cite ref 23 here.

We include this citation.

Line 439: I think this cutoff should also include ref 41 (2009) – it is unclear what “was already contained within our dataset” (line 441) implies.

We have revised these lines for improved clarity. However, we believe it is best to retain the original references, as only these references were used to guide the methodological approach during the literature search.

Line 456: “available type of”?

Thank you, we have rephrased.

Line 462: Write “queens” rather than “individuals”. You excluded gamergates and parthenogenetic species (line 510-518) so your dataset should not contain any cases of inseminated workers.

We have revised to ‘queens.’

Line 492: Numbered reference for Dornhaus is missing.

We have included the numbered reference.

Line 517: Consider adding after “sexually)” something like “after the loss of the original queen caste”?

We revise to include ‘following the evolutionary loss of the original queen caste.’

Line 523-524: To make more explicit rephrase into: “This decision to remove species characterized by evolutionarily derived social systems (often of recent origin) improved our ability to detect which variables improved the deeper evolution of the number of worker castes.” The rationale is that parthenogenesis, gamergates, ergatoids and inquilines essentially never created adaptive radiations in the ant tree of life.

We have rephrased to ‘This decision to remove species characterised by evolutionarily derived social systems improved our ability to detect which variables could influence the evolution of the number of worker castes.’

Line 563: “greater than”.

Thank you for spotting, we have changed.

Line 757: Specify better by writing “that may have evolved for different reasons and aligns ...”

We have included the suggested change.

Line 761: Add 1-2 references about where these categorizations were used before.

We have included Boomsma, 2013 and Boomsma et al., 2014.

Using past tense where appropriate has been implemented but not fully. We picked up some more places where we felt it would be formally more correct to use past rather than present tense: line 90 (provided and helped); line 94 (allowed); line 204 (facilitated); line 213 (was); line 214 (were); line 219 (allowed); line 220 (required); line 309 (were); 316 (were); line 325 (were); line 333 (were); line 338 (was); line 364 (favoured); line 369 (preceded); line 400 (was); line 417 (examined and allowed); line 472 (were); line 479 (were); line 573 (were); line 583 (was); line 588 (influenced), line 589 (did); line 590 (influenced and predicted); line 591 (influenced); line 592 (influenced); line 593 (did); line 595 (was); line 597 (has been); line 601 (hypothesized); line 736 (required); line 998 (was).

Suggested changes to past tense have been included.

Copenhagen, 17 April 2024 Koos Boomsma and Joel Vizqueta

Reviewer #2:

Remarks to the Author:

I have read the responses from the authors and the updated manuscript. My comments have been satisfyingly addressed, and the manuscript has been considerably improved in response to all reviewers.

Thank you!

Reviewer #3:

Remarks to the Author:

My concerns were fully resolved or addressed by the authors.

Thank you!

Final Decision Letter:

18th July 2024

Dear Louis,

The two reviewers who saw your revised manuscript are satisfied with the revision (their final comments are below this email). Therefore, we are pleased to inform you that your Article entitled "Larger colony sizes favoured the evolution of more worker castes in ants", has now been accepted for publication in Nature Ecology & Evolution.

Over the next few weeks, your paper will be copyedited to ensure that it conforms to Nature Ecology and Evolution style. Once your paper is typeset, you will receive an email with a link to choose the appropriate publishing options for your paper and our Author Services team will be in touch regarding any additional information that may be required

Due to the importance of these deadlines, we ask you please us know now whether you will be difficult to contact over the next month. If this is the case, we ask you provide us with the contact information (email, phone and fax) of someone who will be able to check the proofs on your behalf, and who will be available to address any last-minute problems . Once your paper has been scheduled for online publication, the Nature press office will be in touch to confirm the details.

Acceptance of your manuscript is conditional on all authors' agreement with our publication policies

64(see www.nature.com/authors/policies/index.html). In particular your manuscript must not be published elsewhere and there must be no announcement of the work to any media outlet until the publication date (the day on which it is uploaded onto our web site).

Please note that *Nature Ecology & Evolution* is a Transformative Journal (TJ). Authors may publish their research with us through the traditional subscription access route or make their paper immediately open access through payment of an article-processing charge (APC). Authors will not be required to make a final decision about access to their article until it has been accepted. Find out more about Transformative Journals

Authors may need to take specific actions to achieve compliance with funder and institutional open access mandates. If your research is supported by a funder that requires immediate open access (e.g. according to Plan S principles) then you should select the gold OA route, and we will direct you to the compliant route where possible. For authors selecting the subscription publication route, the journal's standard licensing terms will need to be accepted, including <https://www.nature.com/nature-portfolio/editorial-policies/self-archiving-and-license-to-publish>. Those licensing terms will supersede any other terms that the author or any third party may assert apply to any version of the manuscript.

We welcome the submission of potential cover material (including a short caption of around 40 words) related to your manuscript; suggestions should be sent to Nature Ecology & Evolution as electronic files (the image should be 300 dpi at 210 x 297 mm in either TIFF or JPEG format). Please note that such pictures should be selected more for their aesthetic appeal than for their scientific content, and that colour images work better than black and white or grayscale images. Please do not try to design a cover with the Nature Ecology & Evolution logo etc., and please do not submit composites of images related to your work. I am sure you will understand that we cannot make any promise as to whether any of your suggestions might be selected for the cover of the journal.

To assist our authors in disseminating their research to the broader community, our SharedIt initiative provides you with a unique shareable link that will allow anyone (with or without a subscription) to

65read the published article. Recipients of the link with a subscription will also be able to download and print the PDF.

You can generate the link yourself when you receive your article DOI by entering it here: <http://authors.springernature.com/share>.

[REDACTED]

Reviewer 1 (Remarks to the Author):

We are satisfied with the authors' responses and with the changes made in the manuscript.

Koos Boomsma and Joel Vizueta

P.S. Click on the following link if you would like to recommend Nature Ecology & Evolution to your librarian <http://www.nature.com/subscriptions/recommend.html#forms>

** Visit the Springer Nature Editorial and Publishing website at www.springernature.com/editorial-and-publishing-jobs for more information about our career opportunities. If you have any questions please click here.**